# PHYSICS-INFORMED INFERENCE TIME SCALING FOR SOLVING HIGH-DIMENSIONAL PARTIAL DIFFERENTIAL EQUATIONS VIA DEFECT CORRECTION

**Zexi Fan**
School of Metathetical Science
Peking University
Beijing, China
`fanzexi_francis@stu.pku.edu.cn`

**Yan Sun & Shihao Yang**
H. Milton Stewart School of Industrial and Systems Engineering
Georgia Institute of Technology
Atlanta, GA, 30332
`shihao.yang@isye.gatech.edu`

**Yiping Lu**
Department of Industrial Engineering and Management Sciences
Northwestern University
Evanston, IL, 60208
`yiping.lu@northwestern.edu`

## ABSTRACT

Solving high-dimensional partial differential equations (PDEs) is a critical challenge where modern data-driven solvers often lack reliability and rigorous error guarantees. We introduce Simulation-Calibrated Scientific Machine Learning (SCaSML), a framework that systematically improves pre-trained PDE solvers at inference time without any retraining. Our core idea is to use defect correction method that derive a new PDE, which we term the `Structural-preserving Law of Defect`, that precisely describes the error of a given surrogate model. Because this defect PDE retains the structure of the original problem, we can solve it efficiently with traditional stochastic simulators, yielding a targeted correction to the initial machine-learned solution. We prove that SCaSML achieves a faster convergence rate, with a final error bounded by the *product* of the surrogate and simulation errors. On challenging PDEs up to 160 dimensions, SCaSML reduces the error of various surrogate models, including PINNs and Gaussian Processes, by 20-80%. SCaSML provides a principled method to fuse the speed of machine learning with the rigor of numerical simulation, enhancing the trustworthiness of AI for scientific discovery.

## 1 INTRODUCTION

Solving high-dimensional partial differential equations (PDEs) is a fundamental challenge across science and engineering. Many critical phenomena are modeled by semi-linear parabolic PDEs whose dimensionality scales with the number of underlying components, a challenge often termed the *curse of dimensionality*. Key examples include the imaginary-time Schrödinger equation in quantum many-body systems, nonlinear Black–Scholes equations in finance, and the Hamilton–Jacobi–Bellman equation in optimal control (Bellman, 1954). Traditional numerical methods, such as finite element and finite difference schemes, become computationally intractable in high dimensions (Larsson & Thomée, 2003). While stochastic simulation methods can be effective, they often suffer from high variance (Briand & Labart, 2014). In response, Scientific Machine Learning (SciML) has emerged as a powerful alternative, using neural networks and other data-driven models to approximate PDE solutions (Karniadakis et al., 2021; Han et al., 2018a; Raissi et al., 2017). However, the "black-box" nature of these models can introduce subtle biases, and they often lack the rigorous error guarantees of their traditional counterparts, raising concerns about their reliability for safety-critical applications.

Recent breakthroughs in large language models (LLMs) have shown that allocating additional computational resources at *inference time* can dramatically improve output quality, a phenomenon

known as inference-time scaling (Snell et al., 2024; Wei et al., 2022). This success inspires our central research question:

*Can we leverage additional computation at inference time to systematically refine and provably improve a pre-trained surrogate model—allocating more compute to harder PDE states just as LLMs spend more search or planning on harder queries—without any retraining or fine-tuning?*

In this work, we provide an affirmative answer by introducing **S**imulation-**Ca**librated **S**cientific **M**achine **L**earning (**SCaSML**), a novel physics-informed framework for improving SciML solvers at inference time. We focus on a broad class of semi-linear parabolic PDEs of the form:

$$\begin{cases} \frac{\partial u}{\partial r} + \mathcal{L}u + F(u, \sigma^\top \nabla u) = 0, & \text{on } [0, T] \times \mathbb{R}^d \\ u(T, \boldsymbol{y}) = g(\boldsymbol{y}), & \text{on } \mathbb{R}^d, \end{cases} \tag{1}$$

where $\mathcal{L}u := \langle \mu, \nabla u \rangle + \frac{1}{2}\text{Tr}(\sigma^\top \text{Hess}(u)\sigma)$ is a second-order linear differential operator. SCaSML operates in two stages. First, a standard SciML solver $\hat{u}$ (e.g., a PINN (Raissi et al., 2017), Gaussian Process (Chen et al., 2021), or Tensor Network (Richter et al., 2021)) is trained to find an approximate solution. At inference time, rather than directly accepting $\hat{u}$, we invoke a defect–correction method (Bank & Weiser, 1985; Stetter, 1978; Böhmer et al., 1984) to derive a governing equation for the approximation error—its **defect**—defined as $\breve{u} := u - \hat{u}$. We term this the `Structural-preserving Law of Defect` (Figure 1). Crucially, unlike classical grid-based defect correction which is intractable in high dimensions, we show that this new PDE describing the exact defect inherits the semi-linear structure of the original problem. This structural preservation allows us to solve it efficiently using well-established stochastic simulation algorithms based on the Feynman–Kac formula. This simulation step acts as a targeted correction, leveraging additional compute to refine the initial surrogate prediction.

Our main contributions are summarized as follows:

- We propose **SCaSML**, the first physics-informed inference-time scaling framework that **improves a pre-trained Scisurrogate model at inference-time, without any retraining or fine-tuning**. SCaSML uses defect correction method that corrects a pre-trained surrogate SCiML model by deriving and solving a new PDE via a branching Monte Carlo Simulation that approximates its error, which we call the `Structural-preserving Law of Defect` (7). Notably, this characterization of the defect is, to our knowledge, the first derivation that preserves the semi-linear structure essential for high-dimensional Monte Carlo solvers.
- We theoretically prove that the final error of SCaSML is bounded by the *product* of the surrogate model's error and the simulation error. Analogous to the classical defect-correction literature—where each correction step systematically improves the convergence rate— **we establish an analogous result for our Monte–Carlo defect-correction procedure at the first time**. The improved convergence rate is corroborated **empirically** in Section 3, with further comprehensive findings presented in Appendix G.3.
- We conduct extensive numerical experiments on challenging high-dimensional PDEs (up to 160 dimensions). Our results show that SCaSML significantly reduces approximation errors by 20-80% across various surrogate models with high statistical significance ($p \ll 0.001$), demonstrating its flexibility, practical efficacy and potential to mitigate the curse of dimensionality. We also demonstrate that, with inference-time scaling, a smaller base PINN can outperform a larger PINN under the same inference-time compute budget by spending its additional computation on targeted refinement rather than parameter count. This enables *elastic compute*: users can trade inference time for accuracy on demand.

## 2  METHODOLOGY

The core of our SCaSML framework is the derivation of a new PDE that describes the error of a pre-trained surrogate model. We term this the `Structural-preserving Law of Defect`. By solving this auxiliary PDE at inference time, we can compute a precise correction to the surrogate's prediction. To build intuition, we first introduce this concept in the context of linear parabolic equations before extending it to the general semi-linear case.

## a) SCaSML Framework Pipeline

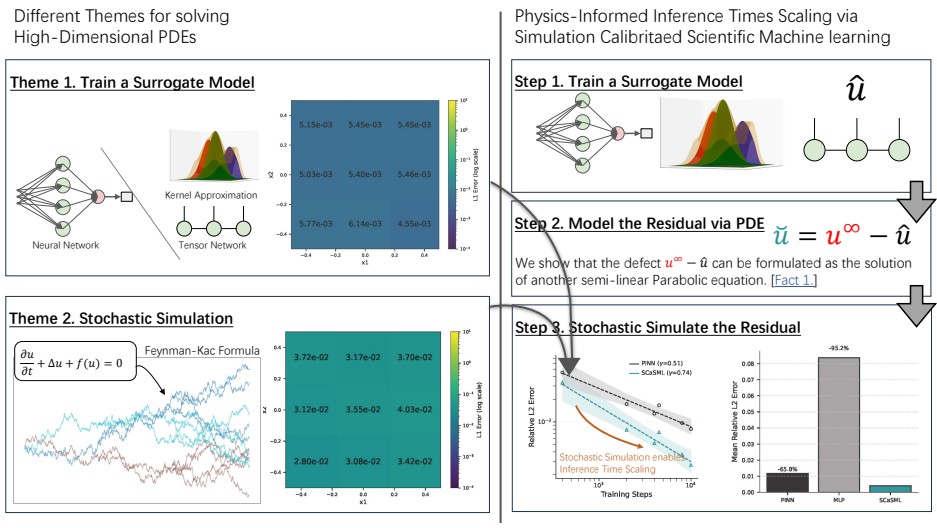

## b) Derivation of the `Structural-preserving Law of Defect`

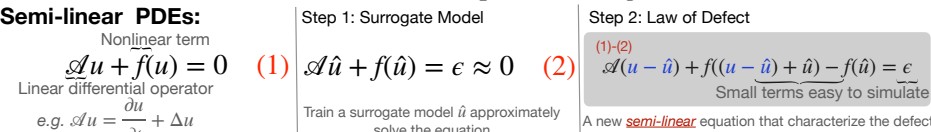

Figure 1: **Overview of the SCaSML framework. (a)** A pre-trained surrogate model $\hat{u}$ provides an initial, approximate solution to the PDE. At inference time, SCaSML calculates the `defect` $\breve{u} = u - \hat{u}$ via a stochastic simulation and adds it back to the surrogate prediction, yielding a more accurate final solution $u = \hat{u} + \breve{u}$. **(b)** The `Structural-preserving Law of Defect` is derived by subtracting the PDE approximately satisfied by the surrogate $\hat{u}$ from the original PDE. This process yields a new semi-linear PDE that describes the defect $\breve{u}$, enabling its estimation through simulation.

### 2.1 WARM UP: THE STRUCTURAL-PRESERVING LAW OF DEFECT FOR LINEAR PARABOLIC PDEs

Let us begin with a high-dimensional linear parabolic PDE, a simpler setting that clarifies our core idea:

$$\begin{cases} \frac{\partial u}{\partial r} + \langle \mu, \nabla_y u \rangle + \frac{1}{2}\mathrm{Tr}\Big(\sigma^\top \mathrm{Hess}_y u\, \sigma\Big) = f(r, \boldsymbol{y}), & \text{on } [0, T) \times \mathbb{R}^d, \\ u(T, \boldsymbol{y}) = g(\boldsymbol{y}), & \text{on } \mathbb{R}^d. \end{cases} \tag{2}$$

Suppose we have a pre-trained surrogate model $\hat{u}$ that approximates the true solution $u$. This surrogate is inevitably imperfect, producing a **residual** when plugged into the PDE. Our goal is to run a defect-correction method at inference time. The defect–correction method (Böhmer et al., 1984; Stetter, 1978) is a classical numerical strategy that improves an approximate solution by formulating and solving an equation for its residual-induced error. The first step is to find a new equation that describes the **defect** $\breve{u}(r, \boldsymbol{y}) := u(r, \boldsymbol{y}) - \hat{u}(r, \boldsymbol{y})$, which represents the true, unknown error. To achieve this, we define this residual as:

$$\epsilon(r, \boldsymbol{y}) := f(r, \boldsymbol{y}) - \left(\frac{\partial \hat{u}}{\partial r} + \langle \mu, \nabla_y \hat{u} \rangle + \frac{1}{2}\mathrm{Tr}\Big(\sigma^\top \mathrm{Hess}_y \hat{u}\, \sigma\Big)\right). \tag{3}$$

By subtracting the equation for $\hat{u}$ from (2), we arrive at the following governing law.

**Definition 2.1** (Structural-preserving Law of Defect for Linear PDEs). *The defect $\breve{u} := u - \hat{u}$ is the solution to the linear parabolic PDE:*

$$\begin{cases} \frac{\partial \breve{u}}{\partial r} + \langle \mu, \nabla_y \breve{u} \rangle + \frac{1}{2}\mathrm{Tr}\Big(\sigma^\top \mathrm{Hess}_y \breve{u}\, \sigma\Big) = \epsilon(r, \boldsymbol{y}), & \text{on } [0, T) \times \mathbb{R}^d, \\ \breve{u}(T, \boldsymbol{y}) = g(\boldsymbol{y}) - \hat{u}(T, \boldsymbol{y}), & \text{on } \mathbb{R}^d. \end{cases} \tag{4}$$

This `Structural-preserving Law of Defect` allows us to solve for the error $\breve{u}$ directly. Since (4) is a linear PDE, its solution can be expressed probabilistically via the Feynman–Kac formula:

$$\breve{u}(s, x) = \mathbb{E}\Big[\big(g(X_T^{s,x}) - \hat{u}(T, X_T^{s,x})\big) + \int_s^T \epsilon(t, X_t^{s,x})dt\Big], \tag{5}$$

where $\{X_t^{s,x}\}_{t\in[s,T]}$ is the stochastic process associated with the PDE's linear operator. This representation allows us to estimate the defect $\breve{u}$ using Monte Carlo simulation.

**Remark 2.2** (Regards Training and Inference Separation)**.** *Training corresponds to solving the PDE globally on the entire domain, learning a map that approximates the solution everywhere. In contrast, inference-time correction solves the PDE only at a specific, user-specified state. This separation is natural and parallels standard practices in machine learning: a base model is trained once to answer all queries, while computationally intensive refinement (like beam search or planning) is invoked at inference time only when high precision is required for a specific input. This separation enables "elastic compute," allowing users to trade inference time for accuracy on demand without incurring the massive fixed cost of retraining the global model.*

**Intuition for Faster Convergence.** Why does this two-step process converge faster? The variance of the Monte Carlo estimator for $\breve{u}$ in (5) depends on the magnitude of the integrand, which is primarily the surrogate's residual $\epsilon$. A more accurate surrogate (i.e., smaller $\epsilon$) leads to lower simulation variance. If the surrogate achieves an error of $e(\hat{u}) \sim m^{-\gamma}$ using $m$ training points, the residual $\epsilon$ will be of a similar order, and the variance of our Monte Carlo estimator will be of order $m^{-2\gamma}$. By averaging over $m$ new Monte Carlo paths at inference time, the final statistical error becomes $\sqrt{m^{-2\gamma}/m} = m^{-\gamma-1/2}$. Thus, for a total budget of $2m$ function evaluations, SCaSML achieves a faster convergence rate than both the surrogate ($m^{-\gamma}$) and a naive Monte Carlo solver ($m^{-1/2}$).

**Why Use Monte Carlo for Correction?** Neural networks and other common surrogates exhibit a *spectral bias*, preferentially learning low-frequency (smooth) components of the solution first (Rahaman et al., 2019). Consequently, the residual error $\epsilon$ is often a high-frequency, irregular function. While challenging for many function approximators, Monte Carlo methods are perfectly suited for this scenario, as their convergence rate is independent of the integrand's smoothness. This makes Monte Carlo an ideal choice for the correction step, as it can efficiently average out the complex error signal left behind by the surrogate model.

## 2.2 EXTENSION TO SEMI-LINEAR PARABOLIC PDEs

We now extend the Monte-Carlo based inference time defect-correction procedure to the general semi-linear PDE in (1). Let $\hat{u}$ be a surrogate solution. We define its residual with respect to the PDE dynamics and the terminal condition as:

$$\begin{cases} \epsilon(r, \boldsymbol{y}) := \frac{\partial \hat{u}}{\partial r} + \mathcal{L}\hat{u} + F(\hat{u}, \sigma^\top \nabla_y \hat{u}), \\ \breve{g}(y) := g(\boldsymbol{y}) - \hat{u}(T, \boldsymbol{y}). \end{cases} \tag{6}$$

By subtracting (6) from the original PDE in (1), we obtain the governing law for the defect $\breve{u} = u - \hat{u}$.

**Fact 2.3** (Structural-preserving Law of Defect for Semi-linear PDEs)**.** *The defect $\breve{u}(r, \boldsymbol{y}) := u(r, \boldsymbol{y}) - \hat{u}(r, \boldsymbol{y})$ is the solution to the following semi-linear parabolic equation:*

$$\begin{cases} \frac{\partial \breve{u}}{\partial r} + \mathcal{L}\breve{u} + \breve{F}(\breve{u}, \sigma^\top \nabla_y \breve{u}) = 0, & on\ [0, T) \times \mathbb{R}^d, \\ \breve{u}(T, \boldsymbol{y}) = \breve{g}(y), & on\ \mathbb{R}^d, \end{cases} \tag{7}$$

*where the modified nonlinear term $\breve{F}$ is given by $\breve{F}(\breve{u}, \sigma^\top \nabla_y \breve{u}) := F(\hat{u} + \breve{u}, \sigma^\top(\nabla_y \hat{u} + \nabla_y \breve{u})) - F(\hat{u}, \sigma^\top \nabla_y \hat{u}) + \epsilon$.*

Notably, the `Structural-preserving Law of Defect` (7) **retains a semi-linear structure**. This is the key property that allows us to apply powerful stochastic solvers, such as the Multilevel Picard (MLP) iteration (Hutzenthaler et al., 2019), to estimate the defect $\breve{u}$ and correct the initial surrogate $\hat{u}$.

**How does the `Structural-preserving Law of Defect` differ from classical defect-correction methods?** Classical finite element methods admit a well-characterized asymptotic error expansion Strang et al. (1973), which enables defect-correction schemes to systematically remove the

leading error term and improve convergence rates (Zienkiewicz & Zhu, 1992a;b; Bank & Weiser, 1985). In contrast, no such asymptotic structure is available for neural networks: NN approximations lack any mesh-refinement hierarchy, their errors do not exhibit a polynomial expansion with respect to a single resolution parameter, and the optimization-induced approximation error provides no perturbative decomposition. A different family of debiasing techniques in numerical PDEs relies on iterative solvers such as Newton methods (Stetter, 1978; Dutt et al., 2000; Xu, 1994; Böhmer, 1981) and quasi-Newton methods (Jameson et al., 1974; Heinrichs, 1996). However, these methods present two fundamental limitations in our setting. First, iterative updates produce only approximate corrections, whereas our law of defect is an exact analytical identity that delivers a closed-form unbiased correction in a single step. Second, embedding iterative methods into a Monte–Carlo or Feynman–Kac framework is highly inefficient: each iteration requires recomputing residuals and Jacobian actions through additional Monte–Carlo estimators, producing a nested simulation hierarchy whose convergence rate rapidly deteriorates—from the standard $\mathcal{O}(N^{-1/2})$ rate for a single Monte–Carlo level, to $\mathcal{O}(N^{-1/4})$ for a second iteration, $\mathcal{O}(N^{-1/8})$ for a third, and so on as more levels are introduced. Practitioners are therefore forced to balance early-termination errors against the rapidly declining statistical efficiency of nested Monte–Carlo estimates, making these approaches both computationally expensive and numerically unstable.

**Practical Scenarios**  In many applications, the quantity of interest is required only at a single state rather than across the full domain. For example, in optimal control and financial pricing (e.g., nonlinear Black–Scholes (Eskiizmirliler et al., 2021; Santos & Ferreira, 2024)), practitioners need the value function and its gradient only at the current state to determine the next action or hedge; forward simulations can then be used to compute the Bellman error and correct the current decision. In rare-event analysis and committor problems in molecular dynamics, neural committor estimators (Khoo et al., 2019; Li et al., 2019; Hua et al., 2024; Lucente et al., 2019) can be refined using a small number of targeted simulations initiated from designated configurations. In goal-oriented estimation (Becker & Rannacher, 1996; 2001), the objective is often a specific functional of the solution rather than the full field. In all such settings, training a surrogate to high global accuracy is computationally wasteful. Our method uses the surrogate for a fast initial approximation and then applies a targeted Monte Carlo refinement at inference time, allocating computational effort precisely where accuracy is needed.

## 2.3 SIMULATING STRUCTURAL–PRESERVING LAW OF DEFECT USING MULTILEVEL PICARD ITERATION

The defect PDE (7) is a semi-linear parabolic equation of the same structural form as (24) with a different closed nonlinear term. Under the standard regularity assumptions, the pair $\breve{\mathbf{u}}^{\infty} = (\breve{u}^{\infty}, [\sigma]^{\top}\nabla_y \breve{u}^{\infty})$ admits the Feynman–Kac and Bismut–Elworthy–Li representations presented in (27)–(28). Hence $\mathbf{u}^{\infty}$ is the fixed point of the expectation operator $\Phi$ on $\mathrm{Lip}([0, T] \times \mathbb{R}^d, \mathbb{R}^{1+d})$:

$$\breve{\mathbf{u}}^{\infty} = \Phi(\breve{\mathbf{u}}^{\infty}), \tag{8}$$

with $\Phi$ given exactly as in the appendix at (28). Intuitively, the operator $\Phi$ is a Feynman–Kac–type backward propagator. Given an approximation of the solution at a future time, $\Phi$ maps it back to the present by running a stochastic simulation forward in time and averaging over all resulting trajectories. Concretely, for each initial state $x$, it computes the expected terminal payoff together with the accumulated contribution of the nonlinearity $\breve{F}$ along the simulated path. The exact solution $u^{\star}$ is therefore characterized as the fixed point of this propagation: inserting $u^{\star}$ into the simulation leaves it unchanged, i.e., $\Phi u^{\star} = u^{\star}$. Standard Picard iteration $\breve{\mathbf{u}}_{k+1} = \Phi(\breve{\mathbf{u}}_k)$ converges to $\mathbf{u}^{\infty}$ under standard regularity assumptions (Yong & Zhou, 1999, Theorem 3.4).

Multilevel Picard (MLP) method (E et al., 2021; Hutzenthaler et al., 2020a), uses Multilevel Monte Carlo (MLMC) (Giles, 2008; 2015) to simulate the telescoping formulation $\mathbb{E}[\breve{\mathbf{u}}_n] = \mathbb{E}[\Phi(\breve{\mathbf{u}}_0)] + \sum_{l=1}^{n-1} \mathbb{E}[\Phi(\breve{\mathbf{u}}_l) - \Phi(\breve{\mathbf{u}}_{l-1})]$. The MLMC method exploits a hierarchy of approximations $\Phi(\breve{\mathbf{u}}_0), \Phi(\breve{\mathbf{u}}_1), \ldots, \Phi(\breve{\mathbf{u}}_n)$, ranging from the coarsest to the finest resolution. Crucially, consecutive approximations $(\Phi(\breve{\mathbf{u}}_l))^{(i)}$ and $(\Phi(\breve{\mathbf{u}}_{l-1}))^{(i)}$ are generated using the same underlying sample path $i$, which induces a strong positive correlation between them. As a result, the variance of their difference is significantly reduced. Moreover, as the level $l$ increases, the iterates converge linearly $\breve{\mathbf{u}}_l - \breve{\mathbf{u}}_{l-1} \to 0$, and the variance of the difference decreases linearly toward zero. As a consequence, the required number of samples $M^{n-l}$ can decrease as $l$ increases, meaning very few expensive samples are needed at the finest levels. The majority of the computational cost is thereby shifted to the coarser levels, significantly reducing the overall complexity of the estimation.

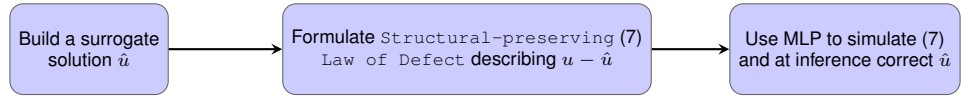

Figure 2: **Flow diagram of SCaSML.** We formulate the error $u - \hat{u}$ of surrogate solution $\hat{u}$ as the solution `Structural-preserving Law of Defect` (7), a new **semi-linear** PDE. At inference time, we approximate $u - \hat{u}$ via solving `Structural-preserving Law of Defect` using Multilevel Picard (MLP) iteration. The generated estimation of $u - \hat{u}$ helps us to calibrate the surrogate solution $\hat{u}$.

Another factor affecting the variance is how the time integral is computed; we used two MLP variants to simulate `Structural-preserving Law of Defect`:

- **Quadrature MLP:** (E et al., 2021) Simulate the time integrals by the Gauss–Legendre quadrature.
- **Full-history MLP:** (Hutzenthaler et al., 2021) Simulate the time integrals by Monte Carlo.

We leave all the preliminaries and implementation details of the MLP methods in Appendix B.2.1. The overall SCaSML procedure, which involves first training a surrogate model and then solving the `Structural-preserving Law of Defect` with MLP methods to correct it, is summarized in Algorithm C.

## 2.4 PROVABLY ACCELERATED CONVERGENCE

We now provide theoretical guarantees for SCaSML, showing that it achieves a provably faster convergence rate. For simplicity, we present results for the case $\mu = \mathbf{0}$ and $\sigma = s\mathbf{I}_d$. Our analysis relies on the assumption that the pre-trained surrogate is reasonably accurate.

**Why SCaSML Enjoys Provable Faster Convergence Rate.** The Monte Carlo error in MLP methods depends on the scale of the terminal defect $\breve{g}$ and the modified nonlinearity $\breve{F}$, which depends on the error of the surrogate model. A more accurate surrogate model yields smaller $\breve{g}$ and $\breve{F}$, resulting in reduced variance during inference. If the surrogate achieves an error of $e(\hat{u}) \sim m^{-\gamma}$ (Assumption 2.4) from $m$ training points, then the variance is $O(m^{-2\gamma})$. During inference, we average over $m$ additional Monte Carlo paths, which reduces the statistical error $\sqrt{\frac{m^{-2\gamma}}{m}} = m^{-\gamma-\frac{1}{2}}$ (Blanchet et al., 2023). With a total computation cost of $2m$ function evaluations, SCaSML therefore attains a convergence rate that surpasses both the surrogate method $m^{-\gamma}$ and the standard MLP / Monte-Carlo estimator $m^{-1/2}$. Full constant-tracking and rigorous proofs are in Appendices F and E.

**Assumption 2.4** (Surrogate Model Accuracy)**.** *Let the true defect be well-behaved such that $\sup_{t\in[0,T]} \|\breve{u}(t,\cdot)\|_{W^{1,\infty}} < \infty$. We assume the surrogate error is bounded by a measure $e(\hat{u})$, such that for constants $C_{F,1}, C_{F,2} > 0$:*

*1. $L^\infty$ **Residual:** $\sup_{r,\boldsymbol{y}} |\epsilon(r,\boldsymbol{y})| \le C_{F,1}\, e(\hat{u})$.*

*2. $W^{1,\infty}$ **Error:** $\sup_r \|\breve{u}(r,\cdot)\|_{W^{1,\infty}} \le C_{F,2}\, e(\hat{u})$.*

**Proof Sketch.** Our main theoretical result stems from the observation that the computational complexity of the MLP solver depends on the Lipschitz constant of the nonlinearity $\breve{F}$ and the magnitude of the "source terms". They appear multiplicatively because nonlinearities—through their Lipschitz bounds—propagate and magnify variance at every Picard iteration. The "source term" driving the Multilevel Picard simulation for the defect is the residual $\epsilon$, already reduced by the surrogate. At the same time, We show that the regularity in the law of defect is no worse than that of the original PDE, ensuring that the refinement introduces no additional smoothness requirements. Combining the previous fact, a more accurate surrogate makes the defect PDE "easier" to solve. This leads to our main error bound.

**Theorem 2.5** (Global $L^2$ Error Bound)**.** *Under standard regularity assumptions on the PDE coefficients (Assumptions E.2–D.7), the global $L^2$ error of the SCaSML estimator using a full-history MLP approximation $\breve{\mathbf{U}}_{N,M}$ with $N$ levels and use $M^l$ Monte Carlo samples at $l$−th level is bounded by:*

$$\sup_{(t,\boldsymbol{x})\in[0,T]\times\mathbb{R}^d} \left\| \breve{\mathbf{U}}_{N,M}(t,\boldsymbol{x}) - \breve{\mathbf{u}}(t,\boldsymbol{x}) \right\|_{L^2} \le E(M,N) \cdot (C_F\, e(\hat{u})), \qquad (9)$$

Table 1: Comparative performance of full-history SCaSML against the surrogate model (**SR**: PINN or GP) and a naive MLP solver. We report total runtime (s) and relative errors in $L^2$, $L^\infty$, and $L^1$ norms. Bold values indicate the best performance in each category. SCaSML consistently achieves the lowest error across nearly all settings.

| Problem | | Time (s) | | | Relative $L^2$ Error | | | $L^\infty$ Error | | | $L^1$ Error | | |
|---|---|---|---|---|---|---|---|---|---|---|---|---|---|
| | | SR | MLP | SCaSML | SR | MLP | SCaSML | SR | MLP | SCaSML | SR | MLP | SCaSML |
| LCD | 10d | 0.45 | 6.77 | 13.31 | 5.20E-02 | 2.27E-01 | **2.74E-02** | 2.50E-01 | 9.06E-01 | **1.65E-01** | 3.39E-02 | 1.67E-01 | **1.78E-02** |
| | 20d | 0.54 | 6.73 | 17.11 | 9.00E-02 | 2.35E-01 | **4.72E-02** | 4.72E-01 | 1.35E+00 | **3.30E-01** | 9.37E-02 | 2.37E-01 | **4.52E-02** |
| | 30d | 0.46 | 6.89 | 22.44 | 1.45E-01 | 2.38E-01 | **9.72E-02** | 2.04E+00 | 1.59E+00 | **7.69E-01** | 1.61E-01 | 2.84E-01 | **1.04E-01** |
| | 60d | 0.28 | 6.94 | 37.59 | 3.13E-01 | 2.39E-01 | **1.32E-01** | 3.24E+00 | 2.05E+00 | **1.57E+00** | 5.35E-01 | 4.07E-01 | **2.06E-01** |
| VB-PINN | 20d | 0.54 | 6.80 | 10.59 | 1.17E-02 | 8.36E-02 | **4.03E-03** | 3.26E-02 | 2.96E-01 | **2.26E-02** | 5.36E-03 | 3.39E-02 | **1.29E-03** |
| | 40d | 0.29 | 8.11 | 14.09 | 4.06E-02 | 1.04E-01 | **2.92E-02** | 8.43E-02 | 3.57E-01 | **7.43E-02** | 2.00E-02 | 4.36E-02 | **1.24E-02** |
| | 60d | 3.14 | 11.36 | 38.30 | 3.95E-02 | 1.17E-01 | **2.88E-02** | 8.20E-02 | 3.93E-01 | **7.20E-02** | 1.94E-02 | 4.82E-02 | **1.22E-02** |
| | 80d | 3.65 | 11.78 | 42.50 | 6.74E-02 | 1.19E-01 | **5.64E-02** | 1.90E-01 | 3.35E-01 | **1.80E-01** | 3.21E-02 | 4.73E-02 | **2.46E-02** |
| VB-GP | 20d | 1.74 | 10.56 | 61.82 | 1.47E-01 | 1.90E-01 | **6.23E-02** | 3.54E-01 | 5.72E-01 | **2.54E-01** | 7.01E-02 | 8.00E-02 | **2.48E-02** |
| | 40d | 1.78 | 12.28 | 61.28 | 1.81E-01 | 2.20E-01 | **8.55E-02** | 4.00E-01 | 8.71E-01 | **3.00E-01** | 9.19E-02 | 9.06E-02 | **3.82E-02** |
| | 60d | 1.68 | 9.70 | 57.79 | 2.40E-01 | 2.57E-01 | **1.28E-01** | 3.84E-01 | 9.50E-01 | **2.84E-01** | 1.27E-01 | 9.99E-02 | **6.11E-02** |
| | 80d | 1.69 | 10.12 | 60.69 | 2.66E-01 | 3.02E-01 | **1.52E-01** | 3.61E-01 | 1.91E+00 | **2.61E-01** | 1.45E-01 | 1.09E-01 | **7.59E-02** |
| LQG | 100d | 0.42 | 8.27 | 21.33 | 7.97E-02 | 5.63E+00 | **5.53E-02** | 7.82E-01 | 1.26E+01 | **6.82E-01** | 1.40E-01 | 1.21E+01 | **8.72E-02** |
| | 120d | 0.32 | 8.52 | 23.98 | 9.40E-02 | 5.50E+00 | **6.66E-02** | 9.06E-01 | 1.27E+01 | **8.06E-01** | 1.74E-01 | 1.22E+01 | **1.06E-01** |
| | 140d | 0.40 | 8.65 | 27.31 | 9.87E-02 | 5.37E+00 | **6.84E-02** | 9.96E-01 | 1.23E+01 | **8.96E-01** | 1.93E-01 | 1.23E+01 | **1.12E-01** |
| | 160d | 0.34 | 8.09 | 29.95 | 1.12E-01 | 5.27E+00 | **9.94E-02** | 1.40E+00 | 1.28E+01 | **1.30E+00** | 2.17E-01 | 1.23E+01 | **1.79E-01** |
| DR | 100d | 0.32 | 7.59 | 58.51 | 1.41E-02 | 8.99E-02 | **1.11E-02** | 9.58E-02 | 6.37E-01 | **8.58E-02** | 1.87E-02 | 9.74E-02 | **1.38E-02** |
| | 120d | 0.33 | 7.16 | 68.28 | 1.11E-02 | 9.13E-02 | **1.03E-02** | 7.50E-02 | 5.74E-01 | **6.50E-02** | 1.39E-02 | 9.97E-02 | **1.29E-02** |
| | 140d | 0.42 | 7.73 | 79.99 | 3.22E-02 | 8.97E-02 | **3.00E-02** | 1.82E-01 | 8.56E-01 | **1.72E-01** | 4.03E-02 | 9.77E-02 | **3.75E-02** |
| | 160d | 0.37 | 7.22 | 86.77 | 3.45E-02 | 9.00E-02 | **3.22E-02** | 2.08E-01 | 8.02E-01 | **1.98E-01** | 4.30E-02 | 9.75E-02 | **4.00E-02** |

*where $\breve{\mathbf{u}} = (\breve{u}, \sigma\nabla_x\breve{u})$ is the true defect and its gradient, and $E(M, N)$ represents the error term of the underlying MLP solver, which depends on $M$ and $N$ but is independent of the surrogate.*

Theorem 2.5 shows that the final error is the product of the MLP simulation error and the surrogate model error. This synergistic relationship implies that the computational cost to reach a global $L^2$ error of $\varepsilon$ is reduced from $O(d\,\varepsilon^{-(2+\delta)})$ for a naive MLP solver to $O(d\,\varepsilon^{-(2+\delta)}\,e(\hat{u})^{2+\delta})$ for SCaSML (see Corollary E.9 in Appendix). This means the cost of our correction step *decreases* as the quality of the initial surrogate improves. This directly leads to an improved scaling law.

**Corollary 2.6** (Improved Scaling Law). *Under the assumptions of Theorem 2.5, suppose the surrogate model's error scales as $e(\hat{u}) = O(m^{-\gamma})$ with $m$ training points. By allocating an additional $m$ samples for the inference-time simulation, the total error of the SCaSML procedure improves from $O(m^{-\gamma})$ to $O(m^{-\gamma-1/2+o(1)})$.*

## 3 NUMERICAL RESULTS

We now empirically validate the SCaSML framework across a suite of challenging high-dimensional PDEs. In each experiment, we first train a baseline surrogate model $\hat{u}$ (either a Physics-Informed Neural Network or a Gaussian Process) to obtain an approximate solution. Then, at inference time, we apply a full-history Multilevel Picard (MLP) solver to the `Structural-preserving Law of Defect` (Fact 2.3) to compute a correction term $\breve{u}$. The final SCaSML solution is the sum $u_{\text{SCaSML}} = \hat{u} + \breve{u}$.

The primary goal of these experiments is to demonstrate the value added by the correction step. Thus, our key comparison is between the baseline surrogate model (SR) and the final corrected solver (SCaSML). We also include the naive MLP solver for reference, to show that the hybrid approach succeeds where pure simulation often fails. Our implementation leverages JAX (Bradbury et al., 2018) and DeepXDE (Lu et al., 2021) for efficient, parallelized computation.

As shown in Figure 3a, SCaSML consistently tightens the error distribution compared to the base surrogate. Refer to Appendix G.6 for detailed pointwise error maps. Figure 3b demonstrates SCaSML's effective inference-time scaling: as more computational resources (i.e., Monte Carlo samples) are allocated, the accuracy of the solution progressively improves. A comprehensive comparison of error metrics and timings is provided in Table 1, and the empirical validation of our theoretical scaling law is shown in Figure 4. **More experiments, including statistical significance tests ($p \ll 0.001$, Appendix G.4) and fixed-budget efficiency comparisons (Appendix G.7), are shown in the Appendix G.**

## 3.1 LINEAR CONVECTION-DIFFUSION EQUATION

**Problem Formulation.** We investigate a linear convection-diffusion equation given by $\frac{\partial}{\partial r}u(r, \boldsymbol{y}) + \left\langle -\frac{1}{d}\mathbf{1}, \nabla_y u(r, \boldsymbol{y}) \right\rangle + \Delta_y u(r, \boldsymbol{y}) = 0, \quad (r, \boldsymbol{y}) \in [0, T] \times \mathbb{R}^d$, with the terminal condition $u(T, \boldsymbol{y}) = \sum_{i=1}^{d} y_i + T, \quad y \in \mathbb{R}^d$. This PDE admits the explicit solution $u(r, \boldsymbol{y}) = \sum_{i=1}^{d} y_i + r$.

**Experimental Setup.** The problem is solved over the hypercube $[0, 0.5] \times [0, 0.5]^d$ for dimensions $d \in \{10, 20, 30, 60\}$ with Dirichlet boundary conditions enforced by the PINN loss. We deploy a Physics-Informed Neural Network (PINN) with 5 hidden layers, 50 neurons each, and a $\tanh$ activation function. Training uses the Adam optimizer (learning rate $7 \times 10^{-4}$, $\beta_1 = 0.9, \beta_2 = 0.99$) for $10^4$ iterations. At each iteration, the network is trained on $2.5 \times 10^3$ interior and $10^2$ boundary collocation points. For the inference step, we use a 2-level simulation with $M = 10$ as the basis of Monte Carlo samples at each level in Multilevel Picard iteration for the tabulated results and $M \in \{10, \ldots, 16\}$ for the scaling study. A clipping threshold of $0.5(d+1)$ is applied to the solution and gradients for both the naive MLP and SCaSML.

**Results.** As reported in Table 1 (**LCD**), SCaSML achieves a reduction in the relative $L^2$ error from 20% to 56.9% compared to the baseline PINN surrogate. Moreover, SCaSML exhibits robust inference scaling, with performance improving as more inference time is allocated (see Figure 10).

## 3.2 VISCOUS BURGERS EQUATION

**Problem Formulation.** Next, we consider a viscous Burgers equation from (Hutzenthaler et al., 2019), a standard benchmark for nonlinear PDEs: $\frac{\partial u}{\partial r} + \left\langle -\left(\frac{1}{d} + \frac{\sigma_0^2}{2}\right)\mathbf{1}, \nabla_y u \right\rangle + \frac{\sigma_0^2}{2}\Delta_y u + \sigma_0 u \sum_{i=1}^{d}(\sigma_0 \nabla_y u)_i = 0$, with terminal condition $u(T, \boldsymbol{y}) = \frac{\exp(T + \sum_{i=1}^{d} y_i)}{1 + \exp(T + \sum_{i=1}^{d} y_i)}$. The exact solution is $u(r, \boldsymbol{y}) = \frac{\exp(r + \sum_{i=1}^{d} y_i)}{1 + \exp(r + \sum_{i=1}^{d} y_i)}$.

**Experimental Setup.** We solve the PDE on $[0, 0.5] \times [-0.5, 0.5]^d$ for dimensions $d \in \{20, 40, 60, 80\}$ with $\sigma_0 = \sqrt{2}$. We test SCaSML with two types of surrogates. The PINN was trained for $10^4$ iterations using the Adam optimizer (learning rate $7 \times 10^{-4}$, $\beta_1 = 0.9$, and $\beta_2 = 0.99$), utilizing 2,500 interior, 100 boundary, and 160 terminal condition sample points. A Gaussian Process (GP) regression surrogate was trained over 20 iterations via Newton's method, using 1,000 interior and 200 boundary points. For the 2-level MLP and SCaSML solvers with the basis of Monte Carlo samples $M = 10$, we set clipping thresholds of 1.0 and 0.01, respectively, to handle the nonlinearity.

**Results.** SCaSML demonstrates strong performance with both surrogate types. For the PINN surrogate (**VB-PINN**), it reduces the relative $L^2$ error by 16.2% to 66.1%. For the GP surrogate (**VB-GP**), the reduction is even more pronounced, ranging from 42.7% to 57.5% (Table 1). This highlights SCaSML's versatility as a plug-and-play corrector for different SciML models.

## 3.3 HIGH-DIMENSIONAL HAMILTON-JACOBI-BELLMAN EQUATION

**Problem Formulation.** To showcase SCaSML on problems central to control theory, we tackle a high-dimensional Hamilton-Jacobi-Bellman (HJB) equation arising from a linear-quadratic-Gaussian (LQG) control problem (Han et al., 2018b). The HJB equation is given by $\frac{\partial u}{\partial r} + \Delta_y u - \|\nabla_y u\|^2 = 0$, with terminal condition $u(T, \boldsymbol{y}) = \log(\frac{1 + \sum_{i=1}^{d-1}[c_{1,i}(y_i - y_{i+1})^2 + c_{2,i}y_{i+1}^2]}{2})$, where $c_{1,i}$ and $c_{2,i}$ are independent random draws from interval $[0.5, 1.5]$. The reference solution is computed via $u(r, \boldsymbol{y}) = -\log \mathbb{E} \exp(-u(T, \boldsymbol{y} + \sqrt{2}W_{T-r}))$ with sufficiently large sample sizes(e.g. $100d$).

**Experimental Setup.** Following (Hu et al., 2024), we use a complex, non-trivial terminal condition and evaluate the problem in very high dimensions, $d \in \{100, 120, 140, 160\}$. The PINN surrogate is trained for $2.5 \times 10^3$ iterations on the domain $[0, 0.5] \times \mathbb{B}^d$, where $\mathbb{B}^d$ is the unit ball in $\mathbb{R}^d$, with 100 interior and 1,000 boundary points per iteration. We use the Adam optimizer with a learning rate of $10^{-3}$, $\beta_1 = 0.9$, and $\beta_2 = 0.99$. For inference steps, we set total level $n = 2$ and the basis of Monte Carlo samples at each level $M = 10$, where $n$ is the total level and $M^l$ is sample used at level $l$ for $0 \le l \le n$. To stabilize the simulation for this strongly nonlinear problem, we use a clipping threshold of 10 for the naive MLP and a much smaller threshold of 0.1 for SCaSML, reflecting

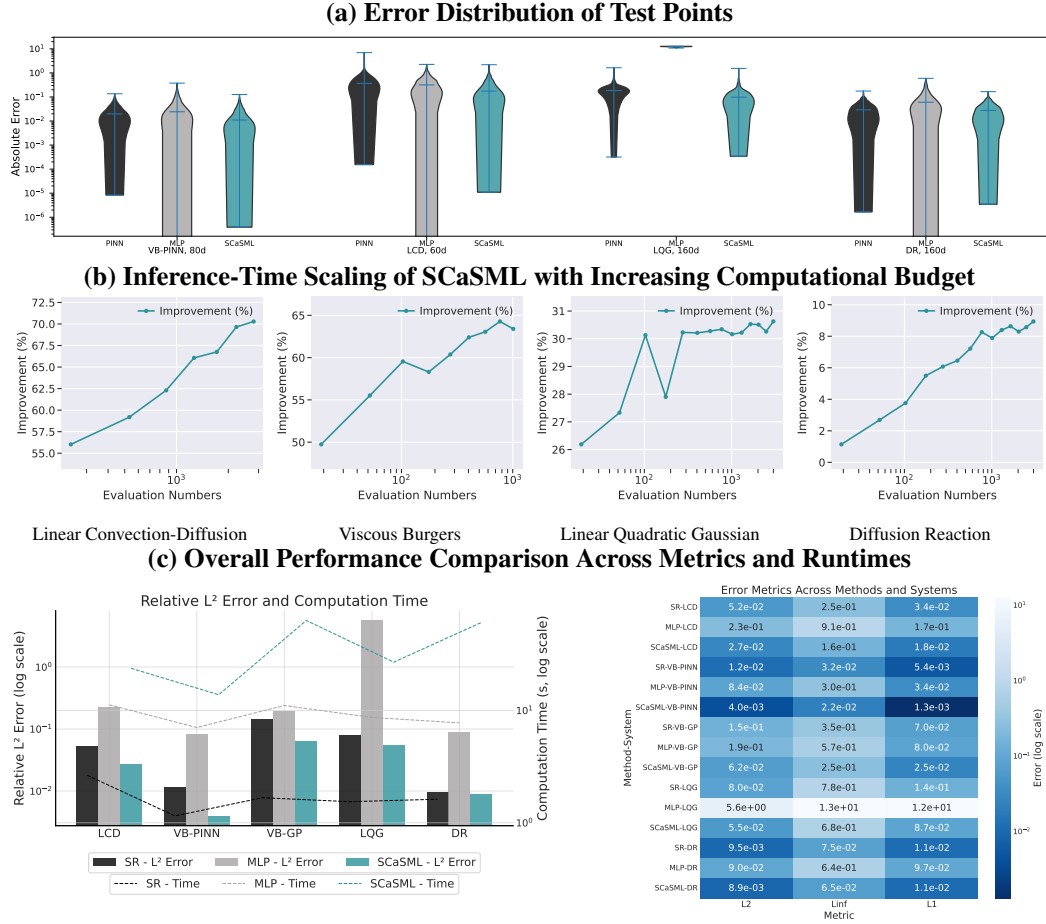

Figure 3: **Efficiency and performance of the SCaSML methodology. (a)** Violin plots showing the distribution of pointwise errors. SCaSML consistently reduces the mean error and tightens the distribution compared to the surrogate (SR) model. **(b)** Inference-time scaling. As the number of inference-time simulation samples increases, SCaSML's error steadily decreases, demonstrating effective use of additional compute. **(c)** Summary of performance. The left panel shows that SCaSML (blue stars) consistently achieves lower $L^2$ error than both the surrogate (SR) and naive MLP methods across all problems. The right panel (heatmap) confirms that SCaSML also dominates in $L^\infty$ and $L^1$ error metrics.

the smaller magnitude of the defect. To accelerate computations, we use Hutchinson's method to stochastically estimate the Laplacian and divergence terms, sampling $d/4$ dimensions at each step (Hutchinson, 1989; Girard, 1989; Shi et al., 2025).

**Results.** In this challenging high-dimensional setting (**LQG**), the naive MLP solver fails entirely, producing large errors. In contrast, SCaSML successfully refines the PINN solution, reducing the relative $L^2$ error by 11.7% to 30.8% and achieving the lowest error across all metrics (Table 1).

### 3.4 DIFFUSION-REACTION EQUATION WITH AN OSCILLATING SOLUTION

**Problem Formulation.** Finally, we consider a diffusion-reaction system designed to have a highly oscillatory solution (Gobet & Turkedjiev, 2017; Han et al., 2018b), making it particularly difficult for standard neural network surrogates $\frac{\partial u}{\partial r} + \frac{1}{2}\Delta_y u + \min\{1, (u - u^\star)^2\} = 0$, where $u^\star(r, \boldsymbol{y}) = 1.6 + \sin(0.1 \sum_{i=1}^{d} y_i) \exp(\frac{0.01d(r-1)}{2})$ is the exact solution.

**Experimental Setup.** We solve the problem for dimensions $d \in \{100, 120, 140, 160\}$ on the domain $[0, 1] \times \mathbb{B}^d$. The PINN surrogate is trained for $2.5 \times 10^3$ iterations with 1,000 interior and 1,000 boundary points, using the Adam optimizer with a learning rate of $10^{-3}$, $\beta_1 = 0.9$, and $\beta_2 = 0.99$. For inference steps, we set total level $n = 2$ and the basis of Monte Carlo samples at each level $M = 10$. The MLP and SCaSML solvers use clipping thresholds of 10 and 0.01, respectively.

Due to the solution's oscillatory nature, we found that the Hutchinson estimator for the Laplacian introduced instability; therefore, we computed the full Laplacian in this experiment.

**Results.** Even though the PINN surrogate is already quite accurate for this problem, SCaSML is still able to provide a consistent refinement. As shown in Table 1 (**DR**), SCaSML further reduces the relative $L^2$ error by 6.6% to 10.9%, demonstrating its capability to improve even well-performing surrogates on complex, high-frequency problems.

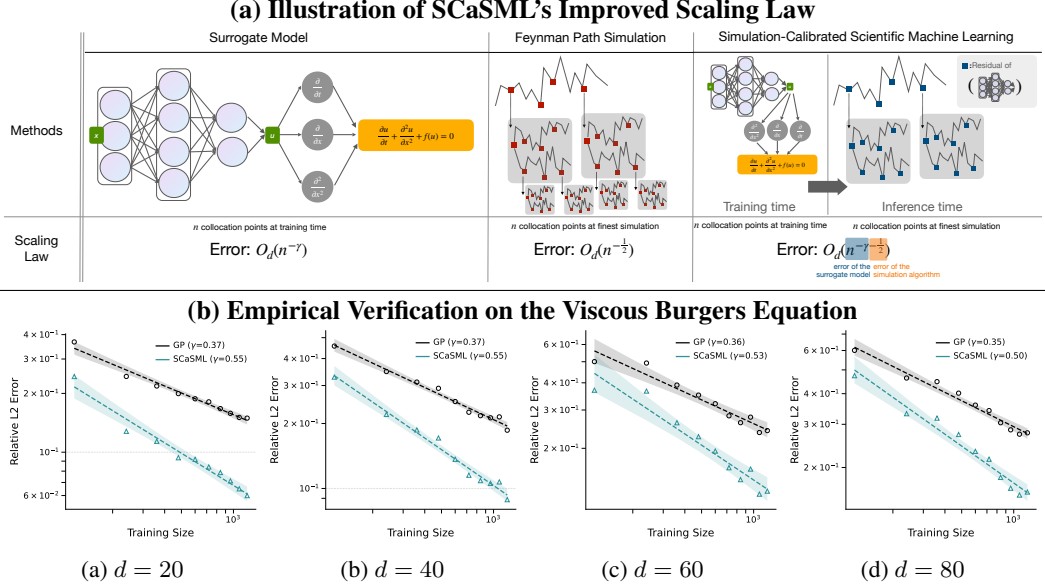

**(a) Illustration of SCaSML's Improved Scaling Law**

**(b) Empirical Verification on the Viscous Burgers Equation**

(a) $d = 20$     (b) $d = 40$     (c) $d = 60$     (d) $d = 80$

Figure 4: **Empirical verification of the improved scaling law for SCaSML. (a)** Conceptual diagram. The final SCaSML error is a product of the surrogate error and the simulation error (Theorem 2.5). By balancing the computational budget between training the surrogate and performing inference-time simulation, SCaSML achieves a faster overall convergence rate (Corollary 2.6). **(b)** Numerical results. We plot the $L^2$ error versus the number of collocation points ($m$) on a log-log scale for a GP surrogate and SCaSML. The slope of the line corresponds to the convergence rate $\gamma$. SCaSML consistently exhibits a steeper slope than the base surrogate, empirically confirming its accelerated convergence.

## 4 CONCLUSION AND DISCUSSION

We introduced **SCaSML**, **the first physics-informed inference time scaling framework** that integrates surrogate models with Monte-Carlo numerical simulations for solving high-dimensional PDE. By introducing `Structural-preserving Law of Defect`, we use the output of a pre-trained SciML solver as an efficient starting point for inference-time corrections. Our theory and experiments show this hybrid approach achieves faster convergence and reduces errors by up to 80% in complex high-dimensional PDEs. SCaSML represents a new approach in hybrid scientific computing. Unlike previous work that used machine learning for discovering numerical schemes (Long et al., 2018) or as preconditioners (Hsieh et al., 2019), our framework uses the machine learning model as a control variate in stochastic simulations to reduce the variance of Monte Carlo simulation. The surrogate handles the low-frequency part, allowing the simulation to focus on the small high-frequency residual, and enhances computational efficiency by addressing model bias at inference time. This establishes an *elastic compute* paradigm, allowing users to trade inference time for accuracy on demand—achieving gains that are often computationally intractable through further training alone. **SCaSML is the first inference-time scaling algorithm that enhances the learned surrogate solution during inference without requiring fine-tuning or retraining.**

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

APPENDIX OUTLINE

This appendix provides supplementary materials to support the main paper. We include a centralized notation glossary, detailed background on the methods used, the complete algorithm, full proofs of our theoretical results, and extensive additional experimental validation including statistical significance tests and computational budget analyses.

The appendix is organized as follows:

- **Appendix A: Notations.** We provide a centralized glossary defining the mathematical symbols and operators used throughout the paper and appendices.

- **Appendix B: Preliminaries.** We provide background on the core technical components of the SCaSML framework.

  - *Surrogate Models for PDEs (§B.1):* We detail the architectures used: Physics-Informed Neural Networks (PINNs) and Gaussian Processes (GPs).

  - *Multilevel Picard (MLP) Iterations (§B.2):* We overview the quadrature and full-history MLP methods which form the basis of our inference-time correction.

- **Appendix C: Algorithm.** We present the complete SCaSML algorithm in detailed pseudocode, including practical implementation details like outlier thresholding and Hutchinson's estimator for high-dimensional Laplacians.

- **Appendix D: Proof Settings.** We establish the common probability space definitions and specific regularity assumptions on the surrogate models required for our theoretical analysis.

- **Appendix E: Proof for Full-History MLP.** We provide the theoretical analysis for the full-history MLP variant using Monte Carlo time integration.

  - *Global $L^2$ Error Bound (§E.3.2):* We derive the error bound for the full-history case.

  - *Improved Scaling Law (§E.3.3):* We provide the proof for the accelerated asymptotic convergence rate of $O(m^{-\gamma-1/2})$.

- **Appendix F: Proof for Quadrature MLP.** We present the theoretical analysis for SCaSML using the quadrature MLP solver.

  - *Global $L^2$ Error Bound (§F.2.1):* We derive the error bound showing dependence on the surrogate's accuracy $e(\hat{u})$.

  - *Computational Complexity (§F.2.2):* We prove the reduction in complexity afforded by the SCaSML framework.

- **Appendix G: Auxiliary Experimental Results.** We include comprehensive additional experiments to validate robustness, statistical significance, and efficiency.

  - *Violin Plots for Error Distribution (§G.1):* Visualizations of the full error distribution for all test cases.

  - *Inference Time Scaling Curves (§G.2 and §G.3):* Plots demonstrating monotonic error reduction with increased inference compute.

  - *Statistical Analysis of $L^1$ Errors (§G.4):* Detailed tables reporting means, standard deviations, 95% confidence intervals, and p-values from paired t-tests.

  - *Relative $L^2$ Error Improvement (§G.5):* Visualization of the percentage error reduction across all dimensions.

  - *Pointwise Error Reduction Analysis (§G.6):* Scatter plots confirming that SCaSML systematically reduces error on the vast majority of individual test points.

  - *Performance Comparison Under Fixed Computational Budgets (§G.7):* A Pareto efficiency analysis comparing SCaSML to baselines when total wall-clock time (Training + Inference) is held constant.

  - *Performance Comparison: Large PINN vs. SCaSML Correction (§G.8):* A Pareto efficiency analysis comparing SCaSML to PINN with increasing scales of the same computing budget.

# A   NOTATION

This section establishes the rigorous mathematical framework, including probability spaces, function classes, and norms used throughout the theoretical analysis. We strictly distinguish between spatial functional norms and probabilistic norms to ensure clarity in the convergence analysis.

## A.1   GENERAL CONVENTIONS AND GEOMETRY

- Let $T \in (0, \infty)$ be a fixed terminal time. We define the spatiotemporal domain as $\Omega_T := [0, T] \times \mathbb{R}^d$, where $d \in \mathbb{N}$ denotes the spatial dimension.
- We adopt the unified coordinate convention $(t, \boldsymbol{x}) \in \Omega_T$ throughout this appendix. The notation $(r, y)$ is reserved strictly for integration variables within time integrals.
- $\mathcal{B}(\mathbb{R}^d)$ denotes the Borel $\sigma$-algebra on $\mathbb{R}^d$.
- $\langle \boldsymbol{x}, \boldsymbol{y} \rangle$ denotes the standard Euclidean inner product for $\boldsymbol{x}, \boldsymbol{y} \in \mathbb{R}^d$, and $|\boldsymbol{x}| := \sqrt{\langle \boldsymbol{x}, \boldsymbol{x} \rangle}$ denotes the Euclidean norm.

## A.2   NORMS AND FUNCTION SPACES

- **Measurable Functions:** Let $\mathcal{M}(A, B)$ denote the set of all measurable functions mapping from measurable space $A$ to $B$.
- **Spatial Spaces and Norms:**
    - For any function $\phi : \mathbb{R}^d \to \mathbb{R}^k$, we define the uniform norm $\|\phi\|_\infty := \sup_{\boldsymbol{x} \in \mathbb{R}^d} |\phi(\boldsymbol{x})|$.
    - $C^{1,2}([0, T] \times \mathbb{R}^d)$ denotes the space of functions that are once continuously differentiable in time and twice continuously differentiable in space.
    - $W^{k,\infty}(\mathbb{R}^d)$ denotes the Sobolev space of functions with essentially bounded weak derivatives up to order $k$, equipped with the norm $\|\phi\|_{W^{k,\infty}} := \sum_{|\alpha| \le k} \|D^\alpha \phi\|_\infty$.
- **Probabilistic Spaces and Norms:**
    - Let $(\Omega, \mathcal{F}, \mathbb{P})$ be a complete probability space equipped with a filtration $(\mathbb{F}_t)_{t \in [0, T]}$ satisfying the usual conditions.
    - For $p \in [1, \infty)$, $L^p(\Omega; \mathbb{R}^k)$ denotes the Lebesgue space of random variables $X : \Omega \to \mathbb{R}^k$ with finite $p$-th moment. We explicitly define the probabilistic norm:

$$\|X\|_{L^p(\Omega)} := \left( \mathbb{E}\left[ |X|^p \right] \right)^{1/p}.$$

## A.3   PDE FORMULATION AND STOCHASTIC PROCESSES

- **The Operator:** Let $\mu : [0, T] \times \mathbb{R}^d \to \mathbb{R}^d$ and $\sigma : [0, T] \times \mathbb{R}^d \to \mathbb{R}^{d \times d}$. We define the second-order linear differential operator $\mathcal{L}$ acting on $\phi \in C^{1,2}$ as:

$$\mathcal{L}\phi(t, \boldsymbol{x}) := \langle \mu(t, \boldsymbol{x}), \nabla_{\boldsymbol{x}} \phi(t, \boldsymbol{x}) \rangle + \frac{1}{2} \mathrm{Tr}\left( \sigma(t, \boldsymbol{x}) \sigma(t, \boldsymbol{x})^\top \mathrm{Hess}_x \phi(t, \boldsymbol{x}) \right).$$

- **The SDE:** For any $(t, \boldsymbol{x}) \in \Omega_T$, let $X^{t, \boldsymbol{x}} = (X_s^{t, \boldsymbol{x}})_{s \in [t, T]}$ be the unique strong solution to the stochastic differential equation (SDE):

$$X_s^{t, \boldsymbol{x}} = x + \int_t^s \mu(r, X_r^{t, \boldsymbol{x}}) dr + \int_t^s \sigma(r, X_r^{t, \boldsymbol{x}}) dW_r, \quad s \in [t, T],$$

where $W$ is a standard $d$-dimensional Brownian motion under $\mathbb{P}$.

## A.4   DEFECT FORMULATION (THE SCASML OBJECT)

- **Surrogate and Defect:** Let $\hat{u} \in C^{1,2}(\Omega_T)$ be the surrogate solution. We define the **defect** pointwise as $\breve{u}(t, \boldsymbol{x}) := u(t, \boldsymbol{x}) - \hat{u}(t, \boldsymbol{x})$.
- **The Residual:** We define the PDE residual $\epsilon : \Omega_T \to \mathbb{R}$ as:

$$\epsilon(t, \boldsymbol{x}) := \frac{\partial \hat{u}}{\partial t}(t, \boldsymbol{x}) + \mathcal{L}\hat{u}(t, \boldsymbol{x}) + F(\hat{u}(t, \boldsymbol{x}), \sigma(t, \boldsymbol{x})^\top \nabla_{\boldsymbol{x}} \hat{u}(t, \boldsymbol{x})).$$

- **Modified Nonlinearity** $\breve{F}$: Let $\breve{F} : \Omega_T \times \mathbb{R} \times \mathbb{R}^d \to \mathbb{R}$ be the modified driver defined by:
$$\breve{F}(v, \boldsymbol{z}, t, \boldsymbol{x}) := F(\hat{u}(t, \boldsymbol{x}) + v, \sigma^\top (\nabla_{\boldsymbol{x}} \hat{u}(t, \boldsymbol{x}) + \boldsymbol{z})) - F(\hat{u}(t, \boldsymbol{x}), \sigma^\top \nabla_{\boldsymbol{x}} \hat{u}(t, \boldsymbol{x})) + \epsilon(t, \boldsymbol{x}).$$
Crucially, observe that $\breve{F}(\boldsymbol{0_{d+1}}, t, \boldsymbol{x}) = \epsilon(t, \boldsymbol{x})$. This identity bridges the deterministic approximation error and the stochastic driver.
- **Terminal Defect:** $\breve{g}(\boldsymbol{x}) := g(\boldsymbol{x}) - \hat{u}(T, \boldsymbol{x})$.

## B  PRELIMINARY

In this section, we provide the necessary background on the two main building blocks of the SCaSML framework. First, we detail the surrogate models—Physics-Informed Neural Networks and Gaussian Processes—used to generate the initial approximate solution $\hat{u}$. Second, we review the Multilevel Picard (MLP) iteration method, the numerical solver we employ at inference time to solve the `Structural-preserving Law of Defect`.

### B.1  SURROGATE MODELS FOR PDEs

In our experiments, we employ two surrogate models to solve high-dimensional PDEs: a Physics-Informed Neural Network (PINN) and a Gaussian Process (GP) regression model. Both models are implemented in `JAX` (Bradbury et al., 2018) and `DeepXDE` (Lu et al., 2021) to leverage efficient parallelization and runtime performance. Furthermore, Hutchinson's estimator technique 3 as delineated in (Shi et al., 2025) is incorporated during the training process to substantially decrease GPU memory consumption, applicable to both the training and inference stages of Physics-Informed Neural Networks (PINN), as well as the inference phase of Gaussian Processes.

#### B.1.1  PHYSICS-INFORMED NEURAL NETWORK (PINN)

Physics-Informed Neural Networks (PINNs) are designed to approximate solutions of PDEs by embedding physical laws into the learning process. In our framework, the neural network $\hat{u}(t, \boldsymbol{x})$ with parameters $\theta$ approximates the true solution $u^\infty(t, \boldsymbol{x})$ of the given PDE. The training loss is constructed as a weighted sum of several components, each designed to enforce key aspects of the problem's constraints.

The first component is the PDE loss, which ensures that the network output adheres to the governing differential equation. This is achieved by penalizing deviations from the expected behavior defined by the differential operator, evaluated at a set of interior collocation points $\{(t_k, x_k)\}_{k=1}^{S_1}$. The PDE loss is defined as

$$\mathcal{L}_{\text{PDE}}(\theta) = \frac{1}{S_1} \sum_{k=1}^{S_1} \left| \frac{\partial \hat{u}}{\partial r}(t_k, x_k) + \frac{\sigma^2}{2} \Delta_y \hat{u}(t_k, x_k) + F(\hat{u}, \sigma \nabla_y \hat{u})(t_k, x_k) \right|^2. \tag{10}$$

In order to satisfy the prescribed boundary conditions, the model employs a Dirichlet boundary loss. This term minimizes the difference between the network output and the given boundary values $h(x_k)$ at selected boundary points $\{(t_k, x_k)\}_{k=1}^{S_2}$, and is expressed as

$$\mathcal{L}_{\text{Dir}}(\theta) = \frac{1}{S_2} \sum_{k=1}^{S_2} |\hat{u}(t_k, x_k) - h(x_k)|^2. \tag{11}$$

Moreover, the initial conditions of the problem are enforced by an initial loss component. This ensures that the solution at time $t = 0$ matches the known initial data $q(x_k)$ for the points $\{(0, x_k)\}_{k=1}^{S_3}$:

$$\mathcal{L}_{\text{initial}}(\theta) = \frac{1}{S_3} \sum_{k=1}^{S_3} |\hat{u}(0, x_k) - q(x_k)|^2. \tag{12}$$

The overall training objective is then formulated as a combination of these losses, with each term scaled by its corresponding weighting coefficient:
$$\mathcal{L}(\theta) = \alpha_1 \, \mathcal{L}(\theta) + \alpha_2 \, \mathcal{L}_{\text{Dir}}(\theta) + \alpha_3 \, \mathcal{L}_{\text{initial}}(\theta). \tag{13}$$
This formulation ensures that the PINN not only fits the observed data but also rigorously respects the underlying physical laws, boundary conditions, and initial conditions governing the PDE.

### B.1.2 GAUSSIAN PROCESSES

In this section, we review the Gaussian Process (GP) framework developed in (Chen et al., 2021; Yang et al., 2021; Chen et al., 2024) to solve nonlinear PDEs. Consider solving a semi-linear parabolic PDE

$$\begin{cases} \dfrac{\partial u}{\partial t}(t, \boldsymbol{x}) = \tau\big(u(t, \boldsymbol{x}), \Delta_{\boldsymbol{x}} u(t, \boldsymbol{x}), \operatorname{div}_x u(t, \boldsymbol{x})\big), & \forall (t, \boldsymbol{x}) \in [0, T] \times \mathbb{R}^d, \\ u(T, \boldsymbol{x}) = g(\boldsymbol{x}), & \forall x \in \mathbb{R}^d, \end{cases} \tag{14}$$

where $\tau$ is a nonlinear function of the solution and its derivatives, and $g$ specifies the terminal condition.

**The GP Framework**   Consider one already sample $M_{\text{in}}$ interior points and $M_{\text{bd}}$ boundary points, denoted as $\mathbf{x}_{\text{in}} = \{\mathbf{x}_{\text{in}}^1, ..., \mathbf{x}_{\text{in}}^{M_{\text{in}}}\} \subset [0, T] \times \mathbb{R}^d$ and $\mathbf{x}_{\text{bd}} = \{\mathbf{x}_{\text{bd}}^1, ..., \mathbf{x}_{\text{bd}}^{M_{\text{bd}}}\} \subset \{T\} \times \mathbb{R}^d$. Then, we assign an unknown GP prior to the unknown function $u$ with mean $0$ and covariance function $K : ([0, T] \times \mathbb{R}^d) \times ([0, T] \times \mathbb{R}^d) \to \mathbb{R}$, the method aims to compute the maximum a posterior estimator of the GP given the sampled PDE data, which leads to the following optimization problem

$$\begin{cases} \underset{u \in \mathcal{U}}{\text{minimize}} & \|u\| \\ \text{s.t.} & \dfrac{\partial u}{\partial t}(\mathbf{x}_{\text{in}}^m) = \tau(u(\mathbf{x}_{\text{in}}^m), \Delta_{\boldsymbol{x}} u(\mathbf{x}_{\text{in}}^m), \operatorname{div}_x u(\mathbf{x}_{\text{in}}^m)), & \text{for} \quad m = 1, \ldots, M_{\text{in}}, \\ & u(\mathbf{x}_{\text{bd}}^m) = g(\mathbf{x}_{\text{bd}}^m), \quad \text{for} \quad m = 1, \ldots, M_{\text{bd}}. \end{cases} \tag{15}$$

Here, $\|\cdot\|$ is the Reproducing Kernel Hilbert Space(RKHS) norm corresponding to the kernel/covariance function $K$. Regarding consistency, once $K$ is sufficiently regular, the above solution will converge to the exact solution of the PDE when $M_{\text{in}}, M_{\text{bd}} \to \infty$; see (Batlle et al., 2023, Theorem 1.2).

We denote the measurement functions by

$$\phi_m^1(u) : u \to \delta_{\mathbf{x}_{\text{in}}^m} \circ u, 1 \le m \le M_{\text{in}}, \qquad \phi_m^2(u) : u \to \delta_{\mathbf{x}_{\text{bd}}^m} \circ u, 1 \le m \le M_{\text{bd}},$$

$$\phi_m^3(u) : u \to \delta_{\mathbf{x}_{\text{in}}^m} \circ \Delta_{\boldsymbol{x}} u, 1 \le m \le M_{\text{in}}, \qquad \phi_m^4(u) : u \to \delta_{\mathbf{x}_{\text{in}}^m} \circ \frac{\partial u}{\partial t}, 1 \le m \le M_{\text{in}}, \tag{16}$$

$$\phi_m^5(u) : u \to \delta_{\mathbf{x}_{\text{in}}^m} \circ \operatorname{div}_x u, 1 \le m \le M_{\text{in}},$$

where $\delta_{\mathbf{x}}$ is the Dirac delta function centered at $\mathbf{x}$. These functions belong to $\mathcal{U}^\top$, the dual space of $\mathcal{U}$, for sufficiently regular kernel functions. We further use the shorthand notation $\phi^1, \phi^3, \phi^4, \phi^5$ for $M_{\text{in}}$ dimensional vectors and $\phi^2$ for $M_{\text{bd}}$ dimensional vectors as finite dimensional representation for corresponding features. We use $[\cdot, \cdot]$ to denote the primal-dual pairing, such that for $u \in \mathcal{U}$ and $\phi_m^i \in \mathcal{U}^\top, \forall i$ it holds that $[u, \phi_m^i] = \int u(\mathbf{x})\phi_m^i(\mathbf{x})d\mathbf{x}$. For instance, for $\phi_m^3$ we have $[u, \phi_m^3] = \int u(\mathbf{x})\phi_m^3(\mathbf{x})d\mathbf{x} = \frac{\partial u}{\partial t}(\mathbf{x}_m)$. Based on the defined notation, we can rewrite the MAP problem (15) as

$$\begin{cases} \underset{u \in \mathcal{U}}{\text{minimize}} & \|u\| \\ \text{s.t.} & z_m^{(1)} = \phi_m^{(1)}(u), z_m^{(3)} = \phi_m^{(3)}(u), z_m^{(4)} = \phi_m^{(4)}(u), z_m^{(5)} = \phi_m^{(5)}(u), \quad m = 1, \ldots, M_{\text{in}}, \\ & z_m^{(1)} = \phi_m^{(1)}(u), m = 1, \ldots, M_{\text{bd}}, \\ & z_m^{(4)} = \tau(z_m^{(1)}, z_m^{(3)}, z_m^{(5)}), \quad m = 1, \ldots, M_{\text{in}}, \\ & z_m^{(2)} = g(\mathbf{x}_{bd}^m), \quad m = 1, \ldots, M_{\text{bd}}. \end{cases} \tag{17}$$

**Finite Dimensioanl Representation via Representer Theorem**   According to Representer Theorem (Chen et al., 2021; Unser, 2021) show that although the original MAP problem (15) is an infinite-dimensional optimization problem, the minimizer enjoys a finite-dimensional structure

$$u^\dagger(\mathbf{x}) = K(\mathbf{x}, \phi)\alpha \tag{18}$$

where $K(\mathbf{x}, \phi)$ is the $(4M_{\text{in}} + M_{\text{bd}})$ dimensional vector with entries $\int K(\mathbf{x}, \mathbf{x}')\phi_j(\mathbf{x}')d\mathbf{x}'$ (here the integral notation shall be interpreted as the primal-dual pairing as above), *i.e.*

$$K(\mathbf{x}, \phi) = \begin{bmatrix} K(\mathbf{x}, \mathbf{x}_{\text{in}}) & K(\mathbf{x}, \mathbf{x}_{\text{bd}}) & \Delta_{x'} K(\mathbf{x}, \mathbf{x}_{\text{in}}) & \frac{\partial}{\partial t} K(\mathbf{x}, \mathbf{x}_{\text{in}}) & \operatorname{div}_{x'} K(\mathbf{x}, \mathbf{x}_{\text{in}}) \end{bmatrix} \in \mathbb{R}^{1 \times (4M_{\text{in}} + M_{\text{bd}})}, \tag{19}$$

and $\alpha \in \mathbb{R}^{4M_{\text{in}} + M_{\text{bd}}}$ is the unknown coeficients. Based on the finite dimensional representation (18), we know

$$\left[ z^{(1)\top}, z^{(2)\top}, z^{(3)\top}, z^{(4)\top}, z^{(5)\top} \right]^\top = K(\phi, \phi)\alpha, \tag{20}$$

where $z^{(1)} = [\phi_1^1(u), \phi_2^1(u), \cdots, \phi_{M_{\text{in}}}^1(u)]^\top \in \mathbb{R}^{M_{\text{in}}}$, $z^{(2)} = [\phi_1^2(u), \phi_2^2(u), \cdots, \phi_{M_{\text{bd}}}^2(u)]^\top \in \mathbb{R}^{M_{\text{bd}}}$, $z^{(3)} = [\phi_1^3(u), \phi_2^3(u), \cdots, \phi_{M_{\text{in}}}^3(u)]^\top \in \mathbb{R}^{M_{\text{in}}}$, $z^{(4)} = [\phi_1^4(u), \phi_2^4(u), \cdots, \phi_{M_{\text{in}}}^4(u)]^\top \in \mathbb{R}^{M_{\text{in}}}$, $z^{(5)} = [\phi_1^5(u), \cdots, \phi_{M_{\text{in}}}^5(u)]^\top \in \mathbb{R}^{M_{\text{in}}}$, and $K(\phi, \phi)$ is the kernel matrix as the $(4M_{\text{in}} + M_{\text{bd}}) \times (4M_{\text{in}} + M_{\text{bd}})$ matrix with entries $\int K(\mathbf{x}, \mathbf{x}')\phi_m(\mathbf{x})\phi_j(\mathbf{x}')d\mathbf{x}d\mathbf{x}'$ where $\phi_m$ denotes the entries of $\phi$. Precisely $K(\phi, \phi)$ can be written down explicitly as:

$$K(\phi, \phi) = \begin{bmatrix} K(\mathbf{x}_{\text{in}}, \mathbf{x}'_{\text{in}}) & K(\mathbf{x}_{\text{in}}, \mathbf{x}'_{\text{bd}}) & \Delta_{x'} K(\mathbf{x}_{\text{in}}, \mathbf{x}'_{\text{in}}) & \frac{\partial}{\partial t} K(\mathbf{x}_{\text{in}}, \mathbf{x}'_{\text{in}}) & \text{div}_{x'} K(\mathbf{x}_{\text{in}}, \mathbf{x}'_{\text{in}}) \\ K(\mathbf{x}_{\text{bd}}, \mathbf{x}'_{\text{in}}) & K(\mathbf{x}_{\text{bd}}, \mathbf{x}'_{\text{bd}}) & \Delta_{x'} K(\mathbf{x}_{\text{bd}}, \mathbf{x}'_{\text{in}}) & \frac{\partial}{\partial t} K(\mathbf{x}_{\text{bd}}, \mathbf{x}'_{\text{bd}}) & \text{div}_{x'} K(\mathbf{x}_{\text{bd}}, \mathbf{x}'_{\text{in}}) \\ \Delta_{\boldsymbol{x}} K(\mathbf{x}_{\text{in}}, \mathbf{x}'_{\text{in}}) & \Delta_{\boldsymbol{x}} K(\mathbf{x}_{\text{in}}, \mathbf{x}'_{\text{bd}}) & \Delta_{\boldsymbol{x}}\Delta_{x'} K(\mathbf{x}_{\text{in}}, \mathbf{x}'_{\text{in}}) & \Delta_{\boldsymbol{x}}\frac{\partial}{\partial t} K(\mathbf{x}_{\text{in}}, \mathbf{x}'_{\text{in}}) & \Delta_{\boldsymbol{x}}\text{div}_{x'} K(\mathbf{x}_{\text{in}}, \mathbf{x}'_{\text{in}}) \\ \frac{\partial}{\partial t} K(\mathbf{x}_{\text{in}}, \mathbf{x}'_{\text{in}}) & \frac{\partial}{\partial t} K(\mathbf{x}_{\text{in}}, \mathbf{x}'_{\text{bd}}) & \frac{\partial}{\partial t}\Delta_{x'} K(\mathbf{x}_{\text{in}}, \mathbf{x}'_{\text{in}}) & \frac{\partial}{\partial t}\frac{\partial}{\partial t} K(\mathbf{x}_{\text{in}}, \mathbf{x}'_{\text{in}}) & \frac{\partial}{\partial t}\text{div}_{x'} K(\mathbf{x}_{\text{in}}, \mathbf{x}'_{\text{in}}) \\ \text{div}_x K(\mathbf{x}_{\text{in}}, \mathbf{x}'_{\text{in}}) & \text{div}_x K(\mathbf{x}_{\text{in}}, \mathbf{x}'_{\text{bd}}) & \text{div}_x \Delta_{x'} K(\mathbf{x}_{\text{in}}, \mathbf{x}'_{\text{in}}) & \text{div}_x \frac{\partial}{\partial t} K(\mathbf{x}_{\text{in}}, \mathbf{x}'_{\text{in}}) & \text{div}_x \text{div}_{x'} K(\mathbf{x}_{\text{in}}, \mathbf{x}'_{\text{in}}) \end{bmatrix}, \tag{21}$$

Here we adopt the convention that if the variable inside a function is a set, it means that this function is applied to every element in this set; the output will be a vector or a matrix, *e.g.* $K(\mathbf{x}_{\text{in}}, \mathbf{x}'_{\text{in}}) = \exp \left( - \frac{\|\mathbf{x}_{\text{in}}^m - \mathbf{x}_{\text{in}}^j\|_2^2}{2(\sigma\sqrt{d})^2} \right), 1 \le m, j \le M_{\text{in}}, \in \mathbb{R}^{M_{\text{in}} \times M_{\text{in}}}$ in the Gaussian kernel of our numerical experiment, where $\sigma$ is the variance of the equation. Thus the finite dimensional representation (18) can be rewritten in terms of the function (derative) values

$$u^\dagger(\mathbf{x}) = K(\mathbf{x}, \phi)K(\phi, \phi)^{-1}z^\dagger, \tag{22}$$

where $z^\dagger = \left[ z^{(1)\top}, z^{(2)\top}, z^{(3)\top}, z^{(4)\top}, z^{(5)\top} \right]^\top \in \mathbb{R}^{4M_{\text{in}} + M_{\text{bd}}}$.

Plug the finite-dimensional representation (22) to the original MAP problem (17) we have that $z^\dagger$ is the solution to the following finite-dimensional quadratic optimization optimization problem with nonlinear constraints

$$\min_{z \in \mathbb{R}^{4M_{\text{in}} + M_{\text{bd}}}} z^\top K(\phi, \phi)^{-1} z$$

subject to

$$z_m^{(4)} = \tau(z_m^{(1)}, z_m^{(3)}, z_m^{(5)}), \quad m = 1, \ldots, M_{\text{in}}, \tag{23}$$

$$z_m^{(2)} = g(\mathbf{x}_{bd}^m), \quad m = 1, \ldots, M_{\text{bd}}.$$

**Solving the Optimization Formulation**   To develop efficient optimization algorithms for (23), observing that the constraints $z_m^{(4)} = \tau\left(z_m^{(1)}, z_m^{(3)}, z_m^{(5)}\right)$   and   $z_m^{(2)} = g\left(\mathbf{x}_{bd}^m\right)$ express $z_m^{(4)}$ and $z_m^{(2)}$ in terms of the other variables, (Chen et al., 2021; 2024) reformulate the optimization problem as an unconstrained problem

$$\min_{z^{(1)}, z^{(3)}, z^{(5)} \in \mathbb{R}^{M_{\text{in}}}} [z^{(1)}; g(\mathbf{x}_{bd}); z^{(3)}; \tau(z^{(1)}, z^{(3)}, z^{(5)}); z^{(5)}]^\top K(\phi, \phi)^{-1} [z^{(1)}; g(\mathbf{x}_{bd}); z^{(3)}; \tau(z^{(1)}, z^{(3)}, z^{(5)}); z^{(5)}].$$

We apply Sparse Cholesky decomposition to the positive-definite $(K(\phi, \phi) + \eta I)$ as $LL^T$. In turn, $b^T(K(\phi, \phi) + \eta I)^{-1}b = b^T(LL^T)^{-1}b = (L^{-1}b)^T(L^{-1}b) = \|L^{-1}b\|_2^2$. Hence, the loss function is defined as $\mathcal{J}(z^{(1)}, z^{(3)}, z^{(5)}) = \|L^{-1}b\|^2$. Optimization is carried out via a Newton method in 20 iterations. We initialize $z^{(1)}, z^{(3)}, z^{(5)} \in \mathbb{R}^{M_{\text{in}}}$ following $N(0, 10^{-6}I_{M_{\text{in}}})$. In each iteration, the gradient $\nabla\mathcal{J}$ and Hessian $\nabla^2\mathcal{J}$ are computed via automatic differentiation, and the Newton direction $\Delta z$ is obtained by solving $(\nabla^2\mathcal{J} + \lambda I)\Delta z = -\nabla\mathcal{J}$, where $\lambda = 10^{-4}$ is an regularization parameter. Then, update $\mathcal{J}$ at Newton direction with step size $\alpha = 1$. Early stopping is triggered when the gradient norm falls below $10^{-5}$. Finally, to apply the represener theorem in 22, the algorithm solves the linear system $(K(\phi, \phi) + \eta I)w^\dagger = z^\dagger$ to obtain the weight vector $w^\dagger$ and the final PDE solution is given as $u^\dagger(\mathbf{x}) = K(\mathbf{x}, \phi)w^\dagger$.

## B.2 QUADRATURE MULTILEVEL PICARD ITERATIONS AND FULL-HISTORY MULTILEVEL PICARD ITERATIONS

Multilevel Picard Iteration (MLP) method (Hutzenthaler et al., 2019) is a simulation-based solver which solves a semilinear parabolic PDEs (Hutzenthaler et al., 2019; Han et al., 2018a; Weinan et al., 2021), represented as the following.

$$
\begin{cases}
\dfrac{\partial}{\partial r}u^\infty + \langle \mu, \nabla_y u^\infty \rangle + \dfrac{1}{2}\mathrm{Tr}(\sigma^\top \mathrm{Hess}\, u^\infty\, \sigma) + F(u^\infty, \sigma^\top \nabla_y u^\infty) = 0, \text{ on } [0,T) \times \mathbb{R}^d \\
u^\infty(T, \boldsymbol{y}) = g(\boldsymbol{y}), \text{ on } \mathbb{R}^d.
\end{cases}
\tag{24}
$$

where $T > 0, d \in \mathbb{N}, g : \mathbb{R}^d \to \mathbb{R}, u^\infty : [0,T] \times \mathbb{R}^{d+1} \to \mathbb{R}, \mu : [0,T] \times \mathbb{R}^d \to \mathbb{R}^d$. Additionally, let $\sigma$ be a regular function mapping $[0,T] \times \mathbb{R}^d$ to a real $d \times d$ invertible matrix.

The MLP method reformulates the PDE into a fixed-point problem using the Feynman–Kac formula to represent the solution as the expected value of a stochastic process's functional. A Picard scheme iteratively solves this fixed-point problem. The MLP method employs a multilevel Monte Carlo approach(Giles, 2008), blending coarse and fine discretizations and allocating more samples to deeper iterations to control variance. This strategy ensures computational costs increase moderately with accuracy. According to Feynman–Kac and Bismut-Elworthy-Li formula(Elworthy & Li, 1994; Da Prato & Zabczyk, 1997), the solution $\mathbf{u}^\infty = (u, \sigma^\top \nabla_y u)$ of semilinear parabolic PDE (24) satisfies the fixed-point equation $\Phi(\mathbf{u}^\infty) = \mathbf{u}^\infty$ where $\Phi : \mathrm{Lip}([0,T] \times \mathbb{R}^d, \mathbb{R}^{1+d}) \to \mathrm{Lip}([0,T] \times \mathbb{R}^d, \mathbb{R}^{1+d})$ is defined as

$$
\begin{aligned}
(\Phi(\mathbf{v}))(s,x) =& \mathbb{E}\left[ g(X_T^{s,x})\left( 1, \frac{[\sigma(s,x)]^\top}{T-s}\int_s^T \left[\sigma(r, X_r^{s,x})^{-1}D_r^{s,x}\right]^\top dW_r \right) \right] \\
&+ \int_s^T \mathbb{E}\left[ F(\mathbf{v}(t, X_t^{s,x}))\left( 1, \frac{[\sigma(s,x)]^\top}{t-s}\int_s^t \left[\sigma(r, X_r^{s,x})^{-1}D_r^{s,x}\right]^\top dW_r \right) \right] dt.
\end{aligned}
\tag{25}
$$

Here $X_t^{s,x}$ and $D_t^{s,x}$ are defined as

$$
\begin{aligned}
X_t^{s,x} &= x + \int_s^t \mu(r, X_r^{s,x})dr + \sum_{j=1}^d \int_s^t \sigma_j(r, X_r^{s,x})dW_r^j, \\
D_t^{s,x} &= \mathrm{I}_{\mathbb{R}^{d\times d}} + \int_s^t (\frac{\partial}{\partial x}\mu)(r, X_r^{s,x})D_r^{s,x}dr + \sum_{j=1}^d \int_s^t (\frac{\partial}{\partial x}\sigma_j)(r, X_r^{s,x})D_r^{s,x}dW_r^j.
\end{aligned}
\tag{26}
$$

where $W_t : [0,T] \times \Omega \to \mathbb{R}^d$ is a standard $(\mathbb{F}_t)_{t\in[0,T]}$-adapted Brownian motion.

The Feynman-Kac formula gives

$$
u^\infty(s,x) = \mathbb{E}[g(X_T^{s,x})] + \int_s^T \mathbb{E}[F(u^\infty(t, X_t^{s,x}), [\sigma(t, X_t^{s,x})]^\top(\nabla_y u^\infty)(t, X_t^{s,x}))]dt.
\tag{27}
$$

Note that $\sigma^\top \nabla_y u^\infty$ appeared on the right-hand side in the fixed point iteration, which necessitates a new representation formula of it to be simultaneous with 27. And that is Bismut-Elworthy-Li formula(Elworthy & Li, 1994; Da Prato & Zabczyk, 1997), which gives

$$
\begin{aligned}
[\sigma(s,x)]^\top(\nabla_y u^\infty)(s,x) =& \mathbb{E}\left[ g(X_T^{s,x})\frac{[\sigma(s,x)]^\top}{T-s}\int_s^T \left[\sigma(r, X_r^{s,x})^{-1}D_r^{s,x}\right]^\top dW_r \right] \\
&+ \int_s^T \mathbb{E}\left[ F(u^\infty(t, X_t^{s,x}), [\sigma(t, X_t^{s,x})]^\top(\nabla_y u^\infty)(t, X_t^{s,x})) \right. \\
&\left. \frac{[\sigma(s,x)]^\top}{t-s}\int_s^t \left[\sigma(r, X_r^{s,x})^{-1}D_r^{s,x}\right]^T dW_r \right] dt,
\end{aligned}
\tag{28}
$$

Concatenating the solution as $\mathbf{u}^\infty = (u, \sigma^\top \nabla_y u)$, we can define the iteration operator $\Phi\text{:Lip}([0,T] \times \mathbb{R}^d, \mathbb{R}^{1+d}) \to \text{Lip}([0,T] \times \mathbb{R}^d, \mathbb{R}^{1+d})$ as the following

$$
\begin{aligned}
(\Phi(\mathbf{v}))(s,x) =& \mathbb{E}\left[g(X_T^{s,x})\left(1, \frac{[\sigma(s,x)]^\top}{T-s}\int_s^T \left[\sigma(r, X_r^{s,x})^{-1} D_r^{s,x}\right]^\top dW_r\right)\right] \\
&+ \int_s^T \mathbb{E}\left[F(\mathbf{v}(t, X_t^{s,x}))\left(1, \frac{[\sigma(s,x)]^\top}{t-s}\int_s^t \left[\sigma(r, X_r^{s,x})^{-1} D_r^{s,x}\right]^\top dW_r\right)\right] dt,
\end{aligned}
\tag{29}
$$

and 27, 28 yield

$$
\mathbf{u}^\infty = \Phi(\mathbf{u}^\infty). \tag{30}
$$

The Multilevel Picard iteration considers simulating the Picard iteartion $\mathbf{u}_k(s,x) = (\Phi(\mathbf{u}_{k-1}))(s,x), k \in \mathbb{N}_+$, which is guaranteed to converge to $\mathbf{u}^\infty$ as $k \to \infty$ for any $s \in [0,T), x \in \mathbb{R}^d$ (Yong & Zhou, 1999, Page 360, Theorem 3.4). Formally, the MLP method uses MLMC(Giles, 2008; 2015) to simulate the following telescope expansion problem derived from the Picard iteration.

$$
\begin{aligned}
\mathbf{u}_k(s,x) =& \mathbf{u}_1(s,x) + \sum_{l=1}^{k-1}\left[\mathbf{u}_{l+1}(s,x) - \mathbf{u}_l(s,x)\right] = \Phi(\mathbf{u}_1)(s,x) + \sum_{l=1}^{k-1}\left[\Phi(\mathbf{u}_l)(s,x) - \Phi(\mathbf{u}_{l-1})(s,x)\right]. \\
=& (g(\boldsymbol{x}), \mathbf{0}_d) + \mathbb{E}\left[(g(X_T^{s,x}) - g(\boldsymbol{x}))\left(1, \frac{[\sigma(s,x)]^\top}{T-s}\int_s^T \left[\sigma(r, X_r^{s,x})^{-1} D_r^{s,x}\right]^\top dW_r\right)\right] \\
&+ \sum_{l=0}^{k-1}\int_s^T \mathbb{E}\Big[(F(\mathbf{u}_l(t, X_t^{s,x})) - \mathbf{1}_\mathbb{N}(l)F(\mathbf{u}_{l-1}(t, X_t^{s,x}))) \\
&\left(1, \frac{[\sigma(s,x)]^\top}{t-s}\int_s^t \left[\sigma(r, X_r^{s,x})^{-1} D_r^{s,x}\right]^\top dW_r\right)\Big] dt.
\end{aligned}
\tag{31}
$$

One can either estimate these integrations with the quadrature method(quadrature MLP (E et al., 2021)) or the Monte-Carlo method(full-history MLP (Hutzenthaler et al., 2020b)), detailed intruction is shown in demonstrated in B.2. A comprehensive summary of MLP variants can be found at (Research Group on Stochastic Analysis, University of Duisburg-Essen, 2025).

### B.2.1 IMPLEMENTING MULTILEVEL PICARD ITERATIONS

Suppose we are given effective simulators (e.g., Euler–Maruyama or Milstein) parameterized by $\varphi$ (e.g. discretization level), which produce the numerical approximations

$$
\mathcal{X}_{k,\varphi}^{(l,i)}(s,x,t) \approx X_t^{s,x}, \quad \mathcal{I}_{k,\varphi}^{(l,i)}(s,x,t) \approx \left(1, \frac{[\sigma(s,x)]^\top}{t-s}\int_s^t \left[\sigma(r, X_r^{s,x})^{-1} D_r^{s,x}\right]^\top dW_r\right), \tag{32}
$$

where $k$ denotes the total level, $l$ the current level, and $i$ (which may be negative) indexes the sample path. To implement the Multilevel Picard Iterations, we need a numerical approximation to the integral $\int_s^T \mathbb{E}F(\mathbf{u}_l(t, X_t^{s,x}))dt$. Following (E et al., 2021; Hutzenthaler et al., 2021), we examine the following two methodologies, using quadrature rule and Monte Carlo algorithm to approximate the integral $\int_s^T \mathbb{E}F(\mathbf{u}_l(t, X_t^{s,x}))dt$:

**Quadrature MLP** In this approach (E et al., 2021), quadrature rules are employed to approximate the time integrals that appear in the MLP formulation. This quadrature-based technique is motivated by the need to efficiently and accurately resolve time integration errors while maintaining the stability of the multilevel scheme. By leveraging well-established Gauss–Legendre quadrature, we obtain a deterministic and high-order accurate approximation that is well-suited to the recursive structure of the SCaSML algorithm.

**Definition B.1** (Gauss–Legendre quadrature). *For each $n \in \mathbb{N}$, let $(c_i^n)_{i=1}^n \subseteq [-1, 1]$ denote the $n$ distinct roots of the Legendre polynomial $x \mapsto \frac{1}{2^n n!} \frac{d^n}{dx^n}[(x^2 - 1)^n]$, and define the function $q^{n,[a,b]} : [a, b] \to \mathbb{R}$ by*

$$
q^{n,[a,b]}(t) = \begin{cases} \displaystyle\int_a^b \prod_{\substack{i=1,\ldots,n \\ c_i^n \neq \frac{2t-(a+b)}{b-a}}} \frac{2x - (b-a)c_i^n - (a+b)}{2t - (b-a)c_i^n - (a+b)} \, dx, & \text{if } a < b \text{ and } \frac{2t-(a+b)}{b-a} \in \{c_1^n, \ldots, c_n^n\}, \\ 0, & \text{otherwise.} \end{cases}
$$

$$(33)$$

The Gauss–Legendre quadrature serve as a fundamental building block to discretize the time variable in the Picard iteration. With these polynomials, one can approximate the time integrals with high-order accuracy while controlling the error propagation in the recursive iterations.

**Definition B.2** (Quadrature Multilevel Picard Iteration). *Let $\left\{ \mathbf{U}_{n,M,Q}^{(l,j)} \right\}_{l,j \in \mathbb{Z}} \subseteq \mathcal{M}\Big(\mathcal{B}([0,T] \times \mathbb{R}^d) \otimes \mathcal{F}, \mathcal{B}(\mathbb{R} \times \mathbb{R}^d)\Big)$ be a family of measurable functions satisfying, for all $l, j \in \mathbb{N}$ and $(s, x) \in [0, T] \times \mathbb{R}^d$, we start with $\mathbf{U}_{n,M,Q}^{(0,\pm j)}(s, x) = \mathbf{0}_{d+1}$. For $n > 0$, we define the quadrature SCaSML iteration as*

$$
\mathbf{U}_{n,M,Q}(s, x) = \Big(g(\boldsymbol{x}), \mathbf{0}_d\Big) + \frac{1}{M^n} \sum_{i=1}^{M^n} \Big(g\big(\mathcal{X}_{k,\varphi}^{(0,-i)}(s, x, T)\big) - g(\boldsymbol{x})\Big)\mathcal{I}_{k,\varphi}^{(0,-i)}(s, x, T)
$$

$$
+ \sum_{l=0}^{n-1} \sum_{t \in (s,T)} \frac{q^{Q,[s,T]}(t)}{M^{n-l}} \sum_{i=1}^{M^{n-l}} \Big( F\big(\mathbf{U}_{n,M,Q}^{(l,i)}(t, \mathcal{X}_{k-l,\varphi}^{(l,i)}(s, x, t))\big) - \mathbf{1}_{\mathbb{N}}(l)\, F\big(\mathbf{U}_{n,M,Q}^{(l-1,-i)}(t, \mathcal{X}_{k-l,\varphi}^{(l,i)}(s, x, t))\big)\Big)
$$

$$
\cdot \mathcal{I}_{k-l,\varphi}^{(l,i)}(s, x, t).
$$

$$(34)$$

The use of quadrature in this context is motivated by its ability to yield a systematic error control over the temporal discretization, thereby enhancing the stability and accuracy of the multilevel Picard iteration in the simulation-calibrated framework.

**Full-history MLP** The full-history MLP scheme (Hutzenthaler et al., 2021) adopts a Monte Carlo approach to approximate the time integral $\int_s^T \mathbb{E}F(\mathbf{u}_l(t, X_t^{s,x}))dt$ instead of deterministic quadrature rules with fixed time grids. This modification considerably simplifies error analysis(Hutzenthaler et al., 2020a) and avoids all temporal discretization error.

In the full-history MLP, we employ a time-sampler that guarantees an unbiased Monte Carlo approximation of time integrals. Let $\mathfrak{r} : \Omega \to (0, 1)$ be a collection of independent and identically distributed random variables with density $\rho$ satisfying $\mathbb{P}\Big(\mathfrak{r}^{(l,i)} \leq b\Big) = \int_0^b \rho(s)\, ds$. Consider numerically approximating the integral $I(f; s, t) = \int_s^t f(r)\, dr$ with $t \in (s, T)$, we construct an importance sampling estimator with sample size $N$:$\hat{I}(f; s, t) = \frac{1}{N} \sum_{i=1}^N \frac{f(R^{(i)}) \mathbf{1}_{\{R^{(i)} \leq t\}}}{\varrho(R^{(i)}, s)}$, where $\varrho$ is the the rescaled density $\rho$ on $(s, T)$ defined as $\varrho(r, s) = \frac{\rho\left(\frac{r-s}{T-s}\right)}{T-s}$ and $R$ is the random sample from the density $\varrho(\cdot, s)$ on $(s, T)$ via $R = s + (T - s)\mathfrak{r}$.

**Definition B.3** (Full-history Multilevel Picard Iteration (Hutzenthaler et al., 2020a)). *Let $\left\{ \mathbf{U}_{n,M}^{(l,j)} \right\}_{l,j \in \mathbb{Z}} \subseteq \mathcal{M}\Big(\mathcal{B}([0,T] \times \mathbb{R}^d) \otimes \mathcal{F}, \mathcal{B}(\mathbb{R} \times \mathbb{R}^d)\Big)$ be a family of measurable functions satisfying, for all $l, j \in \mathbb{N}$ and $(s, x) \in [0, T] \times \mathbb{R}^d$, we start with $\mathbf{U}_{n,M}^{(0,\pm j)}(s, x) = \mathbf{0}_{d+1}$. Then, for $n > 0$,*

*define the full-history SCaSML iteration as*

$$
\mathbf{U}_{n,M}(s,x) = \Big(g(\boldsymbol{x}), \mathbf{0}_d\Big) + \frac{1}{M^n} \sum_{i=1}^{M^n} \Big(g\big(\mathcal{X}_{k,\varphi}^{(0,-i)}(s,x,T)\big) - g(\boldsymbol{x})\Big)\mathcal{I}_{k,\varphi}^{(0,-i)}(s,x,T)
$$

$$
+ \sum_{l=0}^{n-1} \frac{1}{M^{n-l}} \sum_{i=1}^{M^{n-l}} \frac{1}{\varrho(s,\mathcal{R}_s^{(l,i)})} \Big( F\big(\mathbf{U}_{n,M}^{(l,i)}(\mathcal{R}_s^{(l,i)}, \mathcal{X}_{k-l,\varphi}^{(l,i)}(s,x,\mathcal{R}_s^{(l,i)}))\big)
$$

$$
- \mathbf{1}_{\mathbb{N}}(l) F\big(\mathbf{U}_{n,M}^{(l-1,-i)}(\mathcal{R}_s^{(l,i)}, \mathcal{X}_{k-l,\varphi}^{(l,i)}(s,x,\mathcal{R}_s^{(l,i)}))\big)\Big) \cdot \mathcal{I}_{k-l,\varphi}^{(l,i)}\Big(s,x,\mathcal{R}_s^{(l,i)}\Big),
$$

(35)

*here $\mathcal{R}_s^{(l,i)}$ is i-th sampled time point after t at level l which is defined as as $\mathcal{R}_s^{(l,i)} = s + (T-s)\,\mathfrak{r}^{(l,i)}$.*

## C  ALGORITHM

In this section, we describe the complete procedure of Simulation-Calibrated Scientific Machine Learning (SCaSML) for solving high-dimensional partial differential equations (1). The SCaSML framework at any space-time point $(t, \boldsymbol{x})$ can be summarized as follows:

- **Step 1:Train a Base Surrogate.** First, a surrogate model $\hat{u}$ is trained to approximately solve the target PDE (1), serving as a preliminary estimate of the true solution.

- **Step 2:Physics-Informed Inference-Time Scaling via the Structural-preserving Law of Defect.** Recognizing that the defect $\breve{u} := u - \hat{u}$ satisfies a semi-linear parabolic equation, termed the *Structural-preserving Law of Defect*,

$$
\begin{cases}
\frac{\partial}{\partial r}\breve{u} + \langle \mu, \nabla_y \breve{u}\rangle + \frac{1}{2}\operatorname{Tr}\big(\sigma^\top \operatorname{Hess}_y \breve{u}\,\sigma\big) + \breve{F}\big(\breve{u}, \sigma^\top \nabla_y \breve{u}\big) = 0, & \text{on } [0,T) \times \mathbb{R}^d, \\
\breve{u}(T, \boldsymbol{y}) = \breve{g}(y), & \text{on } \mathbb{R}^d,
\end{cases}
$$

(36)

one obtains an estimate of $\breve{u}(t, \boldsymbol{x})$ by employing Multilevel Picard iteration, either through quadrature-based MLP (Definition B.2) or full-history MLP (Definition B.3).

- **Step 3:Final Estimation.** The final estimate of the solution is then given by $u(t, \boldsymbol{x}) \approx \hat{u}(t, \boldsymbol{x}) + \breve{u}(t, \boldsymbol{x})$.

The entire algorithm is detailed in Algorithm 1.

We emphasize that the sample-wise iteration in Algorithm 1 can be substituted by vectorized operations, thereby enabling the algorithm to be applied concurrently to multiple points. These performance enhancements were implemented using `JAX` and `DeepXDE`, resulting in a time reduction by a factor of $5\times$ to $10\times$.

Additionally, methods such as thresholding (Sebastian Becker et al., 2020) and Hutchinson's estimator (Hutchinson, 1989; Shi et al., 2025) could also be employed within the principal algorithm. Thresholding (Algorithm 2) mitigates numerical instability by methodically "clipping" the defect estimator $\breve{\mathbf{U}}$, a critical action when the surrogate model yields outlier values or when unbounded growth may manifest during iterative correction phases. Hutchinson's estimator (Algorithm 3) alleviates the computational and memory demands of $\epsilon_{PDE}$ in $\breve{F}$ by forming an unbiased estimator that necessitates only a subset of second-order derivatives approximating the Laplacian. This partial evaluation not only expedites the simulation process but also minimizes peak memory consumption, thus averting out-of-memory issues.

## D  PROOF SETTINGS

In the following sections, we establish the rigorous mathematical framework for analyzing the SCaSML method. We proceed in three steps:

1. **Notations and Definitions:** We define the probability spaces, norms, and function spaces used throughout the proofs.

---

**Algorithm 1** Simulation-Calibrated Scientific Machine Learning for Solving High-Dimensional Partial Differential Equation

---

**Require:** Level $n$, sample base $M$, target point $(s, x)$, a surrogate model $\hat{u}$, threshold $\varepsilon$, (quadrature order $Q$ for using Qudrature MLP)

1: Train a base surrogate model $\hat{u}$ to approximate the PDE solution.
2: Take MLP_Law_of_Defect$(s, x, n, M, Q) \cdot (1, \mathbf{0}_d) + \hat{u}(s, x)$ as estimation of $u(s, x)$
3: **function** MLP_LAW_OF_DEFECT$(s, x, n, M, Q)$
4:      $\hat{\mathbf{u}}(s, x) \leftarrow \left( \hat{u}(s, x), \; \sigma^\top(s, x) \nabla_y \hat{u}(s, x) \right)$
5:      **if** $n = 0$ **then**               ▷ Start Inference-Time Scaling via Simulating the Structural-preserving Law of Defect
6:          $\breve{\mathbf{U}}_{n,M,Q}(s, x) \leftarrow \mathbf{0}_{d+1}$
7:          **return** $\breve{\mathbf{U}}_{n,M,Q}(s, x)$
8:      **end if**
9:      $\breve{\mathbf{U}}_{n,M,Q}(s, x) \leftarrow (\breve{g}(\boldsymbol{x}), \mathbf{0}_d)$
10:      **for** $i = 1$ to $M^n$ **do**
11:          Sample Feyman-Kac Path $\mathcal{X}_{k,\varphi}^{(0,-i)}(s, x, T)$ and Derivative Process $\mathcal{I}_{k,\varphi}^{(0,-i)}(s, x, T)$ in (32)
12:          $\breve{\mathbf{U}}_{n,M,Q}(s, x) \leftarrow \breve{\mathbf{U}}_{n,M,Q}(s, x) + \frac{1}{M^n} \left( \breve{g}\big(\mathcal{X}_{k,\varphi}^{(0,-i)}(s, x, T)\big) - \breve{g}(\boldsymbol{x}) \right) \cdot \mathcal{I}_{k,\varphi}^{(0,-i)}(s, x, T)$
13:      **end for**
14:      **for** $l = 0$ to $n - 1$ **do**
15:          **for** $i = 1$ to $M^{n-l}$ **do**
16:              **if** using `Quadrature MLP` to calibrate **then**
17:                  Compute $Q$ quadrature points with corresponding weights $q^{Q,[s,T]}(t)$ by B.1
18:                  **for** all quadrature points $t \in [s, T]$ **do**
19:                      Sample Feyman-Kac Path $\mathcal{X}_{k,\varphi}^{(l,i)}(s, x, t)$ and Derivative Process $\mathcal{I}_{k,\varphi}^{(l,i)}(s, x, t)$ according to formula (32)
20:                      $\mathbf{z} \leftarrow$ MLP_Law_of_Defect$(t, \mathcal{X}_{k-l,\varphi}^{(l,i)}(s, x, t), l, M, Q)$
21:                      **if** $l > 0$ **then**
22:                          $\mathbf{z}_{\text{prev}} \leftarrow$ MLP_Law_of_Defect$(t, \mathcal{X}_{k-l,\varphi}^{(l,i)}(s, x, t), l - 1, M, Q)$
23:                          $\Delta\breve{F} \leftarrow \breve{F}(\mathbf{z}) - \breve{F}(\mathbf{z}_{\text{prev}})$
24:                      **else**
25:                          $\Delta\breve{F} \leftarrow \breve{F}(\mathbf{z})$
26:                      **end if**
27:                      $\breve{\mathbf{U}}_{n,M,Q}(s, x) \leftarrow \breve{\mathbf{U}}_{n,M,Q}(s, x) + \frac{q^{Q,[s,T]}(t)}{M^{n-l}} \Delta\breve{F} \cdot \mathcal{I}_{k-l,\varphi}^{(l,i)}(s, x, t)$
28:                  **end for**
29:              **end if**
30:              **if** using `Full History MLP` to calibrate **then**
31:                  Sample time step $\mathcal{R}_s^{(l,i)} \sim \varrho(s, T)$
32:                  Sample Feyman-Kac Path $\mathcal{X}_{k,\varphi}^{(l,i)}(s, x, \mathcal{R}_s^{(l,i)})$ and Derivative Process $\mathcal{I}_{k,\varphi}^{(l,i)}(s, x, \mathcal{R}_s^{(l,i)})$ according to formula (32)
33:                  $\mathbf{z} \leftarrow$ MLP_Law_of_Defect$(\mathcal{R}_s^{(l,i)}, \mathcal{X}_{k-l,\varphi}^{(l,i)}(s, x, \mathcal{R}_s^{(l,i)}), l, M, Q)$
34:                  **if** $l > 0$ **then**
35:                      $\mathbf{z}_{\text{prev}} \leftarrow$ MLP_Law_of_Defect$(\mathcal{R}_s^{(l,i)}, \mathcal{X}_{k-l,\varphi}^{(l,i)}(s, x, \mathcal{R}_s^{(l,i)}), l - 1, M, Q)$
36:                      $\Delta\breve{F} \leftarrow \breve{F}(\mathbf{z}) - \breve{F}(\mathbf{z}_{\text{prev}})$
37:                  **else**
38:                      $\Delta\breve{F} \leftarrow \breve{F}(\mathbf{z})$
39:                  **end if**
40:                  $\breve{\mathbf{U}}_{n,M,Q}(s, x) \leftarrow \breve{\mathbf{U}}_{n,M,Q}(s, x) + \frac{1}{M^{n-l}} \cdot \frac{1}{\varrho(s, \mathcal{R}_s^{(l,i)})} \cdot \Delta\breve{F} \cdot \mathcal{I}_{k-l,\varphi}^{(l,i)}(s, x, \mathcal{R}_s^{(l,i)})$
41:              **end if**
42:          **end for**
43:      **end for**
44:      $\breve{\mathbf{U}}_{n,M,Q}(s, x) \leftarrow$ Thresholding$(\varepsilon, \breve{\mathbf{U}}_{n,M,Q}(s, x))$      ▷ Threshold outliers using Algorithm 2
45:      **return** $\breve{\mathbf{U}}_{n,M,Q}(s, x)$
46: **end function**

---

---

**Algorithm 2** Thresholding the outliers (Sebastian Becker et al., 2020)

---

**Require:** Threshold $\varepsilon$, defect estimator $\check{\mathbf{U}}$
1: **function** THRESHOLDING($\varepsilon$, $\check{\mathbf{U}}$)
2:     **for** $\varsigma = 1$ to $d + 1$ **do**
3:         **if** $\check{\mathbf{U}}_\varsigma > \varepsilon$ **then**
4:             $\check{\mathbf{U}}_\varsigma \leftarrow \varepsilon$
5:         **end if**
6:         **if** $\check{\mathbf{U}}_\varsigma < -\varepsilon$ **then**
7:             $\check{\mathbf{U}}_\varsigma \leftarrow -\varepsilon$
8:         **end if**
9:     **end for**
10:     **return** Clipped $\check{\mathbf{U}}$
11: **end function**

---

**Algorithm 3** Hutchison's estimator for estimating Laplacian (Shi et al., 2025)

---

**Require:** Sample size $K$, target function $f$
1: **function** HTE($K$,$f$)
2:     Draw $K$ different indices from $1, \ldots, d$ with equal probability $1/d$, denoted as $j_1, \ldots, j_K$
3:     Compute $D_{j_k}^2 f, 1 \le \varsigma \le d$
4:     Compute estimator HTE $\leftarrow \frac{d}{K} \sum_{i=1}^{K} D_{j_i}^2 f$
5:     **return** Laplacian estimator HTE
6: **end function**

---

2. **Problem Setup:** We explicitly state the regularity assumptions on the original PDE coefficients and the stochastic basis.

3. **Surrogate and Defect Properties:** We formally define the surrogate model, the defect PDE, and the transfer of Lipschitz properties from the original problem to the defect problem.

## D.1   Mathematical Framework and Definitions

In this section, we rigorously define the measure-theoretic structures, function spaces, and norms required for the convergence analysis. Our framework aligns with the standard stochastic analysis settings found in (Hutzenthaler et al., 2020a; E et al., 2021).

**Definition D.1** (Coordinate System and Vector Norms). *Throughout this article, we fix a time horizon $T \in (0, \infty)$ and a spatial dimension $d \in \mathbb{N}$. We denote the time-space domain by $\Lambda := [0, T] \times \mathbb{R}^d$. We consistently use the coordinate notation $(t, \boldsymbol{x})$ with $t \in [0, T]$ and $x \in \mathbb{R}^d$.*

*For any vector $v = (v_1, \ldots, v_d) \in \mathbb{R}^d$, we denote the standard Euclidean norm by $|v| := (\sum_{i=1}^{d} |v_i|^2)^{1/2}$ and the inner product by $v \cdot w$. For a generic vector $z \in \mathbb{R}^n$ (e.g., neural network parameters), we define the discrete $p$-norm ($p \in [1, \infty)$) and $\infty$-norm as:*

$$\|z\|_p := \left( \sum_{i=1}^{n} |z_i|^p \right)^{1/p}, \quad \text{and} \quad \|z\|_\infty := \max_{1 \le i \le n} |z_i|.$$

**Definition D.2** (Measurable Spaces and Functions). *We denote by $\mathcal{B}(\mathbb{R}^d)$ the Borel $\sigma$-algebra on $\mathbb{R}^d$. For any two measurable spaces $(S_1, \mathcal{F}_1)$ and $(S_2, \mathcal{F}_2)$, we define $\mathcal{M}(S_1, S_2)$ as the set of all measurable mappings from $S_1$ to $S_2$:*

$$\mathcal{M}(S_1, S_2) := \{f : S_1 \to S_2 \mid \forall A \in \mathcal{F}_2, f^{-1}(A) \in \mathcal{F}_1\}.$$

*When the $\sigma$-algebras are clear from context (e.g., Borel for topological spaces), we simply write $\mathcal{M}(\mathbb{R}^d, \mathbb{R})$.*

**Definition D.3** (Probability Space and $L^p$ Norms). *Let $(\Omega, \mathcal{F}, \mathbb{P})$ be a complete probability space. For any measurable random variable $X \in \mathcal{M}(\Omega, \mathbb{R})$ and $p \in [1, \infty)$, the $L^p(\Omega)$-norm is defined as:*

$$\|X\|_{L^p(\Omega)} := (\mathbb{E}[|X|^p])^{1/p} = \left( \int_\Omega |X(\omega)|^p \, d\mathbb{P}(\omega) \right)^{1/p}.$$

*For $p = \infty$, the essential supremum norm is defined as:*

$$\|X\|_{L^\infty(\Omega)} := \inf\{C \geq 0 : \mathbb{P}(|X| > C) = 0\}.$$

**Definition D.4** (Function Spaces). *Let $D \subseteq \mathbb{R}^d$ be an open set. For $k \in \mathbb{N}$ and $p \in [1, \infty]$, the Sobolev space $W^{k,p}(D)$ consists of all functions $u \in L^p(D)$ such that for every multi-index $\alpha \in \mathbb{N}_0^d$ with $|\alpha| \leq k$, the weak derivative $D^\alpha u$ exists and belongs to $L^p(D)$. We define the norm for $W^{k,\infty}(D)$ as $\|u\|_{W^{k,\infty}(D)} := \sum_{|\alpha| \leq k} \|D^\alpha u\|_{L^\infty(D)}$.*

*Furthermore, let $C^{1,2}([0,T] \times \mathbb{R}^d)$ denote the space of functions $\phi(t, \boldsymbol{x})$ that are once continuously differentiable in $t$ and twice continuously differentiable in $x$. This regularity is required for the classical solution $u$ and the surrogate $\hat{u}$.*

**Definition D.5** (Extended Real Arithmetic). *To handle singularities in complexity analysis, we adopt the standard conventions for the extended real number line $\overline{\mathbb{R}} = \mathbb{R} \cup \{-\infty, \infty\}$. Specifically, we define $\frac{0}{0} = 0$, $0 \cdot \infty = 0$, $0^0 = 1$, and $\sqrt{\infty} = \infty$. For any $a > 0$ and $b \in \mathbb{R}$, we set $\frac{a}{0} = \infty$, $\frac{-a}{0} = -\infty$, $0^{-a} = \infty$, $\frac{1}{0^a} = \infty$, $\frac{b}{\infty} = 0$, and $0^a = 0$.*

## D.2 PROBLEM SETUP AND REGULARITY ASSUMPTIONS

We now formalize the specific partial differential equation and the stochastic framework used for our theoretical analysis.

### D.2.1 STOCHASTIC BASIS

Let $T \in (0, \infty)$ be the terminal time and $d \in \mathbb{N}$ be the spatial dimension. Let $(\Omega, \mathcal{F}, \mathbb{P}, (\mathbb{F}_t)_{t \in [0,T]})$ be all stochastic processes are assumed to be adapted to the usual Filtration $(\mathbb{F}_t)_{t \in [0,T]}$.

To facilitate the Multilevel Picard (MLP) analysis, we assume the existence of a family of independent standard Brownian motions. Specifically, let $\{W^{(l,j)} : l, j \in \mathbb{Z}\}$ be a collection of independent $d$-dimensional standard Brownian motions adapted to $(\mathbb{F}_t)_{t \in [0,T]}$. Here, the index $l$ corresponds to the level in the MLP hierarchy, and $j$ corresponds to the Monte Carlo sample index within that level.

### D.2.2 THE TARGET PDE

While the SCaSML framework applies to general semi-linear parabolic PDEs, we perform the theoretical analysis on the semi-linear heat equation. This corresponds to the generator $\mathcal{L}$ with drift $\mu \equiv 0$ and diffusion $\sigma \equiv s\mathbf{I}_d$ for a constant $s \in \mathbb{R} \setminus \{0\}$.

The classical solution $u \in C^{1,2}([0,T] \times \mathbb{R}^d, \mathbb{R})$ satisfies the terminal value problem:

$$\frac{\partial u}{\partial t}(t, \boldsymbol{x}) + \mathcal{L}u(t, \boldsymbol{x}) + F(u(t, \boldsymbol{x}), \sigma^\top \nabla_{\boldsymbol{x}} u(t, \boldsymbol{x})) = 0, \quad (t, \boldsymbol{x}) \in [0, T) \times \mathbb{R}^d, \quad (37)$$

where $\mathcal{L}v := \frac{\sigma^2}{2}\Delta v$, and subject to the terminal condition $u(T, \boldsymbol{x}) = g(\boldsymbol{x})$.

The nonlinearity $F : \mathbb{R} \times \mathbb{R}^d \to \mathbb{R}$ and terminal condition $g : \mathbb{R}^d \to \mathbb{R}$ are assumed to be Borel measurable functions.

### D.2.3 REGULARITY ASSUMPTIONS

The MLP method achieves dimension-independent convergence rates under Lipschitz continuity conditions on the problem data. These conditions ensure bounded variance propagation across Picard iterations (Hutzenthaler et al., 2021).

**Assumption D.6** (Lipschitz Continuity of Nonlinearity and Terminal Condition). *We assume the following:*

1. ***Nonlinearity:*** *There exists a constant $L \geq 0$ such that for all $(v_1, \boldsymbol{z}_1), (v_2, \boldsymbol{z}_2) \in \mathbb{R} \times \mathbb{R}^d$ and $(t, \boldsymbol{x}) \in [0, T] \times \mathbb{R}^d$:*

$$|F(v_1, \boldsymbol{z}_1, t, \boldsymbol{x}) - F(v_2, \boldsymbol{z}_2, t, \boldsymbol{x})| \leq L(|v_1 - v_2| + \|\boldsymbol{z}_1 - \boldsymbol{z}_2\|_1). \quad (38)$$

2. ***Terminal Condition:*** *There exists a constant $K \geq 0$ such that for all $\boldsymbol{x}, \boldsymbol{y} \in \mathbb{R}^d$:*

$$|g(\boldsymbol{x}) - g(\boldsymbol{y})| \leq K\|\boldsymbol{x} - \boldsymbol{y}\|_1. \quad (39)$$

### D.3 SURROGATE MODEL AND DEFECT PROPERTIES

In this section, we rigorously define the relationship between the pre-trained surrogate model and the defect (error) we aim to estimate. We first state the regularity assumptions on the surrogate, then derive the properties of the Defect PDE.

#### D.3.1 SURROGATE REGULARITY

To ensure the classical defect PDE is well-defined, we assume the surrogate is sufficiently smooth. Let $\hat{u} \in C^{1,2}([0,T] \times \mathbb{R}^d, \mathbb{R})$ be a deterministic approximation of $u$. To ensure the defect terminal condition is well-behaved, we require the following:

**Assumption D.7** (Lipschitz Continuity of the Surrogate Terminal). *There exists a constant $\hat{K} \in [0, \infty)$ such that for all $\boldsymbol{x}, \boldsymbol{y} \in \mathbb{R}^d$:*

$$|\hat{u}(T, \boldsymbol{x}) - \hat{u}(T, \boldsymbol{y})| \le \hat{K} \|\boldsymbol{x} - \boldsymbol{y}\|_1. \tag{40}$$

#### D.3.2 THE STRUCTURAL-PRESERVING LAW OF DEFECT

We define the **defect** $\breve{u} : [0,T] \times \mathbb{R}^d \to \mathbb{R}$ as the pointwise error:

$$\breve{u}(t, \boldsymbol{x}) := u(t, \boldsymbol{x}) - \hat{u}(t, \boldsymbol{x}). \tag{41}$$

The core of SCaSML is the observation that $\breve{u}$ satisfies a semi-linear PDE of the same structure as the original. We explicitly define the coefficients of this new PDE below.

**Definition D.8** (Modified Nonlinearity and PDE Residual). *Let $\epsilon : [0,T] \times \mathbb{R}^d \to \mathbb{R}$ be the **PDE residual** of the surrogate $\hat{u}$ defined by:*

$$\epsilon(t, \boldsymbol{x}) := \frac{\partial \hat{u}}{\partial t}(t, \boldsymbol{x}) + \mathcal{L}\hat{u}(t, \boldsymbol{x}) + F(\hat{u}(t, \boldsymbol{x}), \sigma^\top \nabla_{\boldsymbol{x}}\hat{u}(t, \boldsymbol{x})).$$

*We define the modified nonlinearity $\breve{F} : \mathbb{R} \times \mathbb{R}^d \times [0,T] \times \mathbb{R}^d \to \mathbb{R}$ for the defect PDE as follows. For any state $v \in \mathbb{R}$ and gradient-state $\boldsymbol{z} \in \mathbb{R}^d$ at a spacetime point $(t, \boldsymbol{x})$:*

$$\breve{F}(v, \boldsymbol{z}, t, \boldsymbol{x}) := F(\hat{u}(t, \boldsymbol{x}) + v, \sigma^\top \nabla_{\boldsymbol{x}}\hat{u}(t, \boldsymbol{x}) + \boldsymbol{z}) - F(\hat{u}(t, \boldsymbol{x}), \sigma^\top \nabla_{\boldsymbol{x}}\hat{u}(t, \boldsymbol{x})) + \epsilon(t, \boldsymbol{x}). \tag{42}$$

*We similarly define the defect terminal condition $\breve{g}(\boldsymbol{x}) := g(\boldsymbol{x}) - \hat{u}(T, \boldsymbol{x})$.*

**Lemma D.9** (Structural-Preserving Law of Defect). *The defect $\breve{u}$ is a classical solution to the following semi-linear parabolic PDE:*

$$\frac{\partial \breve{u}}{\partial t}(t, \boldsymbol{x}) + \mathcal{L}\breve{u}(t, \boldsymbol{x}) + \breve{F}(\breve{u}, \sigma^\top \nabla_{\boldsymbol{x}}\breve{u}, t, \boldsymbol{x}) = 0, \forall (t, \boldsymbol{x}) \in [0,T) \times \mathbb{R}^d \quad \breve{u}(T, \boldsymbol{x}) = \breve{g}(\boldsymbol{x}). \tag{43}$$

**Remark D.10** (Why Law of Defect is Easier to Solve). *The complexity of MLP depends on the magnitude of source term $\breve{F}$ (Hutzenthaler et al., 2020b, Theorem 3.1). Based on the Lipschitz continuity of $\breve{F}$ and the variance-reduction structure inherent to MLMC, Hutzenthaler et al. (2021) shows that the overall computational complexity of MLP is governed solely by the value of $\breve{F}$ at the origin. Substituting $v = 0$ and $\boldsymbol{z} = \boldsymbol{0}$ into Definition D.8: $\breve{F}(0, \boldsymbol{0}, t, \boldsymbol{x}) = \epsilon(t, \boldsymbol{x})$. The "source term" driving the Multilevel Picard simulation for the defect is the residual $\epsilon$, already reduced by an approximate surrogate. If the surrogate is perfect ($\epsilon \to 0$), the driving force vanishes, and the variance of the Monte Carlo estimator approaches zero. In our later theorem, we show that the variance of MLP can be controlled by the magnitude of $\epsilon$.*

#### D.3.3 REGULARITY ESTIMATIONS

MLP complexity is governed by both the smoothness and magnitude of the source term; these factors enter multiplicatively because nonlinearities—via their Lipschitz bounds—propagate and amplify variance through each Picard iteration. Remark D.10 established that the magnitude component in the law of defect can be improved using the surrogate. It remains to show that the regularity appearing in the law of defect is no worse than that of the original PDE, ensuring that the refinement does not introduce additional smoothness requirements.

**Lemma D.11** (Preservation of Lipschitz Constants). *Suppose $F$ satisfies Assumption D.6 with Lipschitz constants $L$. Then, the modified nonlinearity $\breve{F}$ satisfies the same Lipschitz condition with the same constants. Specifically, for any fixed $(t, \boldsymbol{x})$, and any vectors $(\breve{v}_1, \boldsymbol{z}_1), (\breve{v}_2, \boldsymbol{z}_2) \in \mathbb{R} \times \mathbb{R}^d$:*

$$|\breve{F}(\breve{v}_1, \boldsymbol{z}_1, t, \boldsymbol{x}) - \breve{F}(\breve{v}_2, \boldsymbol{z}_2, t, \boldsymbol{x})| \le L(|\breve{v}_1 - \breve{v}_2| + \|\boldsymbol{z}_1 - \boldsymbol{z}_2\|_1), \tag{44}$$

*Furthermore, the defect terminal condition $\breve{g}$ is Lipschitz continuous with constants $\breve{K} = K + \hat{K}$.*

*Proof.* Let $\boldsymbol{w}_1 = (\breve{v}_1, \boldsymbol{z}_1)$ and $\boldsymbol{w}_2 = (\breve{v}_2, \boldsymbol{z}_2)$. We define the background state vector of the surrogate as $\hat{\boldsymbol{U}} = (\hat{u}(t, \boldsymbol{x}), \sigma^\top \nabla_{\boldsymbol{x}} \hat{u}(t, \boldsymbol{x}))$. From Definition D.8, the difference is:

$$
\begin{aligned}
\breve{F}(\boldsymbol{w}_1, t, \boldsymbol{x}) - \breve{F}(\boldsymbol{w}_2, t, \boldsymbol{x}) &= \Big[F(\hat{\boldsymbol{U}} + \boldsymbol{w}_1) - F(\hat{\boldsymbol{U}}) + \epsilon\Big] - \Big[F(\hat{\boldsymbol{U}} + \boldsymbol{w}_2) - F(\hat{\boldsymbol{U}}) + \epsilon\Big] \\
&= F(\hat{\boldsymbol{U}} + \boldsymbol{w}_1) - F(\hat{\boldsymbol{U}} + \boldsymbol{w}_2).
\end{aligned}
$$

Note that the shift terms $F(\hat{\boldsymbol{U}})$ and the residual $\epsilon(t, \boldsymbol{x})$ cancel out exactly. Thus, the Lipschitz continuity of $F$ (Assumption D.6) transfers directly to $\breve{F}$:

$$|F(\hat{\boldsymbol{U}} + \boldsymbol{w}_1) - F(\hat{\boldsymbol{U}} + \boldsymbol{w}_2)| \le L\|(\hat{\boldsymbol{U}} + \boldsymbol{w}_1) - (\hat{\boldsymbol{U}} + \boldsymbol{w}_2)\|_1 = L\|\boldsymbol{w}_1 - \boldsymbol{w}_2\|_1.$$

This confirms that $\breve{F}$ inherits the Lipschitz constants $L$. For the terminal condition, since $\hat{u} \in C^{1,2}([0, T] \times \mathbb{R}^d)$ by Assumption D.7, the map $x \mapsto \hat{u}(T, \boldsymbol{x})$ is Lipschitz with constants $\hat{K}$. The triangle inequality applied to $\breve{g} = g - \hat{u}(T, \cdot)$ then yields $\breve{K} \le K + \hat{K}$. □

## E  PROOF OF FULL-HISTORY MULTILEVEL PICARD ITERATION

This section establishes the theoretical guarantees for SCaSML when the `Structural-preserving Law of Defect` is solved using the Full-History Multi-level Picard (MLP) iteration. In contrast to the quadrature method, this approach utilizes Monte Carlo sampling for time integration, which relaxes the regularity requirements on the solution.

For the theoretical analysis, we retain the setting of the semi-linear heat equation where $\mu = \boldsymbol{0}_d$ and $\sigma = s\mathbf{I}_d$ for a constant $s \in \mathbb{R}$.

### E.1  PROBABILISTIC SETUP AND TIME SAMPLING

To analyze the Full-History estimator, we must extend our stochastic basis to support random time stepping.

**Definition E.1** (Extended Probability Space and Time Sampling). *Let $(\Omega, \mathcal{F}, \mathbb{P})$ be a probability space that supports the following independent families of random variables:*

1.  ***Brownian Motions:*** *A collection $\{W^{(l,j)}\}_{l,j \in \mathbb{Z}}$ of independent $d$-dimensional standard Brownian motions.*

2.  ***Time Step Samples:*** *A collection $\{\mathfrak{r}^{(l,j)}\}_{l,j \in \mathbb{Z}}$ of independent random variables distributed on $(0, 1)$ according to a probability density function $\rho : (0, 1) \to (0, \infty)$.*

*For our analysis and experiments, we specifically select the density $\rho(s) = (1 - \alpha)s^{-\alpha}$ for a parameter $\alpha \in (0, 1)$. This ensures that the cumulative distribution function is $F_\rho(b) = b^{1-\alpha}$, facilitating efficient inverse transform sampling.*

### E.2  RELAXED SURROGATE ASSUMPTIONS

A key advantage of the Full-History MLP is its robustness. Unlike the quadrature scheme, which incurs a time discretization error scaling with high-order time derivatives of the solution, the Monte Carlo time integration is unbiased. Consequently, we can drop the higher-order regularity requirement (Assumption F.1, Item 3) imposed in Appendix F.

**Assumption E.2** (Accuracy of the Surrogate Model for Full-History MLP). *Let $\breve{u}$ be the solution to the Defect PDE. We assume $\sup_{t \in [0,T]} \|\breve{u}(t, \cdot)\|_{W^{1,\infty}(\mathbb{R}^d)} < \infty$. There exist constants $C_{F,1}, C_{F,2} > 0$ independent of $\hat{u}$ such that the surrogate error measure $e(\hat{u})$ controls the following:*

1. **Residual Bound ($L^\infty$):**

$$\sup_{(t,\boldsymbol{x})\in[0,T]\times\mathbb{R}^d} |\epsilon(t,\boldsymbol{x})| \leq C_{F,1}\, e(\hat{u}).$$

2. **Defect Bound ($W^{1,\infty}$):**

$$\sup_{t\in[0,T]} \|\breve{u}(t,\cdot)\|_{W^{1,\infty}(\mathbb{R}^d)} \leq C_{F,2}\, e(\hat{u}).$$

However, the singularity of density $\rho$ requires a specific moment condition to ensure finite variance.

**Assumption E.3** (Integrability of the Residual). *There exists $p \in \mathbb{N}$ with $p \geq 2$ such that for all $t \in [0,T)$ and $q \in [1,p)$:*

$$\int_0^1 \frac{1}{s^{q/2}\rho(s)^{q-1}}\, ds + \sup_{s\in[t,T)} \mathbb{E}\Big[\big|\epsilon(t,\, \boldsymbol{x} + \sigma W_s - \sigma W_t)\big|^q\Big] < \infty. \tag{45}$$

**Remark E.4.** *In Assumption E.3, we explicitly identified $\breve{F}(\mathbf{0}_{d+1})$ with the residual $\epsilon$. This assumption ensures that the surrogate's residual does not grow too fast in expectation along Brownian paths, and that the time sampling density $\rho$ puts sufficient probability mass near $t = 0$ to counteract the singularity.*

### E.3 Main Results

We now show that, with an appropriately trained surrogate model, the `Structural-preserving Law of Defect` can be simulated with lower complexity than the original PDE. In particular, the error of the full-history MLP is upper-bounded by the surrogate model's error measure $e(\hat{u})$.

#### E.3.1 Sketch of Proof

The computational complexity of the MLP solver depends on the Lipschitz constant of the nonlinearity $\breve{F}$ and the magnitude of the "source terms". The "source term" driving the Multilevel Picard simulation for the defect is the residual $\epsilon$, already reduced by the surrogate. At the same time, we show that the regularity in the law of defect is no worse than that of the original PDE, ensuring that the refinement introduces no additional smoothness requirements. Combining the previous fact, a more accurate surrogate makes the defect PDE "easier" to solve. This leads to our main error bound.

1. **Improved Source Magnitude.** Since the source term $\breve{F}(\mathbf{0}_{d+1}, t, \boldsymbol{x}) = \epsilon(t, \boldsymbol{x})$ is the surrogate's residual by Definition D.8. Consequently, as the surrogate improves with additional training data, the variance of the Monte Carlo estimator decreases proportionally. Lemma E.5 formalizes this argument.

2. **Complexity of MLP** Hutzenthaler et al. (2021) show that the complexity of MLP depends on both the smoothness and the size of the source term; these contributions combine multiplicatively since nonlinearities, controlled by their Lipschitz constants, propagate and amplify variance throughout successive Picard iterations. Then we analyze both the magnitude of the source term in Law of Defect and its regularity.

3. **Preservation of Regularity.** By Lemma D.11, the defect nonlinearity $\breve{F}$ inherits the Lipschitz constants $L$ of the original $F$ exactly. Thus, the regularity requirements for the MLP solver remains unchanged, ensuring that the refinement does not introduce additional smoothness constraints.

4. **Error Bound.** Based on the previous inituition, Theorem E.6 bounds the total $L^2$-error as a multiplicative form, combining the classical MLP complexity with the surrogate's approximation error. Thus this can leads to faster convergence rate if the surrogate's approximation error consistently improves.

5. **Complexity Estimate.** Substituting the reduced source magnitude into the standard MLP bound yields a multiplicative error reduction. Combined with Theorem E.11, we improve $O(d\varepsilon^{-(2+\delta)})$ to $O(de(\hat{u})^{2+\delta}\varepsilon^{-(2+\delta)})$.

### E.3.2 Bound on Global $L^2$ Error

Our proof still utilizes the insight that the overall $L^2$ error in the MLP mainly hinges on the Lipschitz continuity of the PDE's terminal and solution, as well as the extent of nonlinearity at the origin. We illustrate that the parameter linked to the `Structural-preserving Law of Defect` is constrained by the surrogate error. Initially, we present a lemma demonstrating how the complexity of MLP can be capped by the error assessment.

**Lemma E.5** (Complexity Estimation via Surrogate Error for Full-History MLP). *Under Assumptions D.6, D.7, E.2, and E.3, suppose $p \geq 2$. There exists a constant $C_F > 0$ independent of the surrogate such that for all $M, N \geq 2$:*

$$
\sup_{(t,\boldsymbol{x})\in[0,T]\times\mathbb{R}^d} \left\{ \frac{\sigma\sqrt{\max\{T-t,3\}}\,\breve{K}}{\sqrt{M}} + \frac{C\,\sup_{s\in[t,T)}\|\breve{F}(\mathbf{0}_{d+1},s,\boldsymbol{x}+\sigma W_s-\sigma W_t)\|_{L^{\frac{2p}{p-2}}(\Omega)}}{2\sqrt{M}} \right.
$$
$$
\left. + \frac{C\,\sup_{s\in[t,T),\,\varsigma\in\{1,\ldots,d+1\}}\|\breve{\mathbf{u}}(s,\boldsymbol{x}+\sigma W_s-\sigma W_t)_\varsigma\|_{L^{\frac{2p}{p-2}}(\Omega)}}{2} \right\} \leq C_F\,e(\hat{u}),
$$
(46)

*where the constant $C$ is defined as:*

$$
C = \max\left\{ 1, 2T^{\frac{1}{2}} \left|\Gamma\left(\frac{p}{2}\right)\right|^{\frac{1}{p}} (1-\alpha)^{\frac{1}{p}-1} \max\{1,L\} \max\left\{ T^{\frac{1}{2}}, 2^{\frac{1}{2}} \left|\Gamma\left(\frac{p+1}{2}\right)\right|^{\frac{1}{p}} \pi^{-\frac{1}{2p}} \right\} \right\}.
$$

*Proof.* We bound the three terms on the left-hand side of equation 46 using the $L^\infty$ bounds provided by the surrogate accuracy Assumption E.2. We utilize the fact that for any bounded random variable $Z$, $\|Z\|_{L^q(\Omega)} \leq \|Z\|_{L^\infty(\Omega)}$.

**Step 1: Bounding the Terminal Condition.** As shown in the proof of Lemma F.4 (Step 2), we have:

$$
\breve{K} \leq \|\breve{u}(T,\cdot)\|_{W^{1,\infty}(\mathbb{R}^d)} \leq \sup_{r\in[0,T]} \|\breve{u}(r,\cdot)\|_{W^{1,\infty}(\mathbb{R}^d)}.
$$
(47)

Applying the Defect Bound from Assumption E.2 (Item 2):

$$
\breve{K} \leq C_{F,2}\,e(\hat{u}).
$$
(48)

**Step 2: Bounding the Residual Term.** Recall that $\breve{F}(\mathbf{0}_{d+1},t,\boldsymbol{x}) = \epsilon(t,\boldsymbol{x})$. The second term involves the $L^{\frac{2p}{p-2}}(\Omega)$ norm of this residual evaluated along Brownian paths. Since the residual is essentially bounded in space-time:

$$
\|\breve{F}(\mathbf{0}_{d+1},s,\boldsymbol{x}+\sigma W_s-\sigma W_t)\|_{L^{\frac{2p}{p-2}}(\Omega)} = \|\epsilon(s,\boldsymbol{x}+\sigma W_s-\sigma W_t)\|_{L^{\frac{2p}{p-2}}(\Omega)}
$$
$$
\leq \sup_{\omega\in\Omega} |\epsilon(s,\boldsymbol{x}+\sigma W_s(\omega)-\sigma W_t(\omega))|
$$
$$
\leq \sup_{y\in\mathbb{R}^d} |\epsilon(s,y)|.
$$
(49)

Applying the Residual Bound from Assumption E.2 (Item 1):

$$
\sup_{s\in[t,T)} \|\breve{F}(\mathbf{0}_{d+1},s,\cdot)\|_{L^{\frac{2p}{p-2}}} \leq C_{F,1}\,e(\hat{u}).
$$
(50)

**Step 3: Bounding the Defect Norm.** The third term involves the $L^{\frac{2p}{p-2}}$ norm of the defect solution $\breve{u}$ and its gradient. Similarly, we bound the stochastic $L^q$ norm by the deterministic uniform norm:

$$
\|\breve{\mathbf{u}}(s,\boldsymbol{x}+\sigma W_s-\sigma W_t)_\varsigma\|_{L^{\frac{2p}{p-2}}(\Omega)} \leq \|\breve{\mathbf{u}}(s,\cdot)\|_{L^\infty(\mathbb{R}^d)}
$$
$$
\leq \|\breve{u}(s,\cdot)\|_{W^{1,\infty}(\mathbb{R}^d)}.
$$
(51)

Applying the Defect Bound from Assumption E.2 (Item 2):

$$
\sup_{s\in[t,T)} \|\breve{u}(s,\cdot)\|_{W^{1,\infty}(\mathbb{R}^d)} \leq C_{F,2}\,e(\hat{u}).
$$
(52)

Substituting the bounds equation 48, equation 50, and equation 52 back into equation 46:

$$\text{LHS} \leq \frac{\sigma\sqrt{T+3}\,C_{F,2}\,e(\hat{u})}{\sqrt{M}} + \frac{C\,C_{F,1}\,e(\hat{u})}{2\sqrt{M}} + \frac{C\,C_{F,2}\,e(\hat{u})}{2}$$

$$= \left[\left(\frac{\sigma\sqrt{T+3}}{\sqrt{M}} + \frac{C}{2}\right)C_{F,2} + \frac{C}{2\sqrt{M}}C_{F,1}\right]e(\hat{u}). \tag{53}$$

Since $M \geq 1$, we can simplify the coefficient by defining $C_F := \sigma\sqrt{T+3}\,C_{F,2} + \frac{C}{2}(C_{F,1} + C_{F,2})$. This proves the lemma. $\square$

The above lemma, together with standard error estimates for the full-history MLP, yields the following result.

**Theorem E.6** (Bound of Global $L^2$ Error). *Under assumptions D.6, D.7, E.3 and E.2 , suppose $p \geq 2$, $\alpha \in (\frac{p-2}{2(p-1)}, \frac{p}{2(p-1)})$, $t \in [0,T)$, $x \in \mathbb{R}^d$, $\beta = \frac{\alpha}{2} - \frac{(1-\alpha)(p-2)}{2p}$. For $\breve{U}_{N,M}(t,\boldsymbol{x})$ with level $N$ and sample base $M$ as defined in Algorithm 1, it holds that*

$$\sup_{(t,\boldsymbol{x})\in[0,T]\times\mathbb{R}^d} \max_{\varsigma\in\{1,\ldots,d+1\}} \left\|\left(\breve{\mathbf{U}}_{N,M}(t,\boldsymbol{x}) - \breve{\mathbf{u}}(t,\boldsymbol{x})\right)_\varsigma\right\|_{L^2} \leq E(M,N)\cdot\Big(C_F\,e(\hat{u})\Big), \tag{54}$$

*where* $E(M,N) = \dfrac{\left[e\left(\frac{pN}{2}+1\right)\right]^{\frac{1}{8}}(2C)^{N-1}\exp\left(\beta M^{\frac{1}{2\beta}}\right)}{\sqrt{M^{N-1}}}.$

*Proof.* Under assumptions E.3 and D.6, combined with the integrability argument in (Hutzenthaler et al., 2021, Lemma 3.3), the proof of (Hutzenthaler et al., 2021, Proposition 3.5) holds. Setting $n = N$ in this proposition, for all $\varsigma \in \{1,\ldots,d+1\}$, we have

$$\left\|\left(\breve{\mathbf{U}}_{N,M}(t,\boldsymbol{x}) - \breve{\mathbf{u}}(t,\boldsymbol{x})\right)_\varsigma\right\|_{L^2} \leq E(M,N)\cdot\left\{\frac{\sigma\sqrt{\max\{T-t,3\}}\breve{K}}{\sqrt{M}}\right. \tag{55}$$

$$+ \frac{C\,\sup_{s\in[t,T)}\|\breve{F}(\mathbf{0}_{d+1})(s,\boldsymbol{x}+\sigma W_s-\sigma W_t)\|_{L^{\frac{2p}{p-2}}}}{2\sqrt{M}} \tag{56}$$

$$\left. + \frac{C\,\sup_{s\in[t,T),\,\varsigma\in\{1,\ldots,d+1\}}\|\breve{\mathbf{u}}(s,\boldsymbol{x}+\sigma W_s-\sigma W_t)_\varsigma\|_{L^{\frac{2p}{p-2}}}}{2}\right\}. \tag{57}$$

Take $\sup_{(t,\boldsymbol{x})\in[0,T]\times\mathbb{R}^d}\max_{\varsigma\in\{1,\ldots,d+1\}}$ for the LHS, and note that the RHS does not depend on $\varsigma$, we get

$$\sup_{(t,\boldsymbol{x})\in[0,T]\times\mathbb{R}^d} \max_{\varsigma\in\{1,\ldots,d+1\}} \left\|\left(\breve{\mathbf{U}}_{N,M}(t,\boldsymbol{x}) - \breve{\mathbf{u}}(t,\boldsymbol{x})\right)_\varsigma\right\|_{L^2} \tag{58}$$

$$\leq E(M,N)\cdot\sup_{(t,\boldsymbol{x})\in[0,T]\times\mathbb{R}^d}\left\{\frac{\sigma\sqrt{\max\{T-t,3\}}\breve{K}}{\sqrt{M}} + \frac{C\,\sup_{s\in[t,T)}\|\breve{F}(\mathbf{0}_{d+1})(s,\boldsymbol{x}+\sigma W_s-\sigma W_t)\|_{L^{\frac{2p}{p-2}}}}{2\sqrt{M}}\right. \tag{59}$$

$$\left. + \frac{C\,\sup_{s\in[t,T),\,\varsigma\in\{1,\ldots,d+1\}}\|\breve{\mathbf{u}}(s,\boldsymbol{x}+\sigma W_s-\sigma W_t)_\varsigma\|_{L^{\frac{2p}{p-2}}}}{2}\right\}. \tag{60}$$

Substituting the sup term in 58 by Lemma E.5 immediately yields the stated result. $\square$

In practice, a common choice for $M$ is $\lfloor N^{2\beta N}\rfloor$. Plugging it in E.7, we get the error order of the solver w.r.t. $N$:

**Corollary E.7** (Error Order for $M = \lfloor N^{2\beta N}\rfloor$). *Under assumptions D.6, D.7, E.3 and E.2 , suppose $p \geq 2$, $\alpha \in (\frac{p-2}{2(p-1)}, \frac{p}{2(p-1)})$, $t \in [0,T)$, $x \in \mathbb{R}^d$, $\beta = \frac{\alpha}{2} - \frac{(1-\alpha)(p-2)}{2p}$. It holds that*

$$\sup_{(t,\boldsymbol{x})\in[0,T]\times\mathbb{R}^d} \max_{\varsigma\in\{1,\ldots,d+1\}} \left\|\left(\breve{\mathbf{U}}_{N,M}(t,\boldsymbol{x}) - \breve{\mathbf{u}}(t,\boldsymbol{x})\right)_\varsigma\right\|_{L^2} \leq \exp\Big(N\log N\big(-\beta + o(1)\big)\Big)\cdot\Big(C_F\,e(\hat{u})\Big). \tag{61}$$

*Proof.* First, we rewrite $E(M, N)$ in E.6 as exponential form:

$$E(M, N) = \frac{\left[e\left(\frac{pN}{2} + 1\right)\right]^{\frac{1}{8}} (2C)^{N-1} \exp\left(\beta M^{\frac{1}{2\beta}}\right)}{\sqrt{M^{N-1}}} \tag{62}$$

$$= \exp\left(o(N) + N\log(2C) + \beta M^{1/2\beta} - \frac{N-1}{2}\log M\right) \tag{63}$$

Note that $\lfloor N^{2\beta N}\rfloor \le N^{2\beta N}$ and that $M = \lfloor N^{2\beta N}\rfloor \Rightarrow \log M \ge \log(N^{2\beta N} - 1) \ge \log(N^{2\beta N}) - 1 = 2\beta\log N - 1$ for $N \ge 2^{1/2\beta}$. We can simplify 62 to

$$\exp\left(o(N) + N\log(2C) + \beta M^{1/2\beta} - \frac{N-1}{2}\log M\right) \tag{64}$$

$$\le \exp\left(o(N) + N\log(2C) + \beta N - \frac{N-1}{2}(2\beta\log N - 1)\right) \tag{65}$$

$$= \exp\left(N(o(1) + \log(2C) + \beta - \frac{1}{2}(2\beta\log N - 1))\right) \tag{66}$$

$$= \exp\left(N\log N(-\beta + o(1))\right). \tag{67}$$

Plugging 64 to the conclusion of E.6, we get the result we want. $\qquad\square$

**Corollary E.8** (Improved Scaling Law for $M = \lfloor N^{2\beta N}\rfloor$). *Under Assumptions D.6, D.7, E.3 and E.2, suppose that $p \ge 2$, $\alpha \in \left(\frac{p-2}{2(p-1)}, \frac{p}{2(p-1)}\right)$, $t \in [0, T)$, $x \in \mathbb{R}^d$, and define $\beta = \frac{\alpha}{2} - \frac{(1-\alpha)(p-2)}{2p}$. Assume that the error at $(t, x)$ of the surrogate model decays polynomially with respect to the number of training points; namely, $e(\hat{u}) = O(m^{-\gamma})$, for some $\gamma > 0$. Suppose further that $m = (d+1)5^N N^{2\beta N}$. Then, for all sufficiently large $m$, the SCaSML procedure improves the error bound from $O(m^{-\gamma})$ to $O\left(m^{-\gamma - \frac{1}{2} + o(1)}\right)$ with same points number.*

*Proof.* In what follows, we adopt the notation $f(m) \sim g(m)$ to signify that $\lim_{m\to\infty}\frac{f(m)}{g(m)} = 1$. Since $m$ is a continuous and strictly increasing function of $N$, there exists a unique inverse function $N = N(m)$. Taking logarithms, we obtain $\log m = \log(d+1) + N(m)\log 5 + 2\beta N(m)\log N(m)$ which follows immediately that $\log m \sim 2\beta N(m)\log N(m)$

Define

$$z = \frac{\log m}{2\beta} - \frac{\log(d+1) + N(m)\log 5}{2\beta}$$

and set $x = \log N$, so that the relation $x\,e^x = z$ holds. The inverse of this equation is given by the Lambert $W$ function, i.e., $x = W(z)$. Therefore,

$$N(m) = e^x = e^{W(z)} = \frac{z}{W(z)} = \frac{\frac{\log m}{2\beta} - \frac{\log(d+1) + N(m)\log 5}{2\beta}}{W\left(\frac{\log m}{2\beta} - \frac{\log(d+1) + N(m)\log 5}{2\beta}\right)}.$$

Since $W(z) \sim \log z - \log\log z$, we can deduce that $N(m) \sim \frac{\frac{\log m}{2\beta}}{\log(\log m)} = \frac{\log m}{2\beta\log\log m}$. Equivalently, $N(m) = \frac{\log m}{2\beta\log\log m} + o\left(\frac{\log m}{2\beta\log\log m}\right)$.

In contrast to the surrogate model, which uses all $m$ points to achieve an error of $O(m^{-\gamma})$, the SCaSML method allocates $5^N N^{2\beta N}$ points for training and $d5^N N^{2\beta N}$ points for inference (see Footnote E.9), thereby yielding an error bound of the form $O\left(N\log N\left(-\beta + o(1)\right)(5^N N^{2\beta N})^{-\gamma}\right) = O\left(N^{-\beta N(1+o(1))} m^{-\gamma}\right)$. Substituting the asymptotic expression for $N(m)$, and noting that

$$N(m)^{-\beta N(m)} = \frac{\sqrt{d+1}\exp(\frac{\log 5}{2}N(m))}{\sqrt{m}} \tag{68}$$

$$=\sqrt{d+1}\exp(\frac{\log 5}{2}\log m(\frac{N(m)}{\log m}-\frac{1}{\log 5})) \tag{69}$$

$$=\sqrt{d+1}\exp(\frac{\log 5}{2}\log m(-\frac{1}{\log 5}+o(\frac{1}{\log\log m}))) \tag{70}$$

$$=\sqrt{d+1}\exp(-(\frac{1}{2}-o(\frac{1}{\log\log m}))\log m) \tag{71}$$

$$=\sqrt{d+1}m^{-\frac{1}{2}+o(\frac{1}{\log\log m})}=O(m^{-\frac{1}{2}+o(\frac{1}{\log\log m})}). \tag{72}$$

We obtain the SCaSML error bound $O\Big(m^{-\gamma}m^{(-\frac{1}{2}+o(\frac{1}{\log\log m}))(1+o(1))}\Big)=O\Big(m^{-\gamma-\frac{1}{2}+o(1)}\Big)$. Hence, for high-dimensional problems where $m \gg 1$ and for any fixed $\gamma > 0$, we conclude that $O\Big(m^{-\gamma-\frac{1}{2}+o(1)}\Big) \ll O\Big(m^{-\gamma}\Big)$, thereby demonstrating that the SCaSML procedure attains a strictly faster rate of convergence. □

**Corollary E.9** (Error Order for $M = \lfloor N^{2\beta N}\rfloor$). *Under Assumptions E.2, E.3, D.6 and D.7 , suppose $p \geq 2$, $\alpha \in (\frac{p-2}{2(p-1)}, \frac{p}{2(p-1)})$, $t \in [0,T)$, $x \in \mathbb{R}^d$, $\beta = \frac{\alpha}{2} - \frac{(1-\alpha)(p-2)}{2p}$. It holds that*

$$\sup_{(t,\boldsymbol{x})\in[0,T]\times\mathbb{R}^d}\max_{\varsigma\in\{1,\dots,d+1\}}\left\|\Big(\check{\mathbf{U}}_{N,\lfloor N^{2\beta N}\rfloor}(t,\boldsymbol{x})-\check{\mathbf{u}}(t,\boldsymbol{x})\Big)_\varsigma\right\|_{L^2}\leq\exp\Big(N\log N(-\beta+o(1))\Big)\cdot\Big(C_F\,e(\hat{u})\Big). \tag{73}$$

*Specifically, this approximator $\check{\mathbf{U}}_{N,\lfloor N^{2\beta N}\rfloor}$ requires at most $d(5\lfloor N^{2\beta N}\rfloor)^N$ points for evaluation, as detailed in (Hutzenthaler et al., 2020a, Lemma 3.6).*

### E.3.3 BOUND ON COMPUTATIONAL COMPLEXITY

We now define two indicators to quantify the computational complexity of full-history SCaSML:the number of realization variables (RV) and the number of function evaluations (FE).

**Definition E.10** (Computational Complexity of full-history SCaSML). *We define the following complexity:*

- *Let $\{\text{RV}_{n,M}\}_{n,M\in\mathbb{Z}} \subset \mathbb{N}$ satisfy $\text{RV}_{0,M} = 0$ and, for all $n, M \in \mathbb{N}$,*

$$\text{RV}_{n,M} \leq dM^n + \sum_{l=0}^{n-1}\Big[M^{n-l}\Big(1+d+\text{RV}_{l,M}+\mathbf{1}_\mathbb{N}(l)\,\text{RV}_{l-1,M}\Big)\Big]. \tag{74}$$

  *This quantity captures the number of scalar normal and uniform time realizations required to compute one sample of $\check{\mathbf{U}}_{n,M}(s,x)$.*

- *Let $\{\text{FE}_{n,M}\}_{n,M\in\mathbb{Z}} \subset \mathbb{N}$ satisfy $\text{FE}_{0,M} = 0$ and, for all $n, M \in \mathbb{N}$,*

$$\text{FE}_{n,M} \leq M^n + \sum_{l=0}^{n-1}\Big[M^{n-l}\Big(1+\text{FE}_{l,M}+\mathbf{1}_\mathbb{N}(l)+\mathbf{1}_\mathbb{N}(l)\,\text{FE}_{l-1,M}\Big)\Big]. \tag{75}$$

  *This reflects the number of evaluations of $\check{F}$ and $\check{g}$ required to compute one sample of $\check{\mathbf{U}}_{n,M}(s,x)$.*

**Theorem E.11** (Computational Complexity of full-history SCaSML). *Under assumptions D.6, D.7, E.3 and E.2 , suppose $p \geq 2$, $\alpha \in (\frac{p-2}{2(p-1)}, \frac{p}{2(p-1)})$ and $\beta = \frac{\alpha}{2} - \frac{(1-\alpha)(p-2)}{2p} \in (0,\frac{\alpha}{2})$. For any $N \geq 2$ and $\delta > 0$, taking $M = \lfloor N^{2\beta N}\rfloor$, we have*

$$\text{RV}_{N,M}+\text{FE}_{N,M}\leq\exp\left(N\log N\Big(-\beta\delta+o(1)\Big)\right)$$
$$(d+1)(C_F e(\hat{u}))^{2+\delta}\left[\sup_{(t,\boldsymbol{x})\in[0,T]\times\mathbb{R}^d}\max_{\varsigma\in\{1,\dots,d+1\}}\left\|\Big(\tilde{\mathbf{U}}_{N,M}(t,\boldsymbol{x})-\mathbf{u}^\infty(t,\boldsymbol{x})\Big)_\varsigma\right\|_{L^2}\right]^{-(2+\delta)}. \tag{76}$$

*Proof.* From (Hutzenthaler et al., 2020a, Lemma 3.6), we derive that

$$\mathrm{RV}_{N,M} \le d(5M)^N, \mathrm{FE}_{N,M} \le (5M)^N. \tag{77}$$

Suppose the maximum error is $\varepsilon$. To compensate for the $(5M)^N$ term in the complexity by the denominator of Theorem E.6, we multiply the complexity by $\varepsilon^{2+\delta}$ and then divide it, and put everything into the exponent:

$$\mathrm{RV}_{N,M} + \mathrm{FE}_{N,M} \tag{78}$$

$$\le (d+1)(5M)^N \tag{79}$$

$$= (d+1)(5M)^N \varepsilon^{2+\delta} \varepsilon^{-(2+\delta)} \tag{80}$$

$$\le \left[ \frac{\left[ e\left( \frac{pN}{2}+1 \right) \right]^{\frac{1}{8}} (2C)^{N-1} \exp\left( \beta M^{\frac{1}{2\beta}} \right)}{\sqrt{M^{N-1}}} \right]^{2+\delta} (C_F e(\hat{u}))^{2+\delta} (d+1)(5M)^N \tag{81}$$

$$\left[ \sup_{(t,\boldsymbol{x})\in[0,T]\times\mathbb{R}^d} \max_{\varsigma\in\{1,\dots,d+1\}} \left\| \left( \tilde{\mathbf{U}}_{N,M}(t,\boldsymbol{x}) - \mathbf{u}^\infty(t,\boldsymbol{x}) \right)_\varsigma \right\|_{L^2} \right]^{-(2+\delta)} \tag{82}$$

$$= \exp\left( N\left( (2+\delta)\log(2C) + \log 5 \right) + (2+\delta)\beta M^{1/2\beta} + \log M - \frac{\delta}{2}(N-1)\log M + o(N) \right) \tag{83}$$

$$(d+1)(C_F e(\hat{u}))^{2+\delta} \left[ \sup_{(t,\boldsymbol{x})\in[0,T]\times\mathbb{R}^d} \max_{\varsigma\in\{1,\dots,d+1\}} \left\| \left( \tilde{\mathbf{U}}_{N,M}(t,\boldsymbol{x}) - \mathbf{u}^\infty(t,\boldsymbol{x}) \right)_\varsigma \right\|_{L^2} \right]^{-(2+\delta)} \tag{84}$$

$$= \exp\left( N\left( (2+\delta)\log(2C) + \log 5 + \frac{(2+\delta)\beta}{N}M^{1/2\beta} - \left(\frac{\delta}{2} - \frac{2+\delta}{2N}\right)\log M + o(1) \right) \right) \tag{85}$$

$$(d+1)(C_F e(\hat{u}))^{2+\delta} \left[ \sup_{(t,\boldsymbol{x})\in[0,T]\times\mathbb{R}^d} \max_{\varsigma\in\{1,\dots,d+1\}} \left\| \left( \tilde{\mathbf{U}}_{N,M}(t,\boldsymbol{x}) - \mathbf{u}^\infty(t,\boldsymbol{x}) \right)_\varsigma \right\|_{L^2} \right]^{-(2+\delta)} \tag{86}$$

$$\le \exp\left( N\left( (2+\delta)\log(2C) + \log 5 + \frac{(2+\delta)\beta}{N}M^{1/2\beta} - \left(\frac{\delta}{2} - \frac{2+\delta}{2N}\right)\log M + o(1) \right) \right) \tag{87}$$

$$(d+1)(C_F e(\hat{u}))^{2+\delta} \left[ \sup_{(t,\boldsymbol{x})\in[0,T]\times\mathbb{R}^d} \max_{\varsigma\in\{1,\dots,d+1\}} \left\| \left( \tilde{\mathbf{U}}_{N,M}(t,\boldsymbol{x}) - \mathbf{u}^\infty(t,\boldsymbol{x}) \right)_\varsigma \right\|_{L^2} \right]^{-(2+\delta)} \tag{88}$$

Note that $\lfloor N^{2\beta N} \rfloor^{1/2\beta} \le N$ and $\beta < \frac{\alpha}{2}$, thus $\frac{(2+\delta)\beta}{N}M^{1/2\beta} \le (2+\delta)\beta \le \alpha(1+\frac{\delta}{2})$. Therefore, by 78:

$$\exp\left( N\left( (2+\delta)\log(2C) + \log 5 + \frac{(2+\delta)\beta}{N}M^{1/2\beta} - \left(\frac{\delta}{2} - \frac{2+\delta}{2N}\right)\log M + o(1) \right) \right) \tag{89}$$

$$(d+1)(C_F e(\hat{u}))^{2+\delta} \left[ \sup_{(t,\boldsymbol{x})\in[0,T]\times\mathbb{R}^d} \max_{\varsigma\in\{1,\dots,d+1\}} \left\| \left( \tilde{\mathbf{U}}_{N,M}(t,\boldsymbol{x}) - \mathbf{u}^\infty(t,\boldsymbol{x}) \right)_\varsigma \right\|_{L^2} \right]^{-(2+\delta)} \tag{90}$$

$$\le \exp\left( N\left( (2+\delta)\log(2C) + \log 5 + \alpha\left(1+\frac{\delta}{2}\right) - \left(\frac{\delta}{2} - \frac{2+\delta}{2N}\right)\log M + o(1) \right) \right) \tag{91}$$

$$(d+1)(C_F e(\hat{u}))^{2+\delta} \left[ \sup_{(t,\boldsymbol{x})\in[0,T]\times\mathbb{R}^d} \max_{\varsigma\in\{1,\dots,d+1\}} \left\| \left( \tilde{\mathbf{U}}_{N,M}(t,\boldsymbol{x}) - \mathbf{u}^\infty(t,\boldsymbol{x}) \right)_\varsigma \right\|_{L^2} \right]^{-(2+\delta)}. \tag{92}$$

Since $M = \lfloor N^{2\beta N} \rfloor \Rightarrow \log M \geq \log(N^{2\beta N} - 1) \geq \log(N^{2\beta N}) - 1 = 2\beta \log N - 1$ for $N \geq 2^{1/2\beta}$, we can further reduce 89 to:

$$\exp\left(N\left((2+\delta)\log(2C) + \log 5 + \alpha\left(1 + \frac{\delta}{2}\right) - \left(\frac{\delta}{2} - \frac{2+\delta}{2N}\right)\log M + o(1)\right)\right) \tag{93}$$

$$(d+1)(C_F e(\hat{u}))^{2+\delta}\left[\sup_{(t,\boldsymbol{x})\in[0,T]\times\mathbb{R}^d} \max_{\varsigma\in\{1,\dots,d+1\}} \left\|\left(\tilde{\mathbf{U}}_{N,M}(t,\boldsymbol{x}) - \mathbf{u}^\infty(t,\boldsymbol{x})\right)_\varsigma\right\|_{L^2}\right]^{-(2+\delta)} \tag{94}$$

$$\leq \exp\left(N\left((2+\delta)\log(2C) + \log 5 + \alpha\left(1 + \frac{\delta}{2}\right) + \left(\frac{\delta}{2} - \frac{2+\delta}{2N}\right) - \left(\frac{\delta}{2} - \frac{2+\delta}{2N}\right)\cdot 2\beta\log N + o(1)\right)\right) \tag{95}$$

$$(d+1)(C_F e(\hat{u}))^{2+\delta}\left[\sup_{(t,\boldsymbol{x})\in[0,T]\times\mathbb{R}^d} \max_{\varsigma\in\{1,\dots,d+1\}} \left\|\left(\tilde{\mathbf{U}}_{N,M}(t,\boldsymbol{x}) - \mathbf{u}^\infty(t,\boldsymbol{x})\right)_\varsigma\right\|_{L^2}\right]^{-(2+\delta)} \tag{96}$$

$$= \exp\left(N\log N\left(-\beta\delta + o(1)\right)\right) \tag{97}$$

$$(d+1)(C_F e(\hat{u}))^{2+\delta}\left[\sup_{(t,\boldsymbol{x})\in[0,T]\times\mathbb{R}^d} \max_{\varsigma\in\{1,\dots,d+1\}} \left\|\left(\tilde{\mathbf{U}}_{N,M}(t,\boldsymbol{x}) - \mathbf{u}^\infty(t,\boldsymbol{x})\right)_\varsigma\right\|_{L^2}\right]^{-(2+\delta)}. \tag{98}$$

The right-hand side of this expression is clearly decreasing for large enough $N$, and in turn, finite. Hence, quadrature SCaSML boosts a quadrature MLP with complexity $O(d\varepsilon^{-(2+\delta)})$ to a corresponding physics-informed inference solver with complexity $O(de(\hat{u})^{2+\delta}\varepsilon^{-(2+\delta)})$. □

## F   PROOF FOR QUADRATURE MULTILEVEL PICARD ITERATION

In this section, we present the proof for the Quadrature Multilevel Picard (MLP) iteration method. For simplicity, we consider the case where $\mu = 0$ and $\sigma = s\mathbf{I}_d(s \in \mathbb{R})$ in the proof. We first establish the mathematical framework and underlying assumptions, then analyze the convergence properties and computational complexity of our proposed simulation-calibrated variant. The result shows that the error of SCaSML is bounded by the product of MLP error and surrogate error. Likewise, the complexity is bounded by the product of MLP error and surrogate error. Both indicate that surrogate models can substantially reduce computational complexity while maintaining accuracy guarantees.

Since the `Structural-preserving Law of Defect` is also a semi-linear heat equation, we can use the quadrature/full-history multilevel Picard iteration to obtain an estimation $\check{\mathbf{U}}(s, x)$ of $u(s, x) - \hat{u}(s, x)$. In this section, we study the theoretical properties of SCaSML that using Quadrature Multilevel Picard Iteration to solve the `Structural-preserving Law of Defect` and we investigate the full-history multilevel Picard iteration in the next section.

### F.1   SURROGATE ACCURACY AND INTEGRABILITY ASSUMPTIONS

To derive improved convergence rates for the Quadrature MLP, we must quantify the quality of the pre-trained surrogate $\hat{u}$. We introduce a scalar error measure $e(\hat{u}) \in [0, \infty)$ which serves as a uniform bound on both the PDE residual and the approximation error of the surrogate.

**Assumption F.1** (Accuracy of the Surrogate Model for Quadrature MLP). *Assumption needed for quadrature MLP builds directly on Assumption E.2, augmenting it with an additional higher-order regularity condition required for the quadrature rule.*

    3. ***Higher-Order Regularity:*** *To ensure rapid convergence of the time quadrature rules, we assume the defect satisfies the following Gevrey-class regularity bounds:*

$$\sup_{k\in\mathbb{N}_0} \frac{\|(1,\sigma^\top\nabla_{\boldsymbol{x}})\left(\left(\frac{\partial}{\partial t} + \frac{\sigma^2}{2}\Delta_{\boldsymbol{x}}\right)^k\check{u}\right)(t,\boldsymbol{x})\|_{L^\infty}}{(k!)^{3/4}} \leq C_{Q,3}\, e(\hat{u}),$$

> *This condition is required only for the Quadrature MLP variant (Appendix F) and is relaxed for the Full-History variant (Appendix E).*

**Assumption F.2** (Quadrature Integrability). *To ensure the well-posedness of the Feynman-Kac expectations, we assume polynomial growth bounds. There exists $p \in \mathbb{N}$ such that for the zero vector $\mathbf{0}_{d+1} \in \mathbb{R}^{d+1}$:*

$$\sup_{x \in \mathbb{R}^d} \frac{|\breve{g}(\boldsymbol{x})|}{1 + \|\boldsymbol{x}\|_1^p} + \sup_{t \in [0,T], x \in \mathbb{R}^d} \frac{|\breve{F}(\mathbf{0}_{d+1}, t, \boldsymbol{x})|}{1 + \|\boldsymbol{x}\|_1^p} < \infty. \tag{99}$$

**Remark F.3** (Magnitude of Nonlinearity at Zero). *Recall from Definition D.8 that $\breve{F}(\mathbf{0}_{d+1}, t, \boldsymbol{x}) \equiv \epsilon(t, \boldsymbol{x})$. Thus, the second term in Assumption F.2 effectively bounds the growth of the surrogate's residual. In the standard Picard iteration for the defect $\breve{u}$, the first iteration is driven solely by this term. A small "magnitude at zero" implies that the fixed-point iteration starts very close to the true solution (zero), minimizing the Monte Carlo work required.*

## F.2 MAIN RESULTS

We now present our main theoretical results, which characterize both the accuracy and computational complexity of our proposed method. These results demonstrate the substantial efficiency gains achieved by incorporating surrogate models into the multilevel Picard framework.

### F.2.1 BOUND ON GLOBAL $L^2$ ERROR

Follows the sam proof sketch as Section E.3.1,the convergence analysis proceeds in two steps. First, we establish a "Bridge Lemma" that bounds the complexity-determining constants of the Defect PDE (specifically the magnitude of the nonlinearity at zero and the Lipschitz constant of the terminal condition) linearly by the surrogate error $e(\hat{u})$. Second, we substitute these bounds into the standard error estimate for Multilevel Picard iterations to prove that the final error is the product of the simulation error and the surrogate error.

**Lemma F.4** (The Bridge Lemma: Complexity Estimation via Surrogate Error). *Suppose Assumptions D.6, D.7, F.1, and F.2 hold. Then there exists a constant $C_Q > 0$ independent of the surrogate $\hat{u}$ such that:*

$$\sup_{(t,\boldsymbol{x}) \in [0,T] \times \mathbb{R}^d} \left\{ \left|\breve{F}(\mathbf{0}_{d+1}, t, \boldsymbol{x})\right| + \sigma\sqrt{T+3}\,\breve{K} + \sup_{k \in \mathbb{N}} \frac{\|(1, \nabla_{\boldsymbol{x}})\left((\frac{\partial}{\partial t} + \frac{\sigma^2}{2}\Delta_{\boldsymbol{x}})^k \breve{u}\right)(t, \boldsymbol{x})\|_{L^\infty}}{(k!)^{3/4}} \right\} \le C_Q\, e(\hat{u}). \tag{100}$$

*Proof.* We bound each of the three terms on the left-hand side of equation 100 using the surrogate accuracy assumptions defined in Assumption F.1.

**Step 1: Bounding the Residual Term.** Recall from Remark F.3 that $\breve{F}(\mathbf{0}_{d+1}, t, \boldsymbol{x}) \equiv \epsilon(t, \boldsymbol{x})$. Applying the $L^\infty$ residual bound from Assumption F.1 (Item 1):

$$\sup_{(t,\boldsymbol{x}) \in [0,T] \times \mathbb{R}^d} |\breve{F}(\mathbf{0}_{d+1}, t, \boldsymbol{x})| = \sup_{(t,\boldsymbol{x}) \in [0,T] \times \mathbb{R}^d} |\epsilon(t, \boldsymbol{x})| \le C_{Q,1}\, e(\hat{u}). \tag{101}$$

**Step 2: Bounding the Terminal Lipschitz Constant.** Let $e_\alpha$ be the standard basis vector at index $\alpha$ in $\mathbb{R}^d$. The $L^1$ norm of $\breve{K}$ satisfies:

$$\breve{K} \le \sum_{\alpha=1}^{d} \|D^{e_\alpha}\breve{g}\|_{L^\infty} = \sum_{\alpha=1}^{d} \|D^{e_\alpha}\breve{u}(T, \cdot)\|_{L^\infty} \le \|\breve{u}(T, \cdot)\|_{W^{1,\infty}(\mathbb{R}^d)}. \tag{102}$$

Using the Defect Bound from Assumption F.1 (Item 2):

$$\breve{K} \le \sup_{t \in [0,T]} \|\breve{u}(t, \cdot)\|_{W^{1,\infty}} \le C_{Q,2}\, e(\hat{u}). \tag{103}$$

**Step 3: Bounding the Higher-Order Regularity Term.** The third term is directly controlled by the Higher-Order Regularity condition in Assumption F.1 (Item 3):

$$\sup_{k \in \mathbb{N}} \frac{\|(1, \nabla_{\boldsymbol{x}})((\frac{\partial}{\partial t} + \frac{\sigma^2}{2}\Delta_{\boldsymbol{x}})^k \breve{u})(t, \boldsymbol{x})\|_{L^\infty}}{(k!)^{3/4}} \le C_{Q,3}\, e(\hat{u}). \tag{104}$$

Summing the bounds from Steps 1-3, we define $C_Q := C_{Q,1} + \sigma\sqrt{T+3}\,C_{Q,2} + C_{Q,3}$. This yields the desired inequality. $\square$

We now combine this lemma with the general convergence theory of Multilevel Picard iterations to state our main result.

**Theorem F.5** (Global $L^2$ Error Bound). *Under Assumptions D.6, D.7, F.1, and F.2, the error of the SCaSML estimator $\breve{\mathbf{U}}_{N,N,N}$ with level $N$, sample base $N$ and quadrature order $N$(as defined in Algorithm 1) satisfies:*

$$\sup_{(t,\boldsymbol{x})\in[0,T]\times\mathbb{R}^d} \max_{\varsigma\in\{1,\dots,d+1\}} \left\| \left(\breve{\mathbf{U}}_{N,N,N}(t,\boldsymbol{x}) - (\breve{u}(t,\boldsymbol{x}), \sigma\nabla_{\boldsymbol{x}}\breve{u}(t,\boldsymbol{x}))\right)_\varsigma \right\|_{L^2} \le E(N)\cdot\left(C_Q\,e(\hat{u})\right), \tag{105}$$

*where the convergence factor $E(N)$ is defined as:*

$$E(N) = \frac{7C^N\,2^{N-1}e^N}{\sqrt{N^{N-3}}} + \frac{\left(14(4C)^{N-1}+1\right)T^{2N+1}}{\sqrt{N^N}},$$

*with constant $C = 2(\sqrt{T}+1)\sqrt{T\pi}(L+1)+1$.*

*Proof.* We apply the general error bound for Quadrature MLP from (Hutzenthaler & Kruse, 2020) to the specific case of the Defect PDE. (Hutzenthaler & Kruse, 2020, Corollary 4.7) provides a bound of the form:

$$\sup_{(t,\boldsymbol{x})\in[0,T]\times\mathbb{R}^d} \max_{\varsigma\in\{1,\dots,d+1\}} \left\| \left(\breve{\mathbf{U}}_{N,N,N}(t,\boldsymbol{x}) - (\breve{u}(t,\boldsymbol{x}), \sigma\nabla_{\boldsymbol{x}}\breve{u}(t,\boldsymbol{x}))\right)_\varsigma \right\|_{L^2} \tag{106}$$

$$\le E(N) \times \sup_{(t,\boldsymbol{x})\in[0,T]\times\mathbb{R}^d} \left\{ \left|\breve{F}(\mathbf{0}_{d+1},t,\boldsymbol{x})\right| + \sigma\sqrt{T+3}\,\breve{K} + \sup_{k\in\mathbb{N}} \frac{\left\|(1,\nabla_{\boldsymbol{x}})\left((\frac{\partial}{\partial t}+\frac{\sigma^2}{2}\Delta_{\boldsymbol{x}})^k\breve{u}\right)(t,\boldsymbol{x})\right\|_{L^\infty}}{(k!)^{3/4}} \right\}. \tag{107}$$

Specifically, the second term is the supremum bounded in Lemma F.4. By substituting the result of Lemma F.4 directly into the corollary, we replace the generic PDE constants with the term $C_Q\,e(\hat{u})$, thereby proving the factorization. $\square$

**Corollary F.6** (Asymptotic Error Decay). *Under the assumptions of Theorem F.5, the convergence factor $E(N)$ satisfies the following asymptotic bound as $N \to \infty$:*

$$E(N) = \exp\left(-\frac{1}{2}N\log N + O(N)\right). \tag{108}$$

*Consequently, the error decays super-polynomially with respect to the computational depth $N$.*

*Proof.* We determine the leading order asymptotic behavior of $\log E(N)$ by analyzing the two summands in the definition of $E(N)$ separately. Recall:

$$E(N) = \underbrace{\frac{7C^N\,2^{N-1}e^N}{\sqrt{N^{N-3}}}}_{=:T_1(N)} + \underbrace{\frac{\left(14(4C)^{N-1}+1\right)T^{2N+1}}{\sqrt{N^N}}}_{=:T_2(N)}.$$

**Step 1: Asymptotic of the First Term $T_1(N)$.**
Taking the natural logarithm of $T_1(N)$:

$$\log T_1(N) = \log(7\cdot 2^{-1}) + N\log(2Ce) - \frac{N-3}{2}\log N \tag{109}$$

$$= -\frac{1}{2}N\log N + N\log(2Ce) + \frac{3}{2}\log N + \log(3.5). \tag{110}$$

Observing that as $N \to \infty$, the term $-\frac{1}{2}N\log N$ dominates linear terms $O(N)$, we have:

$$\log T_1(N) = -\frac{1}{2}N\log N + O(N). \tag{111}$$

**Step 2: Asymptotic of the Second Term $T_2(N)$.**
We bound the numerator: $14(4C)^{N-1} + 1 \le 15(4C)^{N-1}$ for sufficiently large $C, N$. Thus:

$$\log T_2(N) \le \log\left(15(4C)^{N-1}T^{2N+1}\right) - \frac{N}{2}\log N \tag{112}$$

$$= \log(15 \cdot (4C)^{-1} \cdot T) + N\log(4C) + 2N\log T - \frac{1}{2}N\log N \tag{113}$$

$$= -\frac{1}{2}N\log N + N(\log(4C) + 2\log T) + O(1). \tag{114}$$

Similar to Step 1, the dominant term is $-\frac{1}{2}N\log N$:

$$\log T_2(N) = -\frac{1}{2}N\log N + O(N). \tag{115}$$

Since $E(N) = T_1(N) + T_2(N)$, we have $\log E(N) \le \log(2\max\{T_1, T_2\}) = \log 2 + \max\{\log T_1, \log T_2\}$. Substituting equation 111 and equation 115:

$$\log E(N) \le \log 2 + \left(-\frac{1}{2}N\log N + O(N)\right) = -\frac{1}{2}N\log N + O(N).$$

Exponentiating both sides yields the claim. $\qquad\square$

### F.2.2  BOUND ON COMPUTATIONAL COMPLEXITY

To fully assess the efficiency of our method, we now analyze its computational complexity. We introduce two key metrics that capture different aspects of the computational cost.

**Definition F.7** (Computational Complexity of Quadrature SCaSML). *We define the following complexity measures:*

*First, let $\{\mathrm{RN}_{n,M,Q}\}_{n,M,Q\in\mathbb{Z}} \subset \mathbb{N}$ satisfy $\mathrm{RN}_{0,M,Q} = 0$ and, for all $n, M, Q \in \mathbb{N}$,*

$$\mathrm{RN}_{n,M,Q} \le d\,M^n + \sum_{l=0}^{n-1}\left[Q\,M^{n-l}\left(d + \mathrm{RN}_{l,M,Q} + \mathbf{1}_{\mathbb{N}}(l)\,\mathrm{RN}_{l-1,M,Q}\right)\right]. \tag{116}$$

*This number represents the total scalar normal random variable realizations required for computing one sample of $\check{\mathbf{U}}_{n,M,Q}(s,x)$.*

*Second, let $\{\mathrm{FE}_{n,M,Q}\}_{n,M,Q\in\mathbb{Z}} \subset \mathbb{N}$ satisfy $\mathrm{FE}_{0,M,Q} = 0$ and, for all $n, M, Q \in \mathbb{N}$,*

$$\mathrm{FE}_{n,M,Q} \le M^n + \sum_{l=0}^{n-1}\left[Q\,M^{n-l}\left(1 + \mathrm{FE}_{l,M,Q} + \mathbf{1}_{\mathbb{N}}(l) + \mathbf{1}_{\mathbb{N}}(l)\,\mathrm{FE}_{l-1,M,Q}\right)\right]. \tag{117}$$

*This quantity reflects the number of evaluations of $\check{F}$ and $\check{g}$ necessary to compute of one sample of $\check{\mathbf{U}}_{n,M,Q}(s,x)$.*

These metrics provide a comprehensive measure of the computational resources required by our method. The first metric, $\mathrm{RN}_{n,M,Q}$, accounts for the cost of generating random variables, while the second, $\mathrm{FE}_{n,M,Q}$, captures the number of function evaluations needed.

**Theorem F.8** (Complexity of Quadrature SCaSML). *Under assumptions D.6, D.7, F.2 and F.1, for any $\delta > 0$ and all $N \in \mathbb{N}$, we have*

$$\mathrm{RN}_{N,N,N} + \mathrm{FE}_{N,N,N} \le \left[\sup_{(t,\boldsymbol{x})\in[0,T]\times\mathbb{R}^d}\max_{\varsigma\in\{1,\ldots,d+1\}}\left\|\left(\tilde{\mathbf{U}}_{N,N,N}(t,\boldsymbol{x}) - \mathbf{u}^\infty(t,\boldsymbol{x})\right)_\varsigma\right\|_{L^2}\right]^{-(4+\delta)}$$

$$\cdot 8(d+1)(C_Q e(\hat{u}))^{4+\delta}\exp\left(N\log N(-\frac{\delta}{2} + o(1))\right) < \infty. \tag{118}$$

*Proof.* From established results in (Hutzenthaler et al., 2020a, Lemma 3.6), we know that for all $N \in \mathbb{N}$,

$$\mathrm{RN}_{N,N,N} \leq 8dN^{2N}, \quad \mathrm{FE}_{N,N,N} \leq 8N^{2N}. \tag{119}$$

We want to use the $O(N^{N/2})$ denominator in Theorem F.5 to compensate for the $N^{2N}$ term in the complexity. Suppose the maximum error is $\varepsilon$, and note that $N^{2N} = (N^{N/2})^4 < (N^{N/2})^{4+\delta}, \forall \delta > 0$, we multiply the complexity by $\varepsilon^{4+\delta}$, i.e.

$$(\mathrm{RN}_{N,N,N} + \mathrm{FE}_{N,N,N}) \left[ \sup_{(t,\boldsymbol{x}) \in [0,T] \times \mathbb{R}^d} \max_{\varsigma \in \{1,\dots,d+1\}} \left\| \left( \tilde{\mathbf{U}}_{N,N,N}(t,\boldsymbol{x}) - \mathbf{u}^\infty(t,\boldsymbol{x}) \right)_\varsigma \right\|_{L^2} \right]^{(4+\delta)}$$

$$\leq 8(d+1)N^{2N} \cdot \left( \frac{7 \left( 2(\sqrt{T}+1)\sqrt{T\pi}(L+1) + 1 \right)^N 2^{N-1} e^N}{\sqrt{N^{N-3}}} \right.$$

$$\left. + \frac{(14(8(\sqrt{T}+1)\sqrt{T\pi}(L+1) + 4)^{N-1} + 1)T^{2N+1}}{\sqrt{N^N}} \right)^{(4+\delta)} (C_Q e(\hat{u}))^{4+\delta}$$

$$\leq 8(d+1)N^{2N} \cdot \left( (24(T+1))^{3N} (L+1)^N \sqrt{N}^{-N} \right)^{(4+\delta)} (C_Q e(\hat{u}))^{4+\delta}$$

$$\leq 8(d+1)(C_Q e(\hat{u}))^{4+\delta} \exp\left( N \log N(-\frac{\delta}{2} + o(1)) \right).$$

$$\tag{120}$$

The right-hand side of this expression is clearly decreasing for large enough $N$, and in turn, finite. Hence, quadrature SCaSML boosts a quadrature MLP with complexity $O(d\varepsilon^{-(4+\delta)})$ to a corresponding physics-informed inference solver with complexity $O(de(\hat{u})^{4+\delta}\varepsilon^{-(4+\delta)})$ $\qquad \square$

This theorem provides a comprehensive characterization of the computational complexity of our method. The inclusion of the surrogate model error measure $e(\hat{u})$ in the complexity bound demonstrates how the quality of the surrogate model directly influences the computational efficiency of our approach. Specifically, a more accurate surrogate model (smaller $e(\hat{u})$) leads to a lower computational cost for achieving a given level of accuracy.

# G  AUXILIARY EXPERIMENTS RESULTS

We include supplementary experimental results that further validate our claims, including detailed error distribution plots (violin plots) and additional inference-time scaling curves for all PDE test cases.

## G.1  VIOLIN PLOT FOR ERROR DISTRIBUTION

In this section, we present violin plots of the absolute error distributions for the base surrogate model , the MLP, and the SCaSML method. We uniformly select the test points. By combining kernel density estimation with boxplot-style summaries, these plots capture both the spread and central tendency of the errors. A violin plot exposes the full distribution—its density, variability, skewness, and outliers—offering much deeper insight into model performance. The width of each violin at a given error level reflects the density of the observations. The results indicate that SCaSML reduces the largest absolute error, lowers the median and produces more accurate points for a majority of equations compared to the surrogate and MLP, demonstrating its robustness across different dimensions and equations.

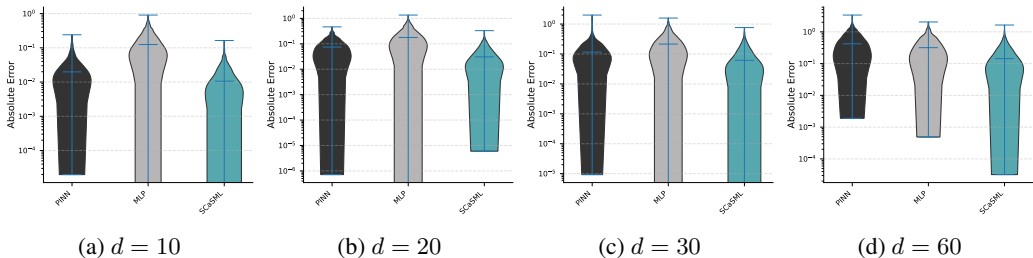

| (a) $d = 10$ | (b) $d = 20$ | (c) $d = 30$ | (d) $d = 60$ |

Figure 5: Violin Plot for comparison of the baseline PINN surrogate (black), MLP (gray), applying qudrature SCaSML (teal) to calibrate the PINN surrogate on linear convection-diffusion equation for $d = 10, 20, 30, 60$.

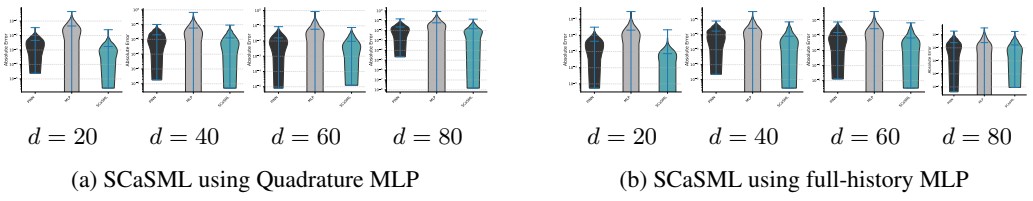

| $d = 20$    $d = 40$    $d = 60$    $d = 80$ | $d = 20$    $d = 40$    $d = 60$    $d = 80$ |

(a) SCaSML using Quadrature MLP        (b) SCaSML using full-history MLP

Figure 6: Violin Plot for comparison of the baseline PINN surrogate (black), MLP (gray), applying qudrature SCaSML (teal) to calibrate the PINN surrogate on viscous Burgers' equation equation for $d = 10, 20, 30, 60$.

### G.2 INFERENCE TIME SCALING CURVE

In this section, we illustrate how SCaSML enhances estimation accuracy as the number of inference-time collocation points increases, as outlined in 2.1 and 2.2. Our findings indicate that allocating additional computational resources during inference consistently improves estimation accuracy.

### G.3 IMPROVED SCALING LAW OF SCASML ALGORITHMS

In this section, we consider the viscous Burgers equation as an illustrative example to demonstrate the improved convergence of SCaSML algorithms, as suggested by Corollary E.8.

We implemented a physics-informed neural network (PINN) with five hidden layers, each containing 50 neurons and employing hyperbolic tangent activation functions. Because the number of training points, $m$, is proportional to the number of iterations in the PINN, the control group was trained using the Adam optimizer (learning rate $7 \times 10^{-4}$, $\beta_1 = 0.9$, $\beta_2 = 0.99$) over iterations set to 400, 2 000, 4 000, 6 000, 8 000, and 10 000 (as illustrated along the $x$-axis). The dataset comprised 2 500 interior points, 100 boundary points, and 160 initial points uniformly sampled from $[0, 0.5] \times [-0.5, 0.5]^d$, ensuring that $m \gg 1$. To replicate the conditions of Corollary E.8, the SCaSML group was trained over iterations set to $\lfloor 400/(d+1) \rfloor$, $\lfloor 2\,000/(d+1) \rfloor$, $\lfloor 1\,000/(d+1) \rfloor$, $\lfloor 6\,000/(d+1) \rfloor$, $\lfloor 8\,000/(d+1) \rfloor$, and $\lfloor 10\,000/(d+1) \rfloor$. In addition, we set the inference level as $N = \lfloor \log m / (2\beta \log \log m) \rfloor$ with $\beta = 1/2$. Theoretically, SCaSML exhibits an improvement in $\gamma$ of $\frac{1}{2} + o(1)$ relative to the control group.

For the Gaussian process regression surrogate model, training was performed over 20 iterations using Newton's method. Due to the increasing inference parameters with $m$ and the consequent GPU memory constraints, it was not possible to replicate the conditions of Corollary E.8 exactly for the Gaussian process model. Consequently, both the control and SCaSML groups employed identical training sizes, which theoretically does not alter the asymptopic convergence rate(i.e. the slope). Specifically, the training data consisted of the following pairs of interior and boundary points:(100, 20), (200, 40), (300, 60), (400, 80), (500, 100), (600, 120), (700, 140), (800, 160), (900, 180), and (1 000, 200), with the $x$-axis representing the total number of training points. Again, the inference level was chosen as $N = \lfloor \log m / (2\beta \log \log m) \rfloor$ with $\beta = 1/2$, and the SCaSML continues to exhibit an improvement in $\gamma$ of $\frac{1}{2} + o(1)$ relative to the control group.

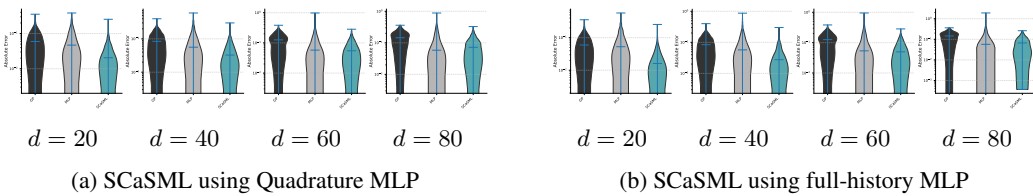

(a) SCaSML using Quadrature MLP                    (b) SCaSML using full-history MLP

Figure 7: Violin Plot for comparison of the baseline Gaussian Process surrogate (black), MLP (gray), applying qudrature SCaSML (teal) to calibrate the Gaussian Process surrogate on viscous Burgers' equation equation for $d = 20, 40, 60, 80$.

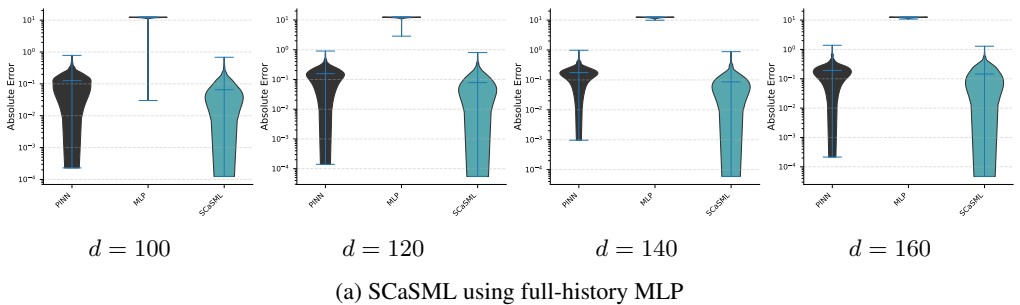

(a) SCaSML using full-history MLP

Figure 8: Violin Plot for comparison of the baseline PINN surrogate (black), MLP (gray), applying qudrature SCaSML (teal) to calibrate the PINN surrogate on LQG control problem for $d = 100, 120, 140, 160$.

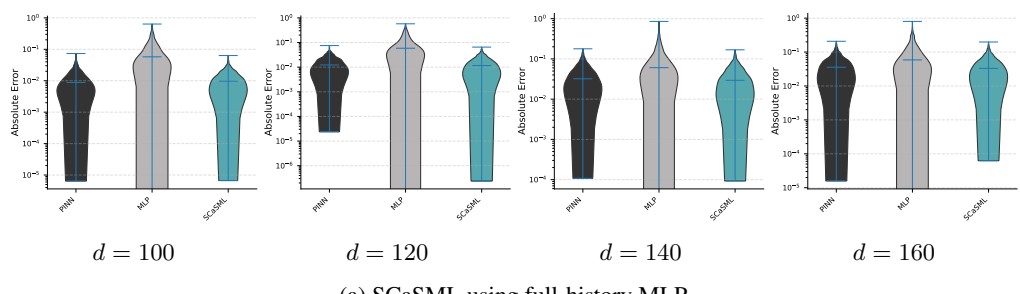

(a) SCaSML using full-history MLP

Figure 9: Violin Plot for comparison of the baseline PINN surrogate (black), MLP (gray), applying qudrature SCaSML (teal) to calibrate the PINN surrogate on diffusion reaction equation for $d = 100, 120, 140, 160$.

We observe that, for the PINNs, full-history SCaSML achieves near-monotonic error reduction across resolutions (with $d$ ranging from 20 to 80), outperforming quadrature SCaSML, which displays oscillatory behavior at higher dimensions. The Gaussian process-based SCaSML similarly accelerates convergence during training. In both cases, the error trajectories generated by SCaSML are generally shifted downward relative to the base models, underscoring its capacity to enhance accuracy without altering the fundamental training dynamics. These findings underscore SCaSML's robustness in diverse settings, ensuring reliable convergence even in high-dimensional or non-monotonic scenarios.

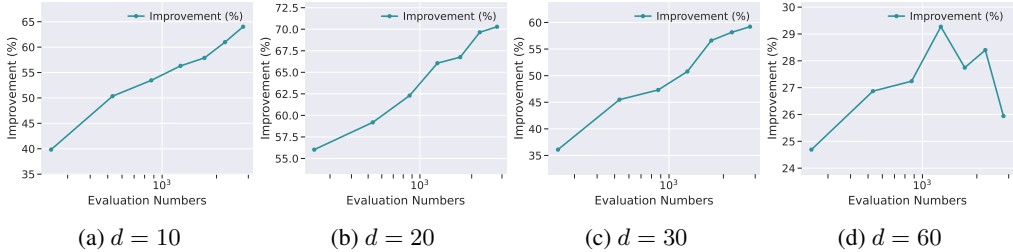

(a) $d = 10$  (b) $d = 20$  (c) $d = 30$  (d) $d = 60$

Figure 10: For the linear convection-diffusion equation, SCaSML for PINNs reliably enhances performance with increased computational resources. Notably, scaling effects are more pronounced in lower dimensions, potentially due to the MLP's convergence rate exhibiting a linear dependency on the dimensionality $d$.

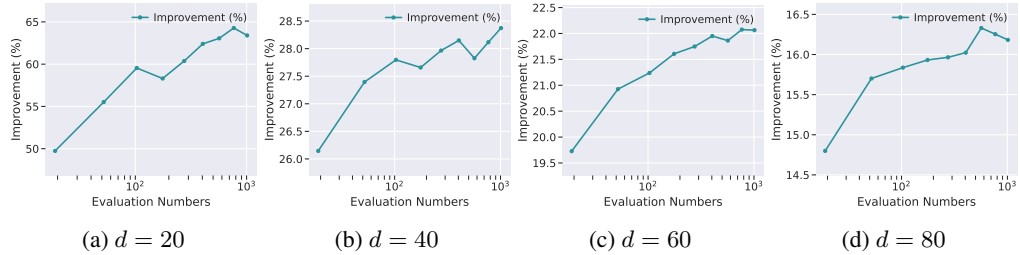

(a) $d = 20$  (b) $d = 40$  (c) $d = 60$  (d) $d = 80$

Figure 11: For the viscous Burgers equation, SCaSML with PINN consistently improves performance as the sample size $M$ increases exponentially.

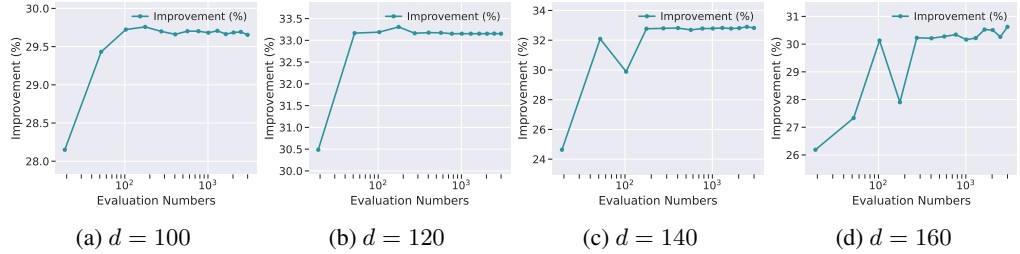

(a) $d = 100$  (b) $d = 120$  (c) $d = 140$  (d) $d = 160$

Figure 12: For the HJB equation, SCaSML with PINN consistently enhances performance with increases in the exponential base of the sample size $M$. However, the scaling curve plateaus at $M = 14$, likely due to the relatively small clipping range of SCaSML compared to the solution magnitude. In general, a larger clipping threshold permits more outliers, thereby requiring additional samples to mitigate variance and ultimately enhancing accuracy; this trade-off must be considered in light of available computational resources.

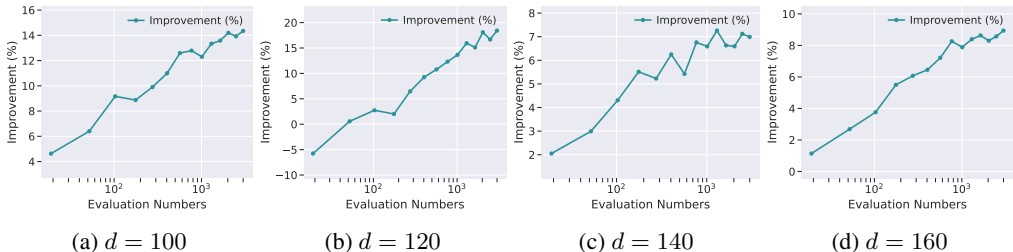

(a) $d = 100$  (b) $d = 120$  (c) $d = 140$  (d) $d = 160$

Figure 13: For the Diffusion Reaction equation, SCaSML with PINN consistently improves performance as the exponential base of the sample size $M$ increases.

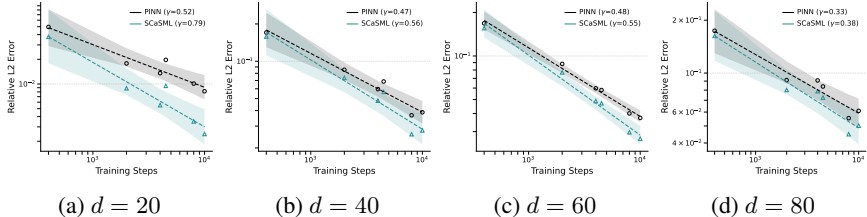

Figure 14: We apply quadrature SCaSML to calibrate a PINN surrogate for the $d$-dimensional viscous Burgers equation. All plots employ logarithmic scales on both axes, and the slope $\gamma$ denotes the polynomial convergence rate. Numerical results demonstrate that, when collocation points for testing and inference are increased simultaneously, SCaSML achieves a faster scaling law than the base surrogate model.

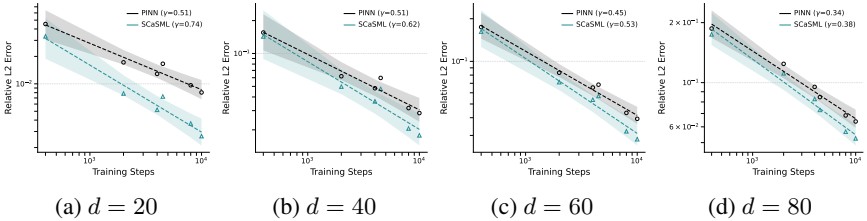

Figure 15: We apply full-history SCaSML to calibrate a PINN surrogate for the $d$-dimensional viscous Burgers equation. Numerical results demonstrate that, when collocation points for testing and inference are increased simultaneously, SCaSML achieves a faster scaling law than the base surrogate model.

### G.4 STATISTICAL ANALYSIS OF ERROR REDUCTION AND CONFIDENCE INTERVALS

In response to reviewer feedback requesting a rigorous statistical validation of our results, we conducted a repeated experiment analysis. Unlike the single-run statistics, this procedure accounts for the randomness inherent in both the training process (e.g., neural network initialization, optimizer noise) and the inference process (Monte Carlo sampling seeds).

#### G.4.1 EXPERIMENTAL DESIGN AND METHODOLOGY

For each problem configuration, we repeated the entire experiment $N_{reps} = 10$ times with different random seeds. In each repetition, we performed the following steps:

1. **Model Training:** A new surrogate model (PINN or GP) was trained from scratch (where applicable). Settings are the same with G.1.

2. **Inference:** The baseline surrogate, the naive MLP solver, and the SCaSML framework were evaluated on a fixed test set of $N_{test} = 1200$ points.

3. **Metric Calculation:** We computed the Mean Relative $L^2$ Error, Mean $L^1$ Error, Mean Squared $L^2$ Error, and for each run.

From these $N_{reps}$ repetitions, we calculated the following statistics:

- **Mean and Standard Deviation:** Computed across the 10 independent runs.
- **95% Confidence Interval (CI):** Calculated for the mean metric as $[a, b] = [\mu - 1.96\frac{\sigma}{\sqrt{N_{reps}}}, \mu + 1.96\frac{\sigma}{\sqrt{N_{reps}}}]$.
- **Paired t-test:** We performed paired t-tests to compare the error distributions of SCaSML against the baselines (GP/PINN and MLP) across the repetitions. The null hypothesis is that the mean difference in error is zero.

The tables below present the full results. Note that while SCaSML requires more execution time (as expected for inference-time scaling), it achieves statistically significant error reductions ($p \ll 0.001$) across all accuracy metrics and dimensions.

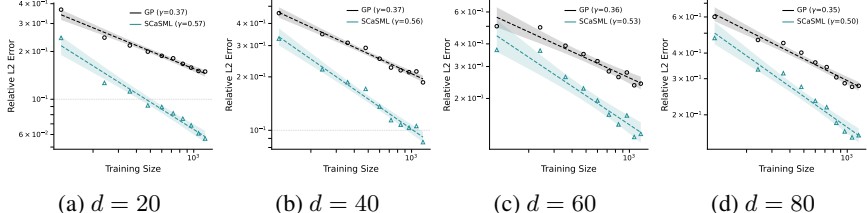

(a) $d = 20$  (b) $d = 40$  (c) $d = 60$  (d) $d = 80$

Figure 16: We apply quadrature SCaSML to calibrate a Gaussian Process surrogate for the $d$-dimensional viscous Burgers equation. Numerical results demonstrate that, when collocation points for testing and inference are increased simultaneously, SCaSML achieves a faster scaling law than the base surrogate model.

Table 2: Statistical analysis for Viscous Burgers (VB) with **GP Surrogate** (10 repetitions). Comparisons are pairwise against SCaSML.

| Metric | Method | **20d** Mean ± Std | 95% CI | Stat (vs SCaSML) | **40d** Mean ± Std | 95% CI | Stat (vs SCaSML) |
|---|---|---|---|---|---|---|---|
| Rel $L^2$ | GP | 1.46e-1 ± 2.8e-3 | [1.43e-1, 1.50e-1] | t=158, p=8e-17 | 1.84e-1 ± 4.2e-3 | [1.76e-1, 1.91e-1] | t=127, p=6e-16 |
| | MLP | 1.84e-1 ± 3.4e-3 | [1.77e-1, 1.89e-1] | t=80.0, p=4e-14 | 2.27e-1 ± 6.4e-3 | [2.17e-1, 2.38e-1] | t=73.1, p=9e-14 |
| | SCaSML | 6.16e-2 ± 2.1e-3 | [5.80e-2, 6.50e-2] | – | 8.91e-2 ± 3.1e-3 | [8.55e-2, 9.60e-2] | – |
| $L^1$ | GP | 6.97e-2 ± 1.4e-3 | [6.80e-2, 7.20e-2] | t=131, p=4e-16 | 9.42e-2 ± 1.3e-3 | [9.28e-2, 9.66e-2] | t=129, p=5e-16 |
| | MLP | 7.62e-2 ± 1.8e-3 | [7.34e-2, 7.94e-2] | t=81.7, p=3e-14 | 9.35e-2 ± 1.8e-3 | [9.10e-2, 9.79e-2] | t=80.6, p=4e-14 |
| | SCaSML | 2.49e-2 ± 6.8e-4 | [2.39e-2, 2.63e-2] | – | 4.01e-2 ± 1.0e-3 | [3.83e-2, 4.20e-2] | – |
| $L^2$ (sq) | GP | 7.63e-3 ± 3.1e-4 | [7.26e-3, 8.18e-3] | t=79.8, p=4e-14 | 1.29e-2 ± 4.3e-4 | [1.25e-2, 1.39e-2] | t=110, p=2e-15 |
| | MLP | 1.22e-2 ± 5.3e-4 | [1.13e-2, 1.28e-2] | t=59.3, p=6e-13 | 1.95e-2 ± 1.2e-3 | [1.80e-2, 2.19e-2] | t=46.9, p=5e-12 |
| | SCaSML | 1.37e-3 ± 9.1e-5 | [1.25e-3, 1.53e-3] | – | 3.02e-3 ± 2.0e-4 | [2.75e-3, 3.50e-3] | – |

| Metric | Method | **60d** Mean ± Std | 95% CI | Stat (vs SCaSML) | **80d** Mean ± Std | 95% CI | Stat (vs SCaSML) |
|---|---|---|---|---|---|---|---|
| Rel $L^2$ | GP | 2.34e-1 ± 6.2e-3 | [2.22e-1, 2.42e-1] | t=174, p=4e-17 | 2.67e-1 ± 4.6e-3 | [2.58e-1, 2.72e-1] | t=141, p=2e-16 |
| | MLP | 2.52e-1 ± 8.5e-3 | [2.40e-1, 2.64e-1] | t=36.9, p=4e-11 | 2.75e-1 ± 8.2e-3 | [2.62e-1, 2.89e-1] | t=43.7, p=9e-12 |
| | SCaSML | 1.23e-1 ± 4.7e-3 | [1.14e-1, 1.28e-1] | – | 1.53e-1 ± 3.0e-3 | [1.48e-1, 1.57e-1] | – |
| $L^1$ | GP | 1.26e-1 ± 2.0e-3 | [1.23e-1, 1.30e-1] | t=222, p=4e-18 | 1.49e-1 ± 2.6e-3 | [1.46e-1, 1.54e-1] | t=134, p=4e-16 |
| | MLP | 1.01e-1 ± 3.9e-3 | [9.43e-2, 1.07e-1] | t=26.3, p=8e-10 | 1.09e-1 ± 2.9e-3 | [1.04e-1, 1.13e-1] | t=35.6, p=5e-11 |
| | SCaSML | 5.98e-2 ± 1.7e-3 | [5.68e-2, 6.20e-2] | – | 7.90e-2 ± 1.6e-3 | [7.65e-2, 8.25e-2] | – |
| $L^2$ (sq) | GP | 2.15e-2 ± 6.8e-4 | [2.05e-2, 2.26e-2] | t=124, p=8e-16 | 2.89e-2 ± 8.1e-4 | [2.80e-2, 3.05e-2] | t=118, p=1e-15 |
| | MLP | 2.50e-2 ± 2.1e-3 | [2.23e-2, 2.84e-2] | t=27.1, p=6e-10 | 3.07e-2 ± 2.2e-3 | [2.73e-2, 3.36e-2] | t=31.8, p=1e-10 |
| | SCaSML | 6.00e-3 ± 3.2e-4 | [5.40e-3, 6.41e-3] | – | 9.49e-3 ± 3.7e-4 | [8.98e-3, 1.03e-2] | – |

## G.5 Relative $L^2$ Error Improvement

In this section, we provide supplementary plots that visualize the relative improvement in $L^2$ error achieved by SCaSML over the baseline surrogate model (PINN or GP). The percentage improvement is calculated as:

$$\text{Improvement \%} = \left( \frac{\|\text{Error}_{\text{Surrogate}}\|_{L^2} - \|\text{Error}_{\text{SCaSML}}\|_{L^2}}{\|\text{Error}_{\text{Surrogate}}\|_{L^2}} \right) \times 100$$

(a) $d = 20$  (b) $d = 40$  (c) $d = 60$  (d) $d = 80$

Figure 17: We apply full-history SCaSML to calibrate a Gaussian Process surrogate for the $d$-dimensional viscous Burgers equation. Numerical results demonstrate that, when collocation points for testing and inference are increased simultaneously, SCaSML achieves a faster scaling law than the base surrogate model.

Table 3: Statistical analysis for Viscous Burgers (VB) with **PINN Surrogate** (10 repetitions).

| Metric | Method | 20d Mean ± Std | 20d 95% CI | 20d Stat (vs SCaSML) | 40d Mean ± Std | 40d 95% CI | 40d Stat (vs SCaSML) |
|---|---|---|---|---|---|---|---|
| Rel $L^2$ | PINN | 1.25e-2 ± 3.9e-4 | [1.19e-2, 1.31e-2] | t=172, p=4e-17 | 4.51e-2 ± 1.3e-3 | [4.23e-2, 4.67e-2] | t=135, p=3e-16 |
| | MLP | 8.34e-2 ± 2.2e-3 | [8.08e-2, 8.78e-2] | t=108, p=3e-15 | 1.06e-1 ± 2.2e-3 | [1.03e-1, 1.10e-1] | t=111, p=2e-15 |
| | SCaSML | 4.37e-3 ± 2.9e-4 | [3.90e-3, 4.78e-3] | – | 3.38e-2 ± 1.1e-3 | [3.16e-2, 3.53e-2] | – |
| $L^1$ | PINN | 5.83e-3 ± 1.8e-4 | [5.48e-3, 6.11e-3] | t=84.5, p=2e-14 | 2.16e-2 ± 4.5e-4 | [2.08e-2, 2.22e-2] | t=77.7, p=5e-14 |
| | MLP | 3.43e-2 ± 1.1e-3 | [3.29e-2, 3.61e-2] | t=93.5, p=9e-15 | 4.38e-2 ± 9.8e-4 | [4.27e-2, 4.55e-2] | t=104, p=3e-15 |
| | SCaSML | 1.44e-3 ± 4.8e-5 | [1.38e-3, 1.55e-3] | – | 1.41e-2 ± 3.5e-4 | [1.36e-2, 1.45e-2] | – |
| $L^2$ (sq) | PINN | 5.63e-5 ± 3.6e-6 | [5.03e-5, 6.23e-5] | t=57.4, p=7e-13 | 7.72e-4 ± 3.3e-5 | [7.19e-4, 8.17e-4] | t=96.6, p=7e-15 |
| | MLP | 2.50e-3 ± 0.0 | [2.30e-3, 2.82e-3] | t=48.0, p=4e-12 | 4.29e-3 ± 2.4e-4 | [3.91e-3, 4.72e-3] | t=51.6, p=2e-12 |
| | SCaSML | 6.90e-6 ± 9.4e-7 | [5.55e-6, 8.30e-6] | – | 4.34e-4 ± 2.2e-5 | [4.03e-4, 4.66e-4] | – |

| Metric | Method | 60d Mean ± Std | 60d 95% CI | 60d Stat (vs SCaSML) | 80d Mean ± Std | 80d 95% CI | 80d Stat (vs SCaSML) |
|---|---|---|---|---|---|---|---|
| Rel $L^2$ | PINN | 4.62e-2 ± 1.1e-3 | [4.43e-2, 4.75e-2] | t=203, p=9e-18 | 6.59e-2 ± 2.0e-3 | [6.17e-2, 6.95e-2] | t=168, p=5e-17 |
| | MLP | 1.17e-1 ± 3.2e-3 | [1.14e-1, 1.23e-1] | t=75.0, p=7e-14 | 1.22e-1 ± 3.7e-3 | [1.15e-1, 1.28e-1] | t=44.8, p=7e-12 |
| | SCaSML | 3.53e-2 ± 9.8e-4 | [3.36e-2, 3.65e-2] | – | 5.51e-2 ± 1.9e-3 | [5.13e-2, 5.90e-2] | – |
| $L^1$ | PINN | 2.24e-2 ± 4.4e-4 | [2.14e-2, 2.29e-2] | t=134, p=4e-16 | 3.20e-2 ± 6.6e-4 | [3.10e-2, 3.31e-2] | t=182, p=2e-17 |
| | MLP | 4.79e-2 ± 1.7e-3 | [4.55e-2, 5.04e-2] | t=56.8, p=8e-13 | 4.94e-2 ± 1.2e-3 | [4.74e-2, 5.12e-2] | t=50.0, p=3e-12 |
| | SCaSML | 1.50e-2 ± 3.4e-4 | [1.41e-2, 1.55e-2] | – | 2.46e-2 ± 6.9e-4 | [2.34e-2, 2.58e-2] | – |
| $L^2$ (sq) | PINN | 8.40e-4 ± 2.9e-5 | [7.72e-4, 8.71e-4] | t=128, p=5e-16 | 1.76e-3 ± 8.6e-5 | [1.64e-3, 1.95e-3] | t=136, p=3e-16 |
| | MLP | 5.44e-3 ± 3.5e-4 | [4.99e-3, 6.09e-3] | t=44.0, p=8e-12 | 6.08e-3 ± 3.5e-4 | [5.37e-3, 6.55e-3] | t=36.9, p=4e-11 |
| | SCaSML | 4.92e-4 ± 2.2e-5 | [4.43e-4, 5.18e-4] | – | 1.23e-3 ± 7.5e-5 | [1.13e-3, 1.41e-3] | – |

Table 4: Statistical analysis for Linear Convection-Diffusion (LCD) (10 repetitions).

| Metric | Method | 10d Mean ± Std | 10d 95% CI | 10d Stat (vs SCaSML) | 20d Mean ± Std | 20d 95% CI | 20d Stat (vs SCaSML) |
|---|---|---|---|---|---|---|---|
| Rel $L^2$ | PINN | 4.85e-2 ± 2.2e-3 | [4.43e-2, 5.23e-2] | t=34.1, p=8e-11 | 8.60e-2 ± 3.6e-3 | [7.80e-2, 9.03e-2] | t=39.2, p=2e-11 |
| | MLP | 2.30e-1 ± 5.5e-3 | [2.20e-1, 2.36e-1] | t=134, p=4e-16 | 2.41e-1 ± 5.5e-3 | [2.32e-1, 2.48e-1] | t=129, p=5e-16 |
| | SCaSML | 2.77e-2 ± 8.6e-4 | [2.60e-2, 2.90e-2] | – | 5.07e-2 ± 1.8e-3 | [4.84e-2, 5.37e-2] | – |
| $L^1$ | PINN | 3.01e-2 ± 1.1e-3 | [2.83e-2, 3.23e-2] | t=35.0, p=6e-11 | 8.71e-2 ± 2.2e-3 | [8.24e-2, 8.95e-2] | t=71.5, p=1e-13 |
| | MLP | 1.68e-1 ± 1.7e-3 | [1.65e-1, 1.70e-1] | t=285, p=4e-19 | 2.38e-1 ± 3.0e-3 | [2.31e-1, 2.42e-1] | t=241, p=2e-18 |
| | SCaSML | 1.77e-2 ± 4.3e-4 | [1.71e-2, 1.84e-2] | – | 4.67e-2 ± 1.3e-3 | [4.50e-2, 4.83e-2] | – |
| $L^2$ (sq) | PINN | 2.20e-3 ± 1.8e-4 | [1.98e-3, 2.58e-3] | t=26.7, p=7e-10 | 1.29e-2 ± 7.9e-4 | [1.15e-2, 1.41e-2] | t=36.7, p=4e-11 |
| | MLP | 4.96e-2 ± 1.0e-3 | [4.85e-2, 5.16e-2] | t=154, p=1e-16 | 1.02e-1 ± 3.1e-3 | [9.47e-2, 1.05e-1] | t=102, p=4e-15 |
| | SCaSML | 7.20e-4 ± 2.3e-5 | [6.82e-4, 7.54e-4] | – | 4.50e-3 ± 2.8e-4 | [4.05e-3, 4.91e-3] | – |

| Metric | Method | 30d Mean ± Std | 30d 95% CI | 30d Stat (vs SCaSML) | 60d Mean ± Std | 60d 95% CI | 60d Stat (vs SCaSML) |
|---|---|---|---|---|---|---|---|
| Rel $L^2$ | PINN | 1.57e-1 ± 9.5e-3 | [1.43e-1, 1.74e-1] | t=19.7, p=1e-08 | 2.85e-1 ± 8.7e-3 | [2.74e-1, 2.96e-1] | t=55.2, p=1e-12 |
| | MLP | 2.42e-1 ± 8.1e-3 | [2.29e-1, 2.54e-1] | t=83.3, p=3e-14 | 2.46e-1 ± 4.8e-3 | [2.38e-1, 2.52e-1] | t=106, p=3e-15 |
| | SCaSML | 9.69e-2 ± 4.6e-3 | [9.12e-2, 1.02e-1] | – | 1.25e-1 ± 2.8e-3 | [1.21e-1, 1.30e-1] | – |
| $L^1$ | PINN | 1.81e-1 ± 6.0e-3 | [1.72e-1, 1.89e-1] | t=37.0, p=4e-11 | 4.90e-1 ± 8.2e-3 | [4.77e-1, 4.99e-1] | t=83.4, p=3e-14 |
| | MLP | 2.89e-1 ± 4.6e-3 | [2.82e-1, 2.95e-1] | t=316, p=2e-19 | 4.18e-1 ± 4.5e-3 | [4.11e-1, 4.24e-1] | t=219, p=4e-18 |
| | SCaSML | 1.05e-1 ± 3.8e-3 | [1.01e-1, 1.11e-1] | – | 1.96e-1 ± 4.7e-3 | [1.86e-1, 2.02e-1] | – |
| $L^2$ (sq) | PINN | 6.43e-2 ± 9.2e-3 | [5.50e-2, 8.25e-2] | t=14.4, p=2e-07 | 4.13e-1 ± 1.7e-2 | [3.87e-1, 4.39e-1] | t=55.7, p=1e-12 |
| | MLP | 1.52e-1 ± 6.3e-3 | [1.43e-1, 1.62e-1] | t=81.8, p=3e-14 | 3.06e-1 ± 7.7e-3 | [2.96e-1, 3.20e-1] | t=134, p=4e-16 |
| | SCaSML | 2.44e-2 ± 2.1e-3 | [2.20e-2, 2.80e-2] | – | 7.97e-2 ± 3.9e-3 | [7.36e-2, 8.54e-2] | – |

These plots directly visualize the 20-80% error reduction claimed in the main text and demonstrate the effectiveness of our correction framework across all test cases and dimensions. The experimental settings are identical to those used for violin plots in Appendix G.1.

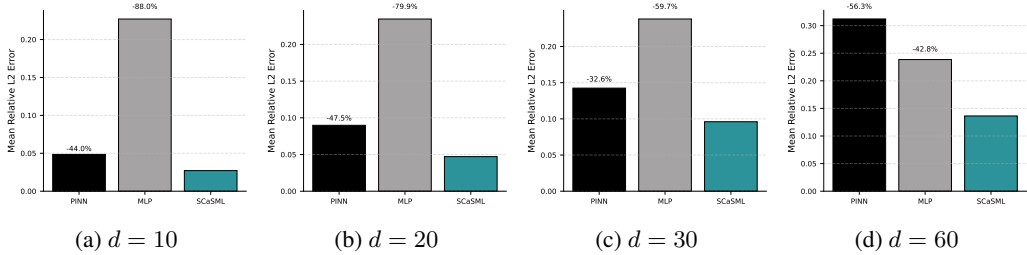

(a) $d = 10$  (b) $d = 20$  (c) $d = 30$  (d) $d = 60$

Figure 18: Relative $L^2$ error improvement (%) of SCaSML (full-history) over the baseline PINN surrogate on the linear convection-diffusion equation for $d = 10, 20, 30, 60$.

Table 5: Statistical analysis for Hamilton-Jacobi-Bellman (LQG) (10 repetitions).

| Metric | Method | 100d Mean ± Std | 95% CI | Stat (vs SCaSML) | 120d Mean ± Std | 95% CI | Stat (vs SCaSML) |
|---|---|---|---|---|---|---|---|
| Rel $L^2$ | PINN | 9.05e-2 ± 2.5e-3 | [8.57e-2, 9.49e-2] | t=6.5, p=1e-04 | 9.13e-2 ± 2.0e-3 | [8.77e-2, 9.43e-2] | t=25.9, p=9e-10 |
| | MLP | 5.69e+0 ± 2.0e-2 | [5.66e+0, 5.73e+0] | t=927, p=1e-23 | 5.50e+0 ± 1.7e-2 | [5.48e+0, 5.53e+0] | t=1073, p=3e-24 |
| | SCaSML | 7.27e-2 ± 9.4e-3 | [6.48e-2, 9.85e-2] | – | 6.42e-2 ± 1.7e-3 | [6.17e-2, 6.68e-2] | – |
| $L^1$ | PINN | 1.50e-1 ± 3.0e-3 | [1.44e-1, 1.55e-1] | t=7.2, p=5e-05 | 1.68e-1 ± 3.3e-3 | [1.62e-1, 1.74e-1] | t=146, p=2e-16 |
| | MLP | 1.21e+1 ± 1.3e-2 | [1.21e+1, 1.21e+1] | t=1599, p=7e-26 | 1.22e+1 ± 1.4e-2 | [1.21e+1, 1.22e+1] | t=2531, p=1e-27 |
| | SCaSML | 1.09e-1 ± 1.8e-2 | [9.68e-2, 1.59e-1] | – | 1.02e-1 ± 2.3e-3 | [9.81e-2, 1.05e-1] | – |
| $L^2$ (sq) | PINN | 3.70e-2 ± 1.8e-3 | [3.34e-2, 4.00e-2] | t=6.2, p=2e-04 | 4.09e-2 ± 1.8e-3 | [3.79e-2, 4.39e-2] | t=75.8, p=6e-14 |
| | MLP | 1.46e+2 ± 2.8e-1 | [1.46e+2, 1.47e+2] | t=1623, p=7e-26 | 1.48e+2 ± 3.0e-1 | [1.48e+2, 1.49e+2] | t=1576, p=8e-26 |
| | SCaSML | 2.42e-2 ± 7.0e-3 | [1.91e-2, 4.37e-2] | – | 2.02e-2 ± 1.1e-3 | [1.88e-2, 2.20e-2] | – |

| Metric | Method | 140d Mean ± Std | 95% CI | Stat (vs SCaSML) | 160d Mean ± Std | 95% CI | Stat (vs SCaSML) |
|---|---|---|---|---|---|---|---|
| Rel $L^2$ | PINN | 1.03e-1 ± 1.8e-3 | [1.01e-1, 1.06e-1] | t=3.7, p=5e-03 | 1.10e-1 ± 2.4e-3 | [1.06e-1, 1.14e-1] | t=21.7, p=4e-09 |
| | MLP | 5.36e+0 ± 2.0e-2 | [5.34e+0, 5.40e+0] | t=823, p=3e-23 | 5.26e+0 ± 1.9e-2 | [5.24e+0, 5.29e+0] | t=870, p=2e-23 |
| | SCaSML | 8.52e-2 ± 1.5e-2 | [7.45e-2, 1.15e-1] | – | 8.48e-2 ± 5.0e-3 | [7.88e-2, 9.49e-2] | – |
| $L^1$ | PINN | 1.92e-1 ± 3.5e-3 | [1.87e-1, 1.99e-1] | t=4.3, p=2e-03 | 2.11e-1 ± 3.3e-3 | [2.05e-1, 2.17e-1] | t=26.0, p=9e-10 |
| | MLP | 1.23e+1 ± 1.4e-2 | [1.22e+1, 1.23e+1] | t=866, p=2e-23 | 1.23e+1 ± 1.3e-2 | [1.23e+1, 1.23e+1] | t=2170, p=5e-27 |
| | SCaSML | 1.41e-1 ± 3.8e-2 | [1.16e-1, 2.14e-1] | – | 1.43e-1 ± 1.0e-2 | [1.32e-1, 1.61e-1] | – |
| $L^2$ (sq) | PINN | 5.55e-2 ± 1.8e-3 | [5.35e-2, 5.89e-2] | t=3.4, p=8e-03 | 6.60e-2 ± 2.5e-3 | [6.15e-2, 7.04e-2] | t=25.3, p=1e-09 |
| | MLP | 1.50e+2 ± 3.1e-1 | [1.50e+2, 1.51e+2] | t=1485, p=1e-25 | 1.52e+2 ± 3.0e-1 | [1.51e+2, 1.52e+2] | t=1583, p=8e-26 |
| | SCaSML | 3.90e-2 ± 1.5e-2 | [2.90e-2, 6.74e-2] | – | 3.95e-2 ± 4.7e-3 | [3.43e-2, 4.87e-2] | – |

Table 6: Statistical analysis for Diffusion-Reaction (DR) (10 repetitions).

| Metric | Method | 100d Mean ± Std | 95% CI | Stat (vs SCaSML) | 120d Mean ± Std | 95% CI | Stat (vs SCaSML) |
|---|---|---|---|---|---|---|---|
| Rel $L^2$ | PINN | 9.83e-3 ± 2.6e-4 | [9.45e-3, 1.02e-2] | t=14.3, p=2e-07 | 1.10e-2 ± 3.0e-4 | [1.04e-2, 1.13e-2] | t=25.9, p=9e-10 |
| | MLP | 8.62e-2 ± 2.6e-3 | [8.20e-2, 9.05e-2] | t=92.6, p=1e-14 | 9.01e-2 ± 1.2e-3 | [8.78e-2, 9.17e-2] | t=249, p=1e-18 |
| | SCaSML | 9.19e-3 ± 2.8e-4 | [8.63e-3, 9.61e-3] | – | 1.00e-2 ± 2.8e-4 | [9.61e-3, 1.03e-2] | – |
| $L^1$ | PINN | 1.20e-2 ± 3.2e-4 | [1.14e-2, 1.24e-2] | t=14.1, p=2e-07 | 1.36e-2 ± 4.0e-4 | [1.29e-2, 1.41e-2] | t=16.7, p=4e-08 |
| | MLP | 9.37e-2 ± 2.6e-3 | [8.97e-2, 9.77e-2] | t=102, p=4e-15 | 9.77e-2 ± 1.9e-3 | [9.51e-2, 1.01e-1] | t=142, p=2e-16 |
| | SCaSML | 1.14e-2 ± 3.4e-4 | [1.07e-2, 1.19e-2] | – | 1.24e-2 ± 2.8e-4 | [1.20e-2, 1.28e-2] | – |
| $L^2$ (sq) | PINN | 2.48e-4 ± 1.3e-5 | [2.29e-4, 2.68e-4] | t=14.2, p=2e-07 | 3.09e-4 ± 1.7e-5 | [2.79e-4, 3.29e-4] | t=24.3, p=2e-09 |
| | MLP | 1.91e-2 ± 1.1e-3 | [1.73e-2, 2.08e-2] | t=53.1, p=1e-12 | 2.09e-2 ± 4.9e-4 | [1.98e-2, 2.15e-2] | t=132, p=4e-16 |
| | SCaSML | 2.17e-4 ± 1.3e-5 | [1.91e-4, 2.35e-4] | – | 2.57e-4 ± 1.4e-5 | [2.38e-4, 2.72e-4] | – |

| Metric | Method | 140d Mean ± Std | 95% CI | Stat (vs SCaSML) | 160d Mean ± Std | 95% CI | Stat (vs SCaSML) |
|---|---|---|---|---|---|---|---|
| Rel $L^2$ | PINN | 3.23e-2 ± 5.4e-4 | [3.14e-2, 3.34e-2] | t=47.4, p=4e-12 | 3.59e-2 ± 8.1e-4 | [3.47e-2, 3.71e-2] | t=72.4, p=9e-14 |
| | MLP | 8.96e-2 ± 2.3e-3 | [8.69e-2, 9.47e-2] | t=80.6, p=4e-14 | 8.74e-2 ± 2.5e-3 | [8.28e-2, 9.17e-2] | t=62.1, p=4e-13 |
| | SCaSML | 3.00e-2 ± 4.8e-4 | [2.92e-2, 3.09e-2] | – | 3.37e-2 ± 8.2e-4 | [3.24e-2, 3.48e-2] | – |
| $L^1$ | PINN | 4.05e-2 ± 8.0e-4 | [3.91e-2, 4.21e-2] | t=33.5, p=9e-11 | 4.50e-2 ± 9.5e-4 | [4.36e-2, 4.67e-2] | t=35.5, p=5e-11 |
| | MLP | 9.89e-2 ± 2.3e-3 | [9.53e-2, 1.04e-1] | t=84.2, p=2e-14 | 9.61e-2 ± 2.3e-3 | [9.34e-2, 1.01e-1] | t=62.9, p=3e-13 |
| | SCaSML | 3.77e-2 ± 6.5e-4 | [3.64e-2, 3.88e-2] | – | 4.22e-2 ± 9.9e-4 | [4.06e-2, 4.39e-2] | – |
| $L^2$ (sq) | PINN | 2.67e-3 ± 8.7e-5 | [2.53e-3, 2.85e-3] | t=41.8, p=1e-11 | 3.31e-3 ± 1.5e-4 | [3.10e-3, 3.53e-3] | t=68.9, p=1e-13 |
| | MLP | 2.06e-2 ± 1.0e-3 | [1.93e-2, 2.30e-2] | t=55.5, p=1e-12 | 1.96e-2 ± 1.1e-3 | [1.77e-2, 2.15e-2] | t=46.9, p=5e-12 |
| | SCaSML | 2.31e-3 ± 7.0e-5 | [2.19e-3, 2.44e-3] | – | 2.91e-3 ± 1.4e-4 | [2.70e-3, 3.10e-3] | – |

## G.6 POINTWISE ERROR REDUCTION ANALYSIS

To further investigate the robustness of our method, we present scatter plots visualizing the pointwise error difference between the baseline methods (Surrogate and Naive MLP) and our proposed SCaSML. The settings are still the same with Appendix G.1.

For a given test point $x$, we calculate the difference in absolute error:

$$\Delta \text{Error}(x) = |\text{Error}_{\text{Baseline}}(x)| - |\text{Error}_{\text{SCaSML}}(x)|$$

In the following figures:

- Red points ($\Delta\text{Error} > 0$) indicate locations where SCaSML has lower error than the baseline.
- Blue points ($\Delta\text{Error} < 0$) indicate locations where SCaSML has higher error.

We provide comparisons for both baselines: Surrogate vs. SCaSML (showing the correction of the initial model) and Naive MLP vs. SCaSML (showing the benefit of using the surrogate as a control

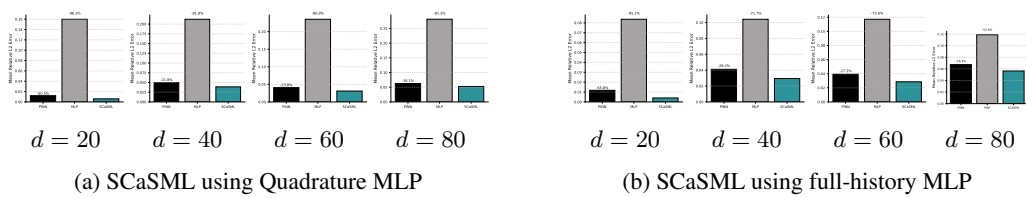

Figure 19: Relative $L^2$ error improvement (%) of SCaSML over the baseline PINN surrogate on the viscous Burgers' equation for $d = 20, 40, 60, 80$.

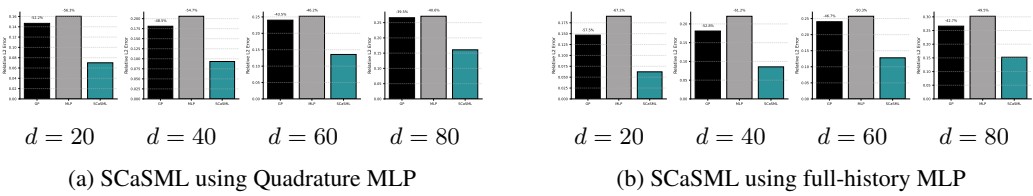

Figure 20: Relative $L^2$ error improvement (%) of SCaSML over the baseline Gaussian Process surrogate on the viscous Burgers' equation for $d = 20, 40, 60, 80$.

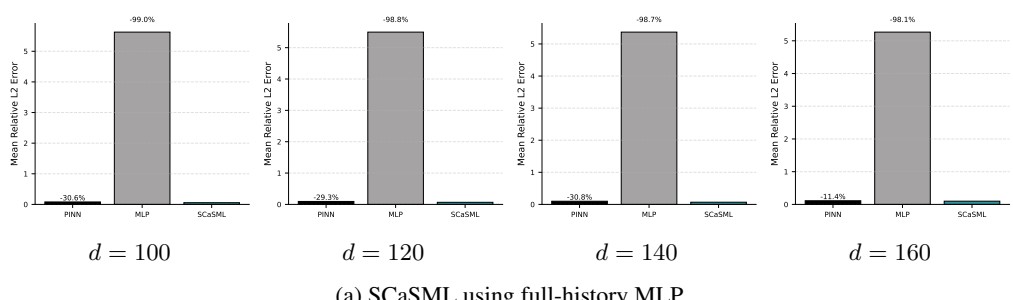

Figure 21: Relative $L^2$ error improvement (%) of SCaSML (full-history) over the baseline PINN surrogate on the LQG control problem for $d = 100, 120, 140, 160$.

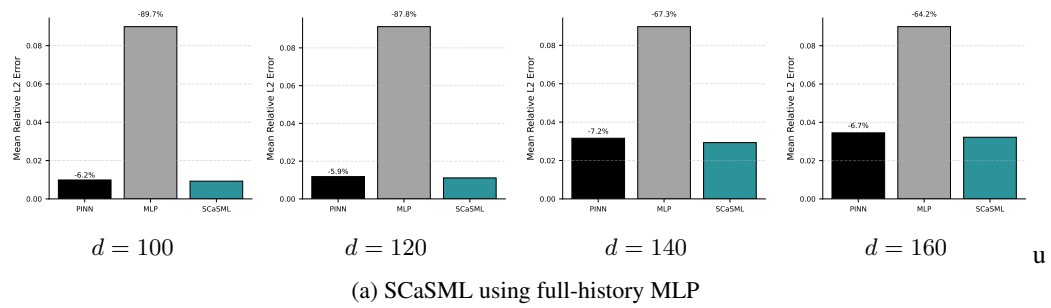

Figure 22: Relative $L^2$ error improvement (%) of SCaSML (full-history) over the baseline PINN surrogate on the diffusion reaction equation for $d = 100, 120, 140, 160$.

variate). Across all experiments, the dominance of red points confirms that SCaSML systematically improves accuracy locally across the high-dimensional domain.

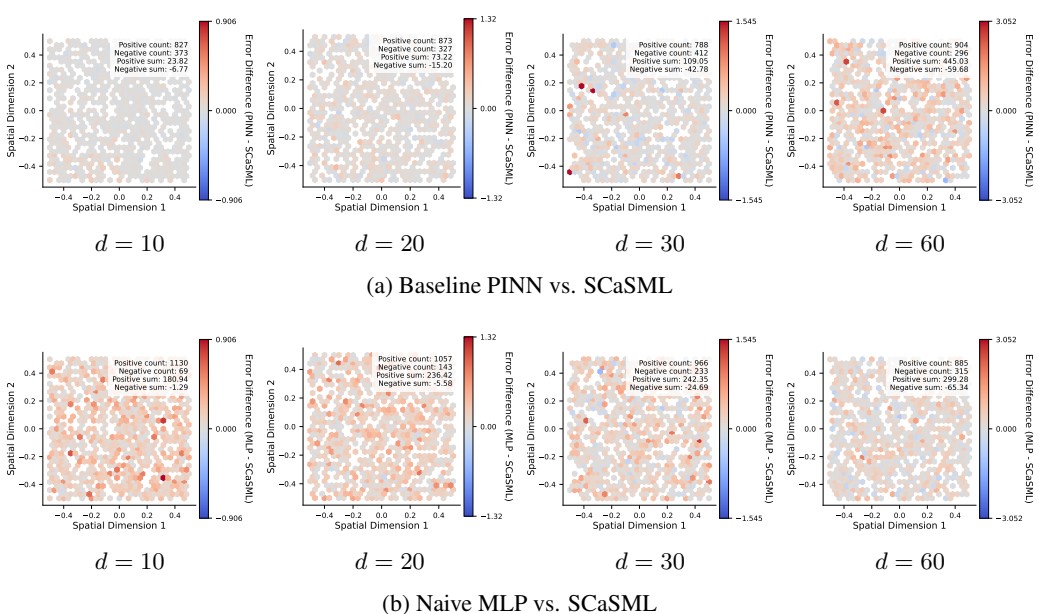

Figure 23: Pointwise error differences for the Linear Convection-Diffusion equation. SCaSML outperforms both the pre-trained PINN and the naive MLP solver across all dimensions.

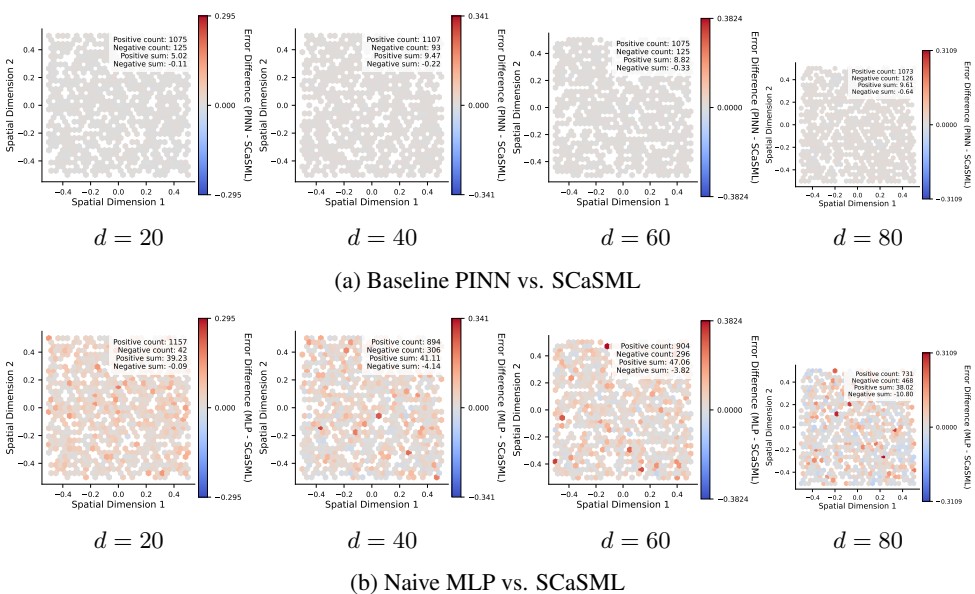

Figure 24: Pointwise error differences for the Viscous Burgers' equation (PINN Surrogate). We observe that SCaSML corrects the PINN's error (top) and significantly outperforms the standalone MLP (bottom).

## G.7 PERFORMANCE COMPARISON UNDER FIXED COMPUTATIONAL BUDGETS

A central question regarding inference-time scaling is whether the performance gain is simply a result of increased wall-clock time, or if the SCaSML framework utilizes computational resources more efficiently than standard training. To address this, we conducted a Fixed Computational Budget analysis.

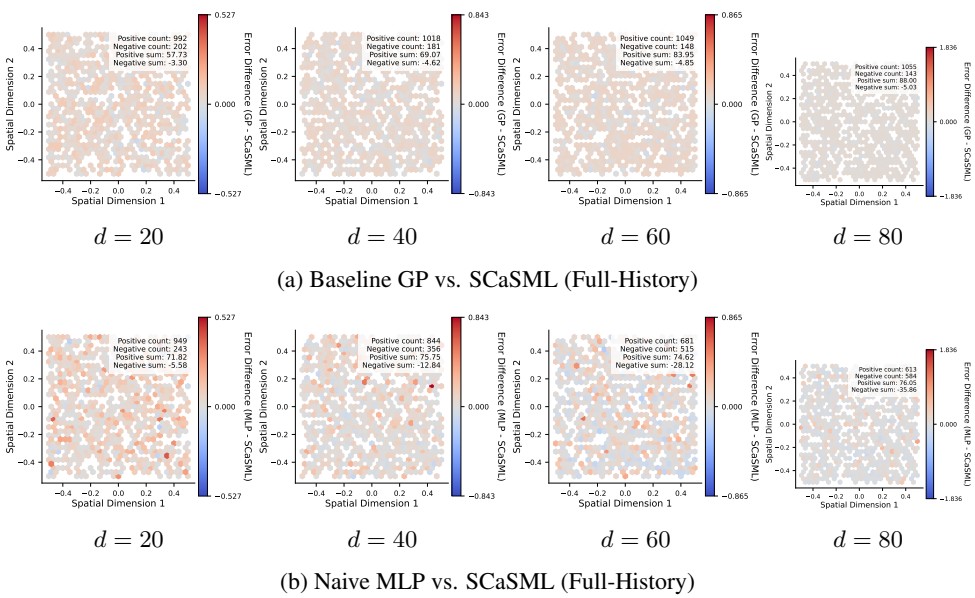

(a) Baseline GP vs. SCaSML (Full-History)

(b) Naive MLP vs. SCaSML (Full-History)

Figure 25: Pointwise error differences for the Viscous Burgers' equation (Gaussian Process Surrogate).

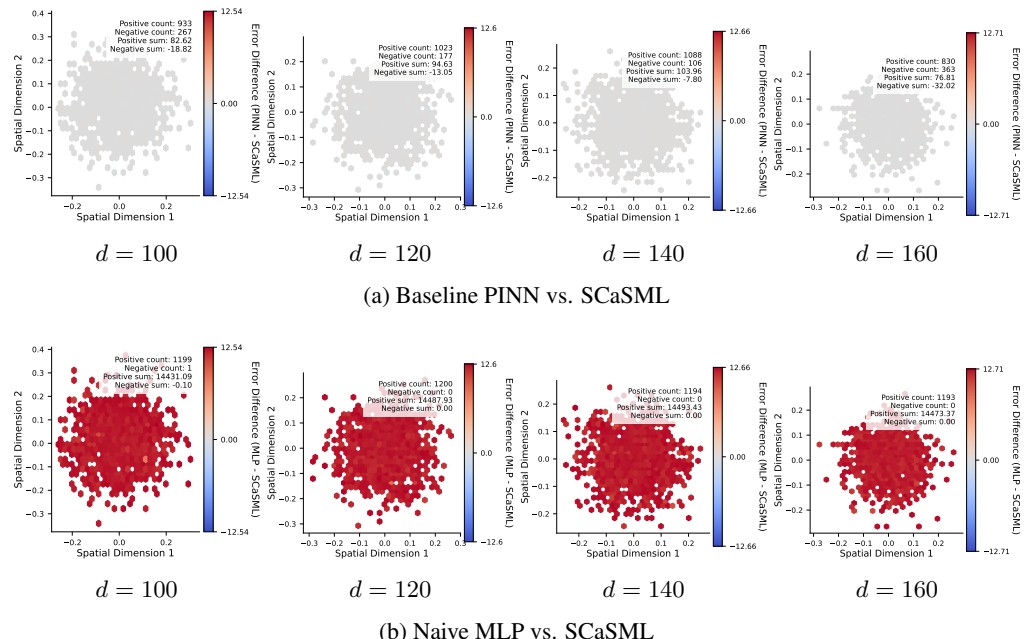

(a) Baseline PINN vs. SCaSML

(b) Naive MLP vs. SCaSML

Figure 26: Pointwise error differences for the LQG control problem. The contrast in the bottom row highlights that the naive MLP fails in high dimensions, whereas SCaSML (stabilized by the surrogate) performs well.

We define a "unit budget" based on a baseline number of training iterations (e.g., 2,000 iterations for a PINN), other settings are still the same with G.1. We then scale this budget by factors of $\times 1, \times 2, \dots, \times 16$. For each budget level, we compare three allocation strategies:

1. **Pure Training (Baseline PINN):** The entire time budget is allocated to training the neural network. A budget of $\times k$ implies training for $k \times N_{base}$ iterations.

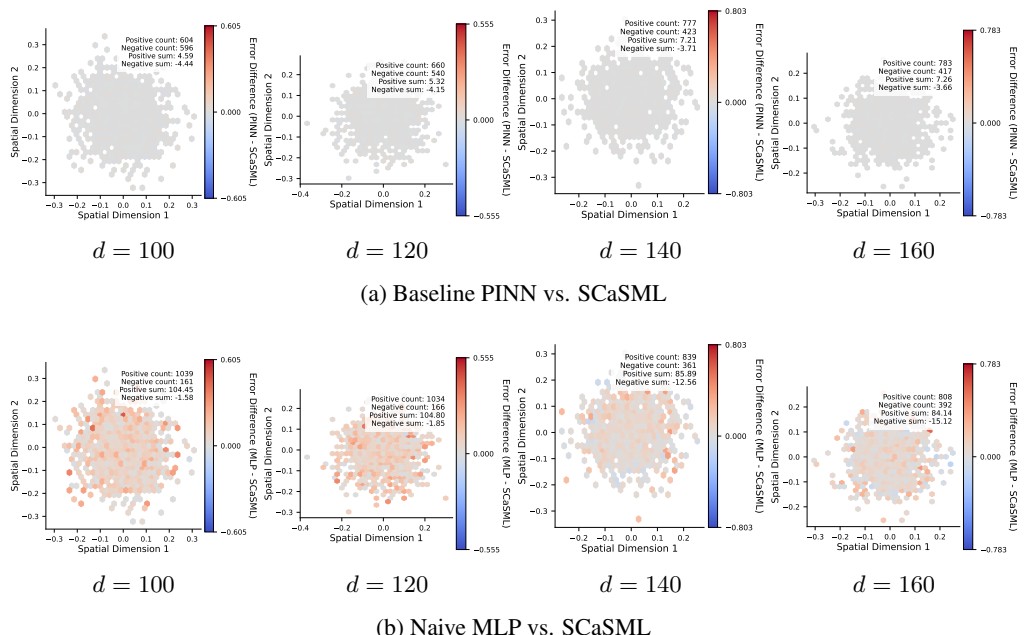

(a) Baseline PINN vs. SCaSML

(b) Naive MLP vs. SCaSML

Figure 27: Pointwise error differences for the Diffusion-Reaction equation.

2. **Pure Simulation (Naive MLP):** The entire time budget is allocated to generating Monte Carlo paths for the MLP solver.

3. **Hybrid Allocation (SCaSML):** This represents our proposed strategy. We allocate a small fraction of the budget (specifically $1/(d+1)$) to training a "weak" surrogate, and allocate the remaining majority of the budget to inference-time correction via the Structural-preserving Law of Defect.

This setup ensures a fair comparison where all methods consume approximately the same total wall-clock time (Training Time + Inference Time). We performed this analysis on the Linear Convection-Diffusion (LCD) equation ($d = 10, 20$) and the Viscous Burgers (VB) equation ($d = 20$) using the full-history SCaSML variant.

The results are visualized in Figure 28.

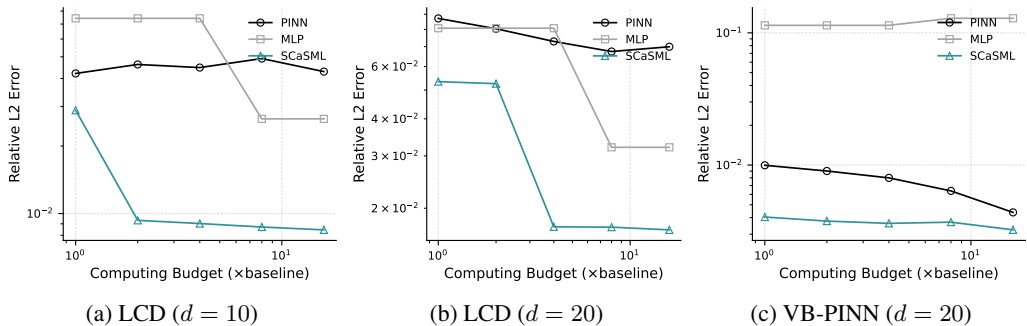

(a) LCD ($d = 10$)                (b) LCD ($d = 20$)                (c) VB-PINN ($d = 20$)

Figure 28: **Error vs. Computational Budget.** The x-axis represents the total computational budget multiplier (log scale), and the y-axis represents the Relative $L^2$ Error (log scale). SCaSML (teal triangles) consistently achieves lower error than the Baseline PINN (black circles) and Naive MLP (gray squares) for the same total cost.

As shown in Figure 28, the SCaSML error curve consistently lies below the PINN training curve. This confirms that allocating marginal compute to inference-time correction yields a higher return on investment (ROI) than allocating it to further training.

Specifically, for the Viscous Burgers equation ($d = 20$), we observe that training the PINN for significantly longer (moving right on the x-axis) results in diminishing returns due to the optimization difficulty of high-frequency error components. In contrast, SCaSML leverages the rigorous convergence rate of the Monte Carlo correction to reduce error rapidly. This empirically validates our theoretical claim that the hybrid ML+MC scaling law ($O(m^{-\gamma-1/2})$) is superior to the pure ML scaling law ($O(m^{-\gamma})$).

### G.8  PERFORMANCE COMPARISON: LARGE PINN VS. SCASML CORRECTION

A critical question in SciML is whether the computational budget is better spent on training a larger, more expressive neural network (increasing model capacity) or on post-hoc inference-time correction (SCaSML). To address this, we conducted a second Fixed Budget analysis where we scaled the **model architecture** while keeping the number of training iterations fixed.

We define a "unit budget" ($B = 1$) corresponding to our standard PINN configuration: a fully connected network with width $W_{base} = 50$ and depth $D_{base} = 5$. As the budget $B$ increases by factors of $\times 1, \times 2, \times 4$, we scale the network architecture to increase its capacity. Specifically, the scaled width $W_B$ and depth $D_B$ are defined as:

$$W_B = \lfloor W_{base} \cdot \sqrt{B} \rfloor, \quad D_B = \max(D_{base}, \lfloor D_{base} + \log_2(B) \rfloor). \tag{121}$$

This scaling strategy ensures that the network's parameter count and computational cost per iteration grow with the budget, allowing us to test the limits of model capacity.

We compare three strategies under these scaling rules using the Linear Convection-Diffusion ($d = 10, 20$) and Viscous Burgers ($d = 20$) equations:

1. **Large PINN (Model Scaling):** We train the scaled network architecture ($W_B, D_B$) for a fixed number of iterations ($N_{iter} = 2000$). The optimizer is Adam with a learning rate of $7 \times 10^{-4}$ and Glorot normal initialization. The increased computational cost arises entirely from the more expensive forward and backward passes of the larger model.

2. **SCaSML (Inference Correction):** We employ the SCaSML framework where the surrogate backbone utilizes the available budget. Crucially, the method allocates resources to the inference-time Monte Carlo correction (using the full-history MLP solver with basis $M = 10$ and levels $N = 2$) rather than relying solely on the surrogate's capacity.

3. **Naive MLP (Pure Simulation):** The entire time budget is allocated to generating Monte Carlo paths for the MLP solver, serving as a pure simulation baseline.

The results (Figure 29) demonstrate that simply increasing the PINN's capacity yields diminishing returns; the model hits a "data efficiency wall" where additional parameters do not translate to proportionally lower errors for high-frequency defects. In contrast, SCaSML consistently achieves lower error for the same total compute time, proving that inference-time correction is a more efficient user of marginal compute than model scaling for these high-dimensional problems.

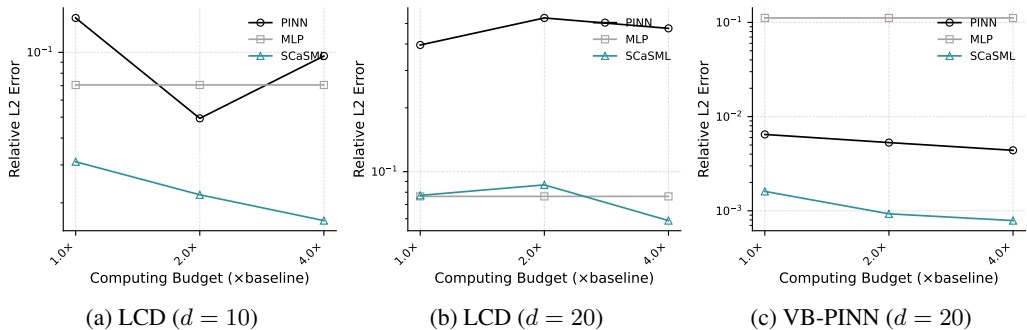

(a) LCD ($d = 10$)  (b) LCD ($d = 20$)  (c) VB-PINN ($d = 20$)

Figure 29: **Large PINN vs. SCaSML.** Comparison of error rates when the computational budget is used to scale up the PINN architecture (black) versus performing inference-time correction (teal).

