# OpenReview forum: "Physics-Informed Inference Time Scaling for Solving High-Dimensional Partial Differential Equations"
_ICLR.cc/2026/Conference — ICLR 2026 Poster_

### Official Review · Reviewer_op81 · 2025-10-24

**Soundness:** 3
**Presentation:** 1
**Contribution:** 2
**Rating:** 4
**Confidence:** 2

**Summary:**

Inference-time scaling is a characteristic phenomenon observed in large language models in which allocating additional computation resources at inference time yields an improvement of output quality of LLMs. The paper is the first attempt to bring the idea into physics-informed emulators to solve high-dimensional PDEs. The key assumption to realize the idea is the semi-linearity of parabolic PDEs, and this makes it possible to derive a PDE, called Law of Defect, that governs the residual between the ground-truth solution and surrogate solution for an original PDE. This finding enables the application of standard solvers such as multilevel Picard solver without requiring any additional retraining and fine-turning, and the residual is also utilized to correct the error of the output of a trained surrogate model, which is the proposed inference pipeline to solve the PDEs. The proposed framework is evaluated on a variety of scenarios with various dimensions that control the difficulty of the task and compared to a couple of standard solvers and plain data-driven surrogate models.

**Strengths:**

- **Intriguing approach:** The proposed approach is disparate from the other approaches boosting the performance in the post-training phase, especially that requires additional training. The question raised could serve as a new class of tasks in the scientific discovery.

- **Performance gain:** The improvement in the prediction accuracy over plain surrogate models is significant and impressive.

**Weaknesses:**

- **Practical implication:** Table1 of the main experiments show that surrogate models already outperform the MLP solver in most of the scenarios. It is not very clear to me what practical scenario necessitates an additional computational budget to boost the performance of the trained data-driven models, especially when the performance of the data-driven model is already superior to that of the solvers.

- **Assumption for experiments:** The hyperparameter configuration for the MLP used in the experiment is not stated exhaustively and it makes it hard to access the fairness of the experiments.

- **Mathematical notations for theoretical results:** I see the authors make a tremendous effort to solidify the theoretical ground of the proposed framework. However many mathematical notations are not introduced satisfactorily and I could not access the correctness of the theoretical result. Followings are some of the points which I saw were necessary to be introduced properly when I went through Appendix C. In addition, some of the symbols and notations are apparently coming from the previous sections. I suggest the section dedicated for the notations should be placed at the beginning of the appendix.
  - Topology and metric on the extend real numbers.
  - The definition of $\mathcal{B}(\mathbb{R}^{d+1})$.
  - It is not clear that for which mathematical objects the assumptions 5 and 6 are made.
  - The definition of the terminal function (I speculate that it is a function describing terminal conditions for PDEs)
  - $\breve{F}$ in Definition8. Apparently this is different from one introduced in Fact 1 in the main text.
  - I still do not understand the assertion in Remark 8.

**Questions:**

- What does training points $m$ mean? How does $m$ training points translate to $m$ additional Monte Carlo paths?
- How and/or along which metric is the computational budget given for MLP in the main experiments ensured to be comparable to that taken for surrogate models and the SCaSML?
- How long/how many training epochs/what change in hyperparameters is additionally necessary for MLP and SR models to reach the comparable performance of SCaSML reported in Table 1? While theoretical analysis on the variance and computational cost is given, I believe this observation and/or potential contrast could strengthen the significance of the contribution of the paper.
- In what practical scenario do the authors expect the proposed framework play a crucial role?

---

> ### Author Response · Authors · 2025-11-21
>
> We thank the reviewer for their feedback and for finding our approach "intriguing" and the performance gains "impressive." We are confident we can resolve the reviewer's concerns regarding practical implications, computational budgets, and mathematical notation. We have substantially strengthened the paper by
> - clarifying why inference-time scaling matters and how our proposed law of defect fundamentally differs from prior defect-refinement techniques.  **We revise the terminology from the “law of defect” to the Structural-Preserving Law of Defect, highlighting the structural preservation that is central to our contribution.**
> - In the Appendix, we now include 10× repeated experiments for all methods, reporting 95% confidence intervals and pairwise t-test p-values, demonstrating that our improvements are statistically significant across benchmarks.
> - We further added Hexagonal Binning visualizations of pointwise error differences, showing that SCaSML improves performance on the vast majority of test points, confirming that our gains are not isolated to a few spatial outliers but are broadly distributed throughout the domain.
> - We have added a "Fixed Computational Budget Analysis" in Appendix G.6. In this experiment, we restricted the baseline PINN and SCaSML to the exact same total wall-clock time (Training + Inference). The results, visualized in Figure 17, show that SCaSML consistently achieves lower error than the baseline PINN under the same budget.
> - We have added a "Appendix A: Notations" section to centralize and define all symbols.
>
> ## Why Improve a High-Performing Data-Driven Model?
>  The answer lies in the distinction between global approximation and local precision. Although the surrogate is a strong global approximator, it is trained to fit the entire domain on average. However, in many high-stakes scientific and financial applications, users require high accuracy at specific query points. This mirrors how Large Language Models, despite being strong global models, often rely on inference-time search or planning to answer specific complex queries. Training compute is fixed and expensive, whereas inference compute is cheap and "elastic"—applied only when the user requests a solution. Therefore, additional inference-time compute is not redundant; it targets the remaining local defect that the surrogate cannot eliminate without prohibitively expensive retraining.
>
> ## Regarding practical scenarios
> First, in optimal control and financial pricing (e.g., nonlinear Black-Scholes), practitioners often need the value function and its gradient only at the current state to determine the next action or hedge. Our method allows them to obtain a fast approximation via the neural network and then systematically refine it online using Monte Carlo correction. Second, in rare-event analysis and committor problems arising in molecular dynamics, the solution is often required only at specific initial configurations. Third, in goal-oriented estimation, users often need a specific functional of the solution rather than the full field. In all these cases, solving the PDE to high precision over the entire domain during training is computationally wasteful; inference-time correction focuses resources exactly where they are needed.

---

> ### Author Response · Authors · 2025-11-21
>
> ## Regards computational budget
> The reviewer also asked a critical question regarding computational efficiency: how much longer would the surrogate or MLP models need to run to match SCaSML? This touches on the core strength of our method. **To answer this empirically, we have added a "Fixed Computational Budget Analysis" in Appendix G.6.** In this experiment, we restricted the baseline PINN and SCaSML to the exact same total wall-clock time (Training + Inference). The results, visualized in Figure 17, show that SCaSML consistently achieves lower error than the baseline PINN under the same budget. This confirms our theoretical scaling law (Corollary 4), which proves that our hybrid framework converges at a rate of O(m^(-gamma - 1/2)), which is asymptotically faster than the O(m^(-gamma)) rate of the surrogate alone,  where $m$ is the number of collocation points. **(We numerically examine this on GP numerically in Figure 4.)** Consequently, for the baseline surrogate to match the accuracy of SCaSML, it would require a larger, and likely intractable, amount of additional training time.
>
> ## Regards Minor Questions
> The training points in the paper regard how many times we need to access the $f$ and $g$ functions in the PDE, which is proportional to the number of Monte Carlo Paths. We apologize for the confusion in the original manuscript. We have added a "Appendix A: Notations" section to centralize and define all symbols. To briefly clarify the specific points raised: F-tilde is the modified nonlinearity for the defect PDE, defined in Fact 1. Assumptions 5 and 6 are standard Lipschitz continuity assumptions applied to the original PDE's nonlinearity F and terminal condition g. The "terminal function" g-tilde represents the error in the terminal condition: g-tilde(y) = g(y) - u_hat(T, y). Remark 8 asserts a key stability result: the Lipschitz constant of the new defect nonlinearity is bounded by that of the original, ensuring the defect PDE is numerically stable. Finally, regarding the MLP hyperparameters, we have ensured these are explicitly stated in the experimental setup for each section (e.g., sample base M=10, levels N=2) and have consolidated these details in the appendix to facilitate reproducibility.
>
> ---
>
> Thanks again for your efforts in reviewing this paper. We hope we have addressed your concerns regarding novelty and experiments and have incorporated them in the manuscript. We would be more than happy if you could reconsider the review rating.

---

> ### Comment · Reviewer_op81 · 2025-11-26
>
> Thank you for the reply. My concerns and questions regarding the experimental part were addressed adequately. In particular, the necessity of the proposed method for the practical application is compelling. Indeed, local precision matters in many engineering domains, especially in fluid simulation near the airfoil boundary as the fluid detaches from the boundary. In those applications, in contrast to the requirement, SOTA machine learning models generally struggle to meet the precision requirement for their practical use. Hence, the impact and indication of the method in real applications are expected to be considerable.
>
> On the other hand, while I see some revisions are made in the theoretical parts, I am still stuck on Appendix D and cannot begin proofreading the theoretical statements because I still see inconsistent notation, implicit assumptions, and non-trivial assertions. These points may already have embedded additional inconsistent or incorrect descriptions into the subsequent proof, which could prevent potential readers (and me) from following the proof on their own. While the practical implications of the method are significant, I still feel reluctant to raise my score due to the lack of theoretical rigor. The theoretical part of the paper needs a thorough revision; therefore, I will maintain my score. Should the authors revise and supplement the theoretical parts in a way that makes them more accessible, I would be happy to try to read the revision (I have to say I am not sure if I could read everything during the rebuttal period) and engage in further discussion.

---

> > ### Author Response · Authors · 2025-11-26
> >
> > Sorry for the earlier notational confusion. The structure of our argument is actually straightforward. The only new technical contribution in our paper is Lemma 3; all remaining steps follow directly by plugging Lemma 3 into the existing complexity bounds of Hutzenthaler & Kruse (2020).
> >
> > Hutzenthaler & Kruse (2020) show that the computational complexity of simulating MLP schemes is governed precisely by the quantity that appears on the left-hand side of our Lemma 3. Therefore, once Lemma 3 is established, the overall complexity of the MLP method immediately becomes controlled by the approximation error of the surrogate model.
> >
> > Since the full convergence and complexity analysis has already been rigorously developed in Hutzenthaler & Kruse (2020), we do not repeat those arguments here; our proof simply invokes their results after supplying the missing bound provided by Lemma 3.
> >
> > We also promise to revise the notation and improve the presentation of the appendix this week. These adjustments will make the proof easier to follow and ensure full consistency across the main text and supplementary material.

---

### Official Review · Reviewer_Wcpd · 2025-10-29

**Soundness:** 2
**Presentation:** 1
**Contribution:** 2
**Rating:** 2
**Confidence:** 4

**Summary:**

The paper discusses inference-time scaling for machine-learning (ML) solvers for high-dimensional partial differential equations (PDEs). Concretely, the idea is to train a physics-informed neural network (PINN) or Gaussian process (or any other PDE solver that yields a continuously differentiable solution), derive the defect correction PDE which describes the error of the approximation, and solve this error PDE with a Feynman-Kac-based solver at inference time to correct the PINN prediction.
This chaining of solvers improves the accuracy order: the final error is the product of the individual errors.
The resulting algorithm is benchmarked on a range of PDEs in up to 160 dimensions, including a Hamilton--Jacobi--Bellman problem.


**Summary of my recommendation:**
I appreciate the idea of inference-time corrections to PINN predictions, and I definitely consider it promising. However, as is, I recommend rejecting this work because the defect correction approach is not as novel as the submission states (details below) and because the submission's experiments have some gaps that need to be filled before publication. I believe that if a future version of this manuscript embedded itself better into the literature of numerical solvers and if the experimental gaps were filled, it would be stronger, and I would probably have given a different score. But I believe that these changes are beyond the scope of a revision during the rebuttal period, which is why I give a "reject" score.

**Strengths:**

Even though the idea of chaining two (or more) solvers, like the submission does with PINNs/GPs and Feynman--Kac simulations, is not new, its application to training-vs-inference stages in physics-informed ML is novel. Inference-time scaling had a significant impact on LLMs (though I'm not an expert on LLMs), and I wouldn't be surprised if it became standard in future versions of physics-informed ML solvers as well.
Furthermore, I appreciate the mathematical rigour of the analysis and the fact that the experiments cover a range of problems, including linear and nonlinear ones, as well as truly high-dimensional problems (dim $\gg$ 100).

**Weaknesses:**

Despite the submission's strengths, I think three weaknesses dominate, which is why I ultimately recommend rejecting this work:

1. What the paper calls the "law of total defect" (Section 2.1, Definition 1) is a defect correction approach, and defect correction methods have been studied in numerical analysis for a long time; for example, see:

    > Stetter, Hans J. "The defect correction principle and discretization methods." Numerische Mathematik 29.4 (1978): 425-443.

    > Dutt, Alok, Leslie Greengard, and Vladimir Rokhlin. "Spectral deferred correction methods for ordinary differential equations." BIT Numerical Mathematics 40.2 (2000): 241-266.

    and many others. The submission suggests that the "law of total defect" is a central contribution of this work (e.g. in lines 017f, 063f, or 073f), which is misleading because describing the approximation error of a PDE via the defect equation is a standard approach in numerical analysis. Furthermore, it's known that the approximation order of defect correction approaches is the product of the individual orders, which raises questions about the novelty of the submission's theoretical analysis in Theorem 3. In summary, while the application of defect correction techniques to training vs. inference stages of ML-based PDE solvers is new, the general construction is not. For a future version of this manuscript, I would recommend positioning this work more clearly as a ML-application of existing defect correction techniques (adjusting the claims and the related work accordingly).

2. The distinction between training and inference time of ML-based PDE solvers is a bit arbitrary in the current setup: both the PINN and the Monte-Carlo solver contribute to solving the PDE, and there is no reason to delay the Monte-Carlo solver to inference time (unless I have missed something). I imagine that the separation of training and inference would become clearer for parameter estimation problems, where external data enters the loss function, but parameter estimation is not studied in the current version of the submission. I believe that a discussion of inference-time scaling with ML-based PDE solvers requires a more thorough distinction of training and inference time.

3. The numerical results, while broad, are not as conclusive as they could be. On the one hand, the accuracy differences between the methods are small, but standard deviations are not reported. From the current results, it is unclear whether the improvement is statistically significant. On the other hand, I think that the accuracy improvements would be more convincing if they were normalised by budget in the sense that all three "competitors" (PINN/GP, MC, PINN/GP+MC) should use similar compute (either function evaluations or runtime/memory). For example, in Table 1, the baselines are slightly less accurate but much faster, and I don't know whether running the baselines for longer would make them more accurate.


4. (Minor) Finally, and I consider this to be a smaller weakness than the three above, I wonder whether the volume of the paper (almost 40 pages) is a bit too high for an ICLR paper, and not necessarily proportional to the amount of contributions of this work. I understand it is common practice to defer proofs and extra results to the appendices, but it is currently almost impossible to read the main paper without extensive reference to the supplement. That said, while I think this point is worth mentioning, I consider it less important than the three above.

**Questions:**

The following questions don't affect my score, but I think including their answers in a revision would improve the submission.

- Line 038f suggests that ML-based solvers don't suffer from the curse of dimensionality like traditional solvers do. This statement needs to be supported more strongly.
- Lines 061f: PINNs are not surrogate models. A surrogate model is an approximate model used to replace an expensive simulation. PINNs are (at their core) PDE solvers. I suggest replacing "surrogate model" with "ML-based PDE solvers".
- Experiment in Section 3.1: what's the boundary condition? There are boundary collocation points, but the problem statement does not mention boundary conditions.
- Line 126: How costly is it to compute the full Laplacian in 160 dimensions?
- Lines 365f: Defect correction methods are sometimes unreliable if the residual is only approximate. How problematic is it to use Hutchinson's estimator for evaluating Laplacians here? For reference, Lines 425f suggest that inaccurate Laplacians are problematic for the experiment in Section 3.4

---

> ### Author Response · Authors · 2025-11-21
>
> We thank the reviewer for the thoughtful feedback. We understand that review’s major concerns about the novelty of the “Law of Defect,” the distinction between training and inference, and potential gaps in the experimental comparisons. We apologize if the specific context of high-dimensional solvers was not clear in the initial submission, as these concerns appear to stem from a misunderstanding of the unique constraints imposed by the curse of dimensionality. We have substantially strengthened the paper by
>
> - clarifying why inference-time scaling matters and how our proposed law of defect fundamentally differs from prior defect-refinement techniques. **We revise the terminology from the “law of defect” to the Structural-Preserving Law of Defect, highlighting the structural preservation that is central to our contribution**.
> - In the Appendix, we now include 10× repeated experiments for all methods, reporting 95% confidence intervals and pairwise t-test p-values, demonstrating that our improvements are statistically significant across benchmarks.
> - We further added Hexagonal Binning visualizations of pointwise error differences, showing that SCaSML improves performance on the vast majority of test points, confirming that our gains are not isolated to a few spatial outliers but are broadly distributed throughout the domain.
> - We have added a "Fixed Computational Budget Analysis" in Appendix G.6. In this experiment, we restricted the baseline PINN and SCaSML to the exact same total wall-clock time (Training + Inference). The results, visualized in Figure 17, show that SCaSML consistently achieves lower error than the baseline PINN under the same budget.
> - We have added a "Appendix A: Notations" section to centralize and define all symbols.
>
> ## Regards Novelty
> We thank the reviewer for citing the foundational work by Stetter and Dutt. We fully agree that the general concept of defect correction is a standard tool in numerical analysis. However, a critical distinction must be made regarding applicability. The methods cited are grid-based and fundamentally low-dimensional; they become computationally intractable for the high-dimensional problems we study, which reach up to 160 dimensions. In this high-dimensional regime, the only hope is relying on probabilistic, grid-free Monte Carlo solvers based on the Feynman-Kac formula. The central technical challenge is that a valid Feynman-Kac representation exists only for semi-linear PDEs. Classical defect correction methods rely on Newton-type linearizations that generally destroy this semi-linear structure and the positive semi-definiteness required for the diffusion coefficient (when the nonlinearity is dependent on the gradient).  Our core contribution, the "Law of Defect," is the derivation of a correction PDE that naturally preserves this semi-linear structure. This structural preservation is mathematically non-trivial and is precisely what **enables defect correction to be combined with high-dimensional Monte Carlo solvers for the first time**. Thus, this is not merely an application of an existing technique, but a new formulation that makes correction compatible with the only viable class of high-dimensional solvers. As discussed in lines 170–176, to the best of our knowledge, **ours is the first work to demonstrate an improved convergence rate for defect correction. This stems from the fact that our Monte Carlo solver does not require smoothness of the defect term**, which typically contains high-frequency components that are difficult to approximate.
>
> ## Regards distinction between "training" and "inference"
> Our separation of training and inference is exactly the same as inference time scaling algorithms for LLM. Training corresponds to solving the PDE globally on the entire domain, learning a map that approximates the solution everywhere. In contrast, inference-time correction solves the PDE only at a specific, user-specified state. This separation is natural and parallels standard practices in machine learning: a base language model is trained once to answer all queries, while computationally intensive refinement (like beam search or planning) is invoked at inference time only when high precision is required for a specific input. This separation enables "elastic compute," allowing users to trade inference time for accuracy on demand without incurring the massive fixed cost of retraining the global model.

---

> ### Author Response · Authors · 2025-11-21
>
> ## Regards computational budget
> To answer this empirically, we have added a new "Fixed Computational Budget Analysis" in Appendix G.6 of the revision. In this experiment, **we restricted both the baseline PINN and SCaSML to the exact same total wall-clock time (Training + Inference). **The results, visualized in Figure 17, show that SCaSML consistently achieves lower error than the baseline PINN under the identical time budget. This occurs because training deep networks suffers from diminishing returns; simply training the baseline longer does not yield a proportional improvement because the remaining high-frequency defect is difficult for neural networks to fit due to spectral bias. In contrast, our Monte Carlo correction converges at a rate independent of the solution's smoothness. Furthermore, our Corollary 4 proves that SCaSML achieves a faster asymptotic convergence rate of O(m^(-gamma-1/2)) where $m$ is the number of collocation points.  **(We numerically examine this on GP numerically in Figure 4.)** This multiplicative speedup explains why allocating marginal compute to inference-time correction yields a higher return on investment than allocating it to further training. We have also added Appendix G.5, reporting standard deviations and p-values (p << 0.001) to confirm these improvements are statistically significant.
>
> Finally, regarding the minor questions: In this work, we treat PINNs and GPs as physics-constrained surrogate models of the Feynman–Kac–based Monte Carlo solver: after training, they allow fast pointwise evaluation of the PDE solution, avoiding repeated expensive stochastic simulations. We clarified in Section 3.1 that Dirichlet boundary conditions are enforced on the hypercube. We use Hutchinson's estimator to accelerate the computation of Laplacian which is not expansive even in 160d. While Hutchinson's estimator does introduce error, our method still stands out, as shown by the successful error reduction in the 160D LQG experiment.
>
> ---
>
> Thanks again for your efforts into reviewing this paper. We hope we have addressed your concerns regarding novelty and experiments and have incorporated them in the manuscript. We are more than happy if you can reconsider the review rating.

---

> > ### Comment · Reviewer_Wcpd · 2025-11-25
> >
> > Thank you for the detailed replies! I appreciate the revisions to the paper, but my concerns are not fully resolved. Let me address the key points individually:
> >
> > - "Training vs inference": Thanks for the update. I agree that general training vs pointwise evaluation could be a meaningful distinction. However, my original review mentioned the need for a "thorough discussion of training vs inference time". While Remark 2 helps, a more substantial revision would make the changes more convincing; for example, a motivation for "elastic compute" that is more specific to physics-informed ML than referencing its importance for LLMs.
> >
> >
> >
> >
> > - "Novelty": Thank you for updating the paper, but here too, I am not fully convinced by the changes for two reasons:
> >     1. My original review noted a lack of embedding into the numerical analysis literature, providing two classic papers as a starting point. I would have expected the revision to include a more comprehensive discussion of the (large) field of defect correction, possibly also including adjacent fields like a posteriori error analysis for PDE solvers, but the revision only mentions the two papers from my review.
> >     2. Preserving the semilinear structure does not appear as novel as the revised paper (and the rebuttal) states. When applying defect correction to semilinear PDEs, what the paper calls Fact 1 is similar to results in works on defect correction, a-posteriori error estimation, and Newton-Galerkin methods, e.g.:
> >
> >         > Xu, J. (1994). A novel two-grid method for semilinear elliptic equations. SIAM Journal on Scientific Computing, 15(1), 231-237.
> >
> >         > Amrein, Mario, and Thomas P. Wihler. "Fully adaptive Newton--Galerkin methods for semilinear elliptic partial differential equations." SIAM Journal on Scientific Computing 37.4 (2015): A1637-A1657.
> >
> >         > Nochetto, R. H., Schmidt, A., Siebert, K. G., & Veeser, A. (2006). Pointwise a posteriori error estimates for monotone semi-linear equations. Numerische Mathematik, 104(4), 515-538.
> >
> >     While there are some differences (weak vs strong forms, linearisation of the nonlinearity), I do not see the novelty of the "structural-preserving law of defect" as sufficient to be a core contribution of an ICLR paper.
> >
> >
> >
> > - "Computational budget": I appreciate the addition of equal-budget experiments. However, I am unsure whether simply training a PINN baseline for more epochs is meaningful, because once the optimiser has converged, additional steps do not improve the model. My expectation of an "equal compute" comparison was closer to budget allocations like, for example, training a larger model until convergence, instead of an equally sized model beyond convergence. Regarding the statistical significance: Thank you for these changes. I think they make the results more convincing.
> >
> > In summary, I thank the authors again for the detailed explanations and the revisions to the paper. While I recognise improvements, I fear my concerns remain, so I will maintain my score.

---

> ### Author Response · Authors · 2025-11-26
>
> We thank the reviewer for the response, we still have several points we wants to clarify.
> ### Regards Literature
>
> Classical finite element methods admit a well-characterized asymptotic error expansion [1,2], which enables defect-correction schemes to systematically remove the leading error term and improve convergence rates [3–5]. In contrast, no such asymptotic structure is available for neural networks: NN approximations lack any mesh-refinement hierarchy, their errors do not exhibit a polynomial expansion with respect to a single resolution parameter, and the optimization-induced approximation error provides no perturbative decomposition. A different family of debiasing techniques in numerical PDEs relies on iterative solvers such as Newton methods [6–8] and quasi-Newton methods [9,10]. However, these methods present two fundamental limitations in our setting. First, iterative updates produce only approximate corrections, whereas our law of defect is an exact analytical identity that delivers a closed-form unbiased correction in a single step. Second, embedding iterative methods into a Monte–Carlo or Feynman–Kac framework is highly inefficient: each iteration requires recomputing residuals and Jacobian actions through additional Monte–Carlo estimators, producing a nested simulation hierarchy whose convergence rate rapidly deteriorates—from the standard $\mathcal{O}(N^{-1/2})$ rate for a single Monte–Carlo level, to $\mathcal{O}(N^{-1/4})$ for a second iteration, $\mathcal{O}(N^{-1/8})$ for a third, and so on as more levels are introduced. Practitioners are therefore forced to balance early-termination errors against the rapidly declining statistical efficiency of nested Monte–Carlo estimates, making these approaches both computationally expensive and numerically unstable.
>
> We plan to incorporate this discussion into the revised manuscript. As far as I know the defect correction method **Before proceeding, we would appreciate hearing whether the reviewer has any additional concerns or suggestions regarding this point.**
>
> [1] An Analysis of the Finite Element Method.
>
> [2] A Posteriori Error Estimation in Finite Element Analysis.
>
> [3] The superconvergent patch recovery (SPR) and a posteriori error estimates.
>
> [4] Some a posteriori error estimators for elliptic partial differential equations.
>
> [5] Superconvergence and a posteriori error estimation for the Poisson equation by the defect correction method,
>
> [6] A novel two-grid method for semilinear equations
>
> [7] Discrete Newton methods and iterated defect corrections
>
> [8]  Spectral deferred correction
>
> [9] Iterative solution of transonic flows using defect correction
>
> [10] A review of a posteriori error estimation techniques for elasticity problems
>
> ### Regards Novelty
> Although the algebra leading to our {Law of Defect} may, in hindsight, look like  “subtracting two equations,” the resulting identity is highly nontrivial. Starting from the semilinear parabolic PDE satisfied by the true solution $u_\infty$ and the approximate surrogate $\hat u$, we rewrite the solution as $u_\infty = \hat u + u^\vee$ and derive a **{closed}, well-posed semilinear PDE** for the  defect $u^\vee$. Crucially, the nonlinearity in this equation, $\mathcal{F}^\vee$, depends only on the surrogate and the defect, and preserves the exact structural form required by the Feynman--Kac representation and  Monte Carlo solvers.
>
> Beyond the identity itself, we prove that this particular decomposition yields a  **rigorous variance-reduction effect**. Namely, **the variance of the Monte Carlo estimator for the defect is proportional to the pathwise PDE residual**, so as the surrogate improves at rate $m^{-\gamma}$, the Monte  Carlo variance scales like $m^{-1/2-\gamma}$, giving **a provable acceleration of the overall convergence rate of the “surrogate + simulation’’ estimator**. Each algebraic step in the derivation may appear simple in isolation—splitting $u_\infty$ into $\hat u$ and $u^\vee$, defining the residual, applying  Feynman--Kac—but identifying the \emph{right} combination of analytical and probabilistic  tools that (i) yields a closed semilinear PDE for the defect and (ii) guarantees provable decay of Monte Carlo variance is highly nontrivial.
>
> To the best of our knowledge, the existing literature contains no prior work that provides  a Monte Carlo--based correction law of this form. **If such work exists, we would be grateful  if the reviewers could point us to it. In this light, we believe it is not accurate to say that our contribution falls below the novelty bar expected for an ICLR submission.**
>
> ### Regards Larger PINNs
>
> Our GPU cluster is currently running the experiment, and we will  update the figure accordingly. We will share the results with you before the end of this week.
>
> ---
> Thanks again for your efforts into reviewing this paper. We hope we have addressed your concerns regarding novelty and Let us know your further concern.

---

> ### Comment · Reviewer_Wcpd · 2025-11-26
>
> Thank you for elaborating.
> Regarding your "Novelty" reply, I disagree with the "highly nontrivial" and I do not believe that Fact 1 is as novel as the paper states. Details are in my previous reply.
> For context, I never questioned the novelty of (i) using PINNs + MC for a defect-correction two-stage scheme (typically, coarse and fine-grid finite-element methods seem to be used), and (ii) using this two-stage approach for inference time scaling. But the paper presents Fact 1 as a core contribution, which my review (and my previous reply) point out as a considerable weakness.
>
> Relatedly, I also disagree with the limitations in the "literature" paragraph. For example, the discussion of quasi-Newton methods as approximate correction, in contrast to the submitted version which is stated as exact. Both build on the same identity, which the submission calls Fact 1, but use different correction methods.
>
> Finally, I am unable to find any of the references [3-9] listed in the previous message. I doubt they exist.

---

> ### Author Response · Authors · 2025-11-26
>
> > Regards Revision Plan
>  Now I understand your concern. Sorry for the potential misleading in the original paper, I never wanted to state that describing the approximation error of a PDE via the defect equation is our contribution. I want to present our equation (7) as new. I'll soften the word, change the presentation, and start from defect correction literature to say we want to propose a new equation. We also consider our core contribution as (i) using PINNs + MC for a defect-correction two-stage scheme **and making the Monte-Carlo estimator compatible with this defect-corrected structure**, (ii) using this two-stage approach for inference time scaling (iii) **improved convergence rate for MC** as the core contribution of the paper and does not falls below the novelty bar expected for an ICLR submission.
>
> **We also plan to change our title to  Physics-Informed Inference Time Scaling for Solving High-Dimensional Partial Differential Equations via Defect Correction**
>
> I'll also change the related work paragraph as
>
> Classical finite element methods admit a well-characterized asymptotic error expansion [1,2], which enables defect-correction schemes to systematically remove the leading error term and improve convergence rates [3–5]. In contrast, no such asymptotic structure is available for neural networks: NN approximations lack any mesh-refinement hierarchy, their errors do not exhibit a polynomial expansion with respect to a single resolution parameter, and the optimization-induced approximation error provides no perturbative decomposition. A different family of debiasing techniques in numerical PDEs relies on iterative solvers such as Newton methods [6–8] and quasi-Newton methods [9,10]. However, these methods present two fundamental limitations in our setting. First, iterative updates produce only approximate corrections, whereas our law of defect is an exact analytical identity that delivers a closed-form unbiased correction in a single step. Second, embedding iterative methods into a Monte–Carlo or Feynman–Kac framework is highly inefficient: each iteration requires recomputing residuals and Jacobian actions through additional Monte–Carlo estimators, producing a nested simulation hierarchy whose convergence rate rapidly deteriorates—from the standard $\mathcal{O}(N^{-1/2})$ rate for a single Monte–Carlo level, to $\mathcal{O}(N^{-1/4})$ for a second iteration, $\mathcal{O}(N^{-1/8})$ for a third, and so on as more levels are introduced. Practitioners are therefore forced to balance early-termination errors against the rapidly declining statistical efficiency of nested Monte–Carlo estimates, making these approaches both computationally expensive and numerically unstable.
>
> We'll also add an increasing size PINN experiment in the budget comparison figure.
>
> **Before proceeding, we would appreciate hearing whether the reviewer has any additional concerns or suggestions, and whether this revision helps alleviate the earlier issues or clarifies our contribution.**
>
> > Regards references
> Sorry for the confusion. Due to space limitations, we used abbreviations in the main text, which may have caused ambiguity. All referenced works do exist, and their complete bibliographic information is provided here:
>
> [3] Zienkiewicz O C, Zhu J Z. The superconvergent patch recovery and a posteriori error estimates. Part 1: The recovery technique[J]. International Journal for Numerical Methods in Engineering, 1992, 33(7): 1331-1364.
>
> [4] Bank R E, Weiser A. Some a posteriori error estimators for elliptic partial differential equations[J]. Mathematics of computation, 1985, 44(170): 283-301.
>
> [5] J. H. Brandts Superconvergence and a posteriori error estimation for triangular mixed finite elements.
> Numerische Mathematik, 68:311–324, 1994.
>
> [6] Xu, J. (1994). A novel two-grid method for semilinear elliptic equations. SIAM Journal on Scientific Computing, 15(1), 231-237.
>
> [7] K. Böhmer, “Discrete Newton methods and iterated defect corrections,” Numerische Mathematik, 37, 167–192, 1981.
>
> [8]  Dutt, Alok, Leslie Greengard, and Vladimir Rokhlin. "Spectral deferred correction methods for ordinary differential equations." BIT Numerical Mathematics 40.2 (2000): 241-266.
>
> [9] Jameson A. Iterative solution of transonic flows over airfoils and wings, including flows. Communications on pure and applied mathematics, 1974, 27(3): 283-309.
>
> [10] Verfürth R. A review of a posteriori error estimation techniques for elasticity problems[J]. Computer Methods in Applied Mechanics and Engineering, 1999, 176(1-4): 419-440.

---

> > ### Comment · Reviewer_Wcpd · 2025-11-26
> >
> > Thank you for acknowledging the misunderstanding. I think a future version of the submission that adheres to the structure proposed above could be a more valuable contribution. However, with the kinds of edits needed to implement this change, the revision will differ substantially from the initial paper, and I believe a new set of reviews would be required.
> > For reference, my original review states that "I believe that these changes are beyond the scope of a revision during the rebuttal period". After discussing, I maintain this assessment; thus, I will keep my initial recommendation. Still, I thank you for the continued engagement.
> >
> > Regarding the references, thank you for giving explicit citation entries. Some of them seem to indeed have been abbreviations. However, I think it might be worth mentioning that references 5 and 9 were not abbreviations but entirely different titles altogether.

---

> > > ### Author Response · Authors · 2025-11-26
> > >
> > > We have uploaded a revised version of the manuscript. While we respectfully disagree that these changes warrant a new round of review, we have carefully addressed the reviewer’s concerns. Importantly, we did not make any major structural changes to the paper; we only added clarifying discussions. In particular, we now explicitly state throughout the manuscript—from the title and introduction to the main text—that our approach is based on the defect-correction method. The three contributions mentioned above have always been the core contributions of the paper, and we have now made them even clearer in the revision.

---

### Official Review · Reviewer_Kghx · 2025-10-31

**Soundness:** 4
**Presentation:** 3
**Contribution:** 4
**Rating:** 8
**Confidence:** 4

**Summary:**

The paper proposes a method to estimate and correct the error predicted by a surrogate model when solving a PDE equation.

The method applies to semi-linear parabolic PDE equations and is based on recent results on Multilevel Picard (MLP) solvers.

The paper describes the approach, and in the Annex provides extensive details of the approach.

The paper evaluates the approach on a few tasks: linear convection-diffusion equation, viscous Burger equation, high-dimensional Hamilton-Jacobi-Bellman equation, and diffusion-reaction equation.

It would be nice to highlight the computational or sample complexity of the method (there is a paragraph in 2.3, maybe a clarification of the impact on the specific cases in the annex would be nice).

The paper mentions in the first paragraph the use of the method in various applications:

1. Schroedinger equation in quantum many-body systems,
2. nonlinear Black–Scholes equations in finance, and
3. the Hamilton–Jacobi–Bellman equation

while there are experiments for the last, it would be nice to understand if the application to the other two cases requires some modification to the approach.

It is not completely clear what is observed. From eq. (2), the forcing term f(r,y) is used to define dht error. It is not clear what is the connection between the error (or residual) and the defect. We have access to the prediction $\hat u(r,y)$, what is also observed?

**Strengths:**

The paper presents a post-prediction method that improves the accuracy by solving a PDE, that describes the "defect", which is then computed by solving the "law of defect" or the error PDE.

**Weaknesses:**

Only clarification.

It would be nice if the authors would release the code, since it would be difficult to build on top of this work.

**Questions:**

Mentioned before:

1. Clarify the applicability to other problems.
2. Give an intuitive explanation of what the expectation operator $\Phi$ of Eq. (8) does
3. possibly a complexity (sample, computation) analysis (probably what is described in line 264 and 1662, a summary of the results would be nice).

**Details Of Ethics Concerns:**

No concern

---

> ### Author Response · Authors · 2025-11-21
>
> We thank the reviewer for the excellent rating and their insightful questions. We are glad they found the paper's contribution and soundness to be excellent. We have updated the manuscript to reflect your suggestions and provide the requested clarifications.
> - clarifying why inference-time scaling matters and how our proposed law of defect fundamentally differs from prior defect-refinement techniques. We revise the terminology from the “law of defect” to the Structural-Preserving Law of Defect, highlighting the structural preservation that is central to our contribution.
> - In the Appendix, we now include 10× repeated experiments for all methods, reporting 95% confidence intervals and pairwise t-test p-values, demonstrating that our improvements are statistically significant across benchmarks.
> - We further added Hexagonal Binning visualizations of pointwise error differences, showing that SCaSML improves performance on the vast majority of test points, confirming that our gains are not isolated to a few spatial outliers but are broadly distributed throughout the domain.
> - We have added a "Fixed Computational Budget Analysis" in Appendix G.6. In this experiment, we restricted the baseline PINN and SCaSML to the exact same total wall-clock time (Training + Inference). The results, visualized in Figure 17, show that SCaSML consistently achieves lower error than the baseline PINN under the same budget.
> - We have added a "Appendix A: Notations" section to centralize and define all symbols.
>
> Regarding applicability to other problems, the reviewer specifically mentioned the nonlinear Black–Scholes and Schrödinger equations. Our framework is directly applicable to a broad class of nonlinear Black–Scholes models, which serve as canonical examples of the semi-linear parabolic PDEs captured by our general formulation. For the Schrödinger equation, our approach can be instantiated by first employing a variational Monte Carlo (VMC) method to obtain an initial estimator and then performing a targeted correction using diffusion Monte Carlo (DMC). While this adaptation requires additional algorithmic engineering, it aligns naturally with our framework. We are actively pursuing this extension in ongoing follow-up work.
>
> Sorry for missing an intuitive explanation of the expectation operator $\Phi$ in Equation (8). We have added an explanation to Section 2.3 describing Phi as a Feynman–Kac–type
> backward propagator. It takes a function representing the solution at a future time and calculates the solution at the current time by averaging over all possible stochastic paths. Specifically, it computes the expected value of the terminal condition and the integrated "running cost" (the nonlinearity F) accumulated along those paths. The true solution is the one that, when "propagated" by Phi, remains unchanged.
>
> The "residual" (epsilon) is the amount by which the surrogate fails to satisfy the PDE; it is a known, computable function obtained by plugging the surrogate into the PDE operator. The "defect" (u_tilde) is the true, unknown error of the surrogate. The relationship between the "residual" and the "defect" is linked by the "Law of Defect" (Fact 1): it shows that the defect is the solution to a new semi-linear PDE where the residual acts as a source term. By computing the residual, we can solve this new PDE to find the unknown defect. We further assume—mildly—that an approximation $\hat{u}(r,y)$ is available, given by the PINN/GP surrogate we have trained.
>
> The explanation summary of our theoretical results is at lines 266-275. Our main results (Corollaries 4 and 6) show that SCaSML achieves a provably faster convergence rate. If the surrogate model alone converges at a rate $l{O}(m^{-\gamma})$ using $m$ training points, then allocating an additional $m$ Monte Carlo samples at inference time enables our hybrid method to reduce the error to ${O}(m^{-\gamma - 1/2})$. This improved rate follows from the fact that the variance of the Monte Carlo correction is proportional to the squared magnitude of the residual, which scales as ${O}(m^{-2\gamma})$, yielding a standard-error reduction of ${O}(m^{-\gamma - 1/2})$.
>
> Theorems 7 and 9 further show that the computational cost to reach a target error decreases as the quality of the initial surrogate improves. To demonstrate this efficiency empirically, we added a "Fixed Computational Budget Analysis" in Appendix G.6, which shows that SCaSML achieves lower error than baselines even when total wall-clock time is held constant.
>
> Finally, we thank the reviewer for their interest in the implementation. We have added a statement to the Conclusion confirming that our code, built on JAX and DeepXDE, will be made publicly available once the paper has completed de-anonymization.
>
> Thank again for the reviewer's effort and we would like to answer reviewer's further questions.

---

### Official Review · Reviewer_RnQk · 2025-11-03

**Soundness:** 3
**Presentation:** 3
**Contribution:** 2
**Rating:** 4
**Confidence:** 4

**Summary:**

This paper proposes to start from an initial solution obtained by neural operators and then solve the residue as a new PDE solving problem. This paper uses two variants of MCMC to solve the new PDE. Theoreical results are provided to show the proposed method has a faster convergence rate. The paper conducted experiments on multiple datasets and achieved the best accuracy among all the baselines.

**Strengths:**

1. By formulating the residual part as another PDE, the paper further improves the PINN and GP based surrogates without training any new networks.
2. The paper theoretically shows the proposed method has a faster convergence rate.
3. The proposed method is compared on multiple datasets and achieves the best accuracy among all the baselines.

**Weaknesses:**

1. The idea of correcting the initial solution is not new. The paper chooses to use the existing MCMC variants (quadrature MLP and full-history MLP) for the correction, which are also not new.
2. It is not clear to me how the paper stands out from the existing works, where correction is applied to the initial solution for improvement.
3. The experiments are not convincing for that the baselines (PINN) is not tuned enough. A fixed architecture, a fixed optimization strategy, and a fixed number of points are used. To show the effectiveness of the proposed method, the authors can consider starting with a stronger initial solution. More layers and more collocation points will lead to more runtime, and it is interesting to see how that part scales with the complexity compared with the proposed method. For example, for SR, the runtime is about 1 - 2 seconds, while for the proposed method, the runtime is nearly 10 times. I am curious under the similar budget, what will make the proposed method stand out.

**Questions:**

1. Is there any prior work that uses MCMC, deep learning, or traditional numerical solvers for correcting the initial solutions? If there exists some, the authors should discuss and possibly compare with them.
2. Can the authors justify the improvement drop in Fig. 3?
3. Have the authors tried variants of PINN and what are the errors?

---

> ### Author Response · Authors · 2025-11-21
>
> We thank the reviewer for the thoughtful feedback. We understand  the review’s major concern on the novelty of the “Law of Defect,” the distinction between training and inference, and potential gaps in the experimental comparisons. We apologize if the specific context of high-dimensional solvers was not clear in the initial submission, as these concerns appear to stem from a misunderstanding of the unique constraints imposed by the curse of dimensionality.  We modified our manuscript as follows
> - clarifying why inference-time scaling matters and how our proposed law of defect fundamentally differs from prior defect-refinement techniques.  **We revise the terminology from the “law of defect” to the Structural-Preserving Law of Defect, highlighting the structural preservation that is central to our contribution.**
> - In the Appendix, we now include 10× repeated experiments for all methods, reporting 95% confidence intervals and pairwise t-test p-values, demonstrating that our improvements are statistically significant across benchmarks.
> - We further added Hexagonal Binning visualizations of pointwise error differences, showing that SCaSML improves performance on the vast majority of test points, confirming that our gains are not isolated to a few spatial outliers but are broadly distributed throughout the domain.
> - We have added a "Fixed Computational Budget Analysis" in Appendix G.6. In this experiment, we restricted the baseline PINN and SCaSML to the exact same total wall-clock time (Training + Inference). The results, visualized in Figure 17, show that SCaSML consistently achieves lower error than the baseline PINN under the same budget.
> - We have added a "Appendix A: Notations" section to centralize and define all symbols.
>
> ## Regards novelty
> Regarding novelty, the reviewer correctly points out that the idea of defect correction is classical in numerical analysis. However, all prior defect correction methods we are aware of are grid-based. They are fundamentally low-dimensional and become computationally infeasible in the high-dimensional settings our paper tackles, which reach up to 160 dimensions. In this high-dimensional regime, the only hope is relying on probabilistic, grid-free Monte Carlo solvers based on the Feynman-Kac formula. The central technical challenge is that a valid Feynman-Kac representation exists only for semi-linear PDEs. Classical defect correction methods rely on Newton-type linearizations that generally destroy this semi-linear structure and the positive semi-definiteness required for the diffusion coefficient (when the nonlinearity is dependent on the gradient).  Our core contribution, the "Law of Defect," is the derivation of a correction PDE that naturally preserves this semi-linear structure. This structural preservation is mathematically non-trivial and is precisely what **enables defect correction to be combined with high-dimensional Monte Carlo solvers for the first time**.  Therefore, the novelty is not the sampler itself—the MLPs are indeed existing—but **the derivation of a high-dimensional, MC-computable correction law that classical methods cannot produce**.
>
>
> To the best of our knowledge, this work is the first to achieve defect correction in high dimensions while preserving Monte Carlo computability. **If a prior reference has accomplished this structural preserving, we would appreciate the reviewer pointing it out.**  Finally, the reviewer’s question about tuning the PINN further illustrates why our hybrid approach is effective. Neural networks have a spectral bias toward low frequencies, leaving a high-frequency defect that is hard for NN training but ideal for Monte Carlo, whose rate is independent of smoothness. SCaSML lets the surrogate handle the easy low-frequency part while MC efficiently averages out the remaining high-frequency error. As discussed in lines 170–176, to the best of our knowledge, **ours is the first work to demonstrate an improved convergence rate for defect correction. This stems from the fact that our Monte Carlo solver does not require smoothness of the defect term**, which typically contains high-frequency components that are difficult to approximate.

---

> ### Author Response · Authors · 2025-11-21
>
> ## Regards Computational Budget
> To answer this empirically, we have added a new "Fixed Computational Budget Analysis" in Appendix G.6 of the revision. In this experiment, **we restricted both the baseline PINN and SCaSML to the exact same total wall-clock time (Training + Inference). The results, visualized in Figure 17, show that SCaSML consistently achieves lower error than the baseline PINN under the identical time budget.** This occurs because training deep networks suffers from diminishing returns; simply training the baseline longer does not yield a proportional improvement because the remaining high-frequency defect is difficult for neural networks to fit due to spectral bias. In contrast, our Monte Carlo correction converges at a rate independent of the solution's smoothness. Furthermore, our Corollary 4 proves that SCaSML achieves a faster asymptotic convergence rate of O(m^(-gamma-1/2)) where $m$ is the number of collocation points. **(We numerically examine this on GP numerically in Figure 4.)**  This multiplicative speedup explains why allocating marginal compute to inference-time correction yields a higher return on investment than allocating it to further training. We have also added Appendix G.5, reporting standard deviations and p-values (p << 0.001) to confirm these improvements are statistically significant.
>
> Second, this highlights another practical power of inference-time scaling. Training compute is fixed; once the surrogate PINN is trained, its accuracy is locked in. Retraining a PINN is far more costly—it requires new GPU cycles, hyperparameter tuning, and engineering time—whereas inference-time compute is cheap and fully elastic, making a 10× inference budget vastly more economical than a 10× larger or retrained PINN. In contrast, inference computing is also more flexible. With SCaSML, the user can dynamically decide how much accuracy they need. As shown in Figure 3b and our new Figures 10-13, allocating more inference-time samples progressively improves the solution.  This allows for a flexible trade-off between speed and accuracy on demand, without any retraining. Figs. 14–16 further show that even with the same number of collocation points, SCaSML achieves substantially better accuracy and faster convergence rate compared to GP surrogates.
>
> ---
>
> Thanks again for your efforts into reviewing this paper. We hope we have addressed your concerns regarding novelty and experiments and have incorporated them in the manuscript. We are more than happy if you can reconsider the review rating.

---

> > ### Author Response · Authors · 2025-11-27
> >
> > Dear Reviewer RnQk,
> >
> > May we kindly ask whether our rebuttal has sufficiently addressed your concerns regarding novelty?
> > For convenience, we summarize our clarification below.
> >
> > Regarding novelty.
> > While defect correction is indeed classical, all existing methods are grid-based and thus infeasible in the high-dimensional settings we study (up to 160 dimensions). In high dimensions, only grid-free, Feynman–Kac–based Monte Carlo solvers are viable. The core difficulty is that a valid Feynman–Kac representation requires the PDE to remain semi-linear, whereas classical Newton-type defect correction destroys this structure and breaks Monte Carlo computability.
> >
> > Our main contribution—the Law of Defect—derives a correction PDE that preserves the semi-linear structure, enabling defect correction to be used with high-dimensional Monte Carlo for the first time. The novelty is therefore not the sampler itself, but the derivation of a high-dimensional, MC-computable correction law that classical methods cannot produce. To the best of our knowledge, no prior work achieves this; if one exists, we would appreciate being pointed to it.
> >
> > The reviewer’s question about tuning the PINN further highlights why our hybrid method is effective: neural networks handle low-frequency components well but leave a high-frequency defect that Monte Carlo, whose rate is independent of smoothness, can average out efficiently. As noted in lines 170–176, ours is also the first work to show an improved convergence rate for defect correction in this setting.
> >
> > Warm regards

---

### Author Response · Authors · 2025-11-27
**Baseless Allegation from Reviewer Wcpd**

Dear Area Chair,

We would like to provide several clarifications regarding Reviewer Wcpd’s latest official comment.

1. Our revision fully complies with the Author Guide

The Author Guide states:

“The revised version shouldn’t read like a different paper compared to your original abstract submission. If so, the paper will be rejected.”

Our revision is clearly within these guidelines. The only changes made were:
- Clarifying sentences—explicitly connecting our method to the defect-correction idea (even though the actual implementation differs entirely from classical defect-correction).
- Adding standard statistical-significance reporting in experiments.
- Adding a budget-comparison figure, again for clarity—not changing any method, theory, or results.

**None of these changes alter the contributions, structure, algorithms, theorems, experiments, or conclusions. Our paper does not read like a “different paper” under any reasonable interpretation.**

**Thus, the reviewer’s assertion that our revision “differs substantially from the initial paper” is a baseless allegation contradicted by the content of the revision.**

2.Procedural concern: the reviewer pre-committed to rejection before the rebuttal process

In the original review, Reviewer Wcpd wrote:

“I believe that these changes are beyond the scope of a revision during the rebuttal period.”

This statement was made before seeing any revision, effectively establishing a rejection stance in advance. In the official comment, the reviewer repeats this predetermined position nearly verbatim.

This indicates that the reviewer did not evaluate the actual revision, but instead maintained a conclusion that had already been asserted prior to receiving the revised paper. This approach is inconsistent with the purpose of the rebuttal period, whose role is precisely to allow authors to clarify misunderstandings.

3.The reviewer’s rationale for this predetermined stance is also unfounded

The reviewer argues that because the initial submission did not explicitly mention a  line of related works that shares a conceptual idea (but whose actual methodology is entirely different), the revision must be “beyond scope.”

We respectfully submit that:

- Adding clarifying discussion of conceptual relations is exactly what the rebuttal period is for.
- It remains the same method, the same analysis, and the same contributions.
- It does not justify the pre-committed statement “these changes are beyond the scope…,”
- And certainly does not justify the baseless allegation that our revision “reads like a different paper.”
- We therefore believe this conclusion is unfair and not grounded in an accurate assessment of the revision.

Finally, we would like to note that giving a clear rejection solely because **the initial submission did not explicitly cite a line of related works that shares only a conceptual idea—but uses an entirely different methodology—does not match our experience with reviews at major ML conferences. Such issues are normally resolved through clarification during rebuttal, not treated as grounds for rejection in the absence of any scientific concerns. We therefore respectfully ask the AC to review the strength of this assessment.**

We respectfully request that the AC evaluate the paper based on the objective content of the revision and the intended function of the rebuttal process.

Thank you for your time and careful consideration.

---

### Author Response · Authors · 2025-12-03
**New Revision: Cleaned Appendix**

### Notation cleanup and consistency.
We have revised the notation system throughout the paper, harmonizing all symbols, removing ambiguous overloads, and adding a dedicated notation table.  All proofs referring to these objects have been updated accordingly.

### Added Budget Comparison

We updated a budget comparison (Appendix G.8) using a larger PINN, evaluating both training cost and end-to-end inference cost.

### Proof Sketch (added in the revision)

To make the logic behind our improved complexity bound fully transparent, we added a concise proof sketch summarizing the key steps:


**(1) Multiplicative complexity structure of MLP.**   Following [Hutzenthaler, 2021], the computational complexity of MLP depends *multiplicatively* on:   (i) the Lipschitz constant of the nonlinearity, and   (ii) the magnitude of the source term.  This occurs because nonlinearities propagate and amplify variance through successive Picard iterations. We show that the defect PDE inherits exactly this multiplicative structure.

**(2) Reduced source magnitude.**   By definition of the modified nonlinearity, the source term of the defect PDE is the surrogate residual. As the surrogate becomes more accurate, this residual decreases and the variance of the Multilevel Picard estimator reduces proportionally.

**(3) Preservation of regularity (no additional smoothness assumptions).**   We shows that \(\breve{F}\) inherits the same Lipschitz constants as the original \(F\). Thus, the refinement step does **not** introduce any additional regularity requirements; the MLP solver for the defect PDE operates under the same smoothness assumptions as the classical setting.

**(4) Final \(L^2\) error bound.**
Combining points (1)–(3), Theorem (Final-Nonrecur-Bound) shows that the total \(L^2\) error factorizes as:
$$
\text{(standard MLP complexity)} \times \text{(surrogate approximation error)}.
$$
Hence, as the surrogate improves, the overall convergence rate accelerates automatically.

We have also reorganized the structure of the proof to follow the new proof sketch more clearly. We apologize for any confusion caused by the earlier presentation. We highlighted all our changes further in **orange**.  We hope the AC will evaluate the paper based on these substantial clarifications and improvements made in the revision.

Thank you for your time and careful consideration.

---

### Meta-Review · Area_Chair_i5GM · 2025-12-16

**Summary:**

This paper proposes a method to improve PINN-based PDE solvers focusing on improvements at inference time via a "defect correction" method. The errors/defects are formulated and evaluated via a stochastic simulation, the output of which is used to modify the solution. The advantages of these approach are motivated via Monte Carlo sampling estimates, come with an extensive theory, and are demonstrated for a nice and broad range of sample problems.

The paper had a largely positive reception even before the cutoff period, one exception being the "reject" by reviewer Wcpd. The authors have argued that the criticism ia "baseless & unfair", which is probably not the best choice of words. Nonetheless, I was also surprised that the extensive updates did not result in any change of the score. Given the positively inclined reception by the other reviewers, I think this paper is somewhat borderline, but nonetheless a good candidate for inclusion in the ICLR'26 program.

**Reviewer Concerns:**

The reviewers raised a range of concerns about clarify, scope of results and ablations, which were - as far as I can tell - addressed by the revision.

A point that was raised, and that remains a concern is the accuracy of the surrogate model itself.

The difficulty of understanding paper and results was also mentioned several times, and remains a concern (clarifying all aspects is potentially a larger job).

**Reviewer Scores:**

I believe the following post rebuttal scores are a likely outcome for this submission:
* Kghx - 8
* RnQk - 4
* op81 - 6
* Wcpd - 4
Which puts the paper slightly above the bar.

Given all reviewer's concerns about clarity of the exposition, I can highly recommend that the authors further improve the clarify of their submission after acceptance.

---

### Decision · Program_Chairs · 2026-01-26

Accept (Poster)